# A Review of Particle Shape Effects on Material Properties for Various Engineering Applications: From Macro to Nanoscale

Ugur Ulusoy

Chemical Engineering Department, Engineering Faculty, Sivas Cumhuriyet University, TR58140 Sivas, Turkey; uulusoy@cumhuriyet.edu.tr

**Abstract:** It is well known that most particle technology studies attempting to predict secondary properties based on primary properties such as size and shape begin with particle characterization, which means the process of determining the primary properties of particles in a wide spectrum from macro to nanoscale. It is a fact that the actual shape of engineering particles used in many industrial applications or processes is neglected, as they are assumed to be "homogeneous spheres" with easily understood behavior in any application or process. In addition, it is vital to control the granular materials used in various industries or to prepare them in desired shapes, to develop better processes or final products, and to make the processes practical and economical. Therefore, this review not only covers basic shape definitions, shape characterization methods, and the effect of particle shape on industrial material properties, but also provides insight into the development of the most suitably shaped materials for specific applications or processes (from nanomaterials used in pharmaceuticals to proppant particles used in hydrocarbon production) by understanding the behavior of particles.

**Keywords:** non-spherical particles; spherical particles; shape characterization; aggregate; pharmaceuticals; nanoparticles; aggregate; proppants; metallic powders; industrial minerals

## 1. Introduction

While a powder is termed as a dry discrete material less than 1 mm [1], a particle can be technically described as any relatively small subdivision of matter, with a diameter between a few millimeters and a few angstroms [2]. In this context, particles vary greatly in size from macroscopic particles that can be seen with the naked eye to nanoparticles that can only be seen with extremely powerful microscopes.

Every year, huge amounts of particles (Figure 1) are treated by most industries including mining, energy, petroleum, paper, paint and plastic, ceramics, food, construction, pharmaceuticals, textiles, chemical, farming, and waste products [3]. Thus, particles and particle technologies affect life deeply because particulate materials such as soil, aggregates, cement, highways, brick, ceramics, glass, pigments, paint, minerals, and metals are indispensable elements of modern society. A great variety of industries, including energy, environmental, civil, pharmaceutical, chemical, petroleum, agricultural, food, plastics, cosmetics, battery, mining, minerals, metallurgy, and advanced materials, are based on particulate materials, powders, or bulk solids [4], which are used in numerous processes such as coal combustion, dissolution, catalysis, energy, agriculture, the production of industrial carbon, processing of food, minerals, ceramics, cement and potassium, the manufacturing of paints, pulp, paper, synthetic fibers, petrochemicals, glass, plastics, rubber, and pharmaceuticals. In the United States alone, industrial production influenced by particle systems is composed chiefly of ten leading industries [5] such as chemicals and related products, textiles, paper and related products, food and beverages, metals, minerals, and coal [6]. The current value of these industries is estimated at more than USD 2 trillion and is expected to rise by 5–10 times in the coming 10 years [7].

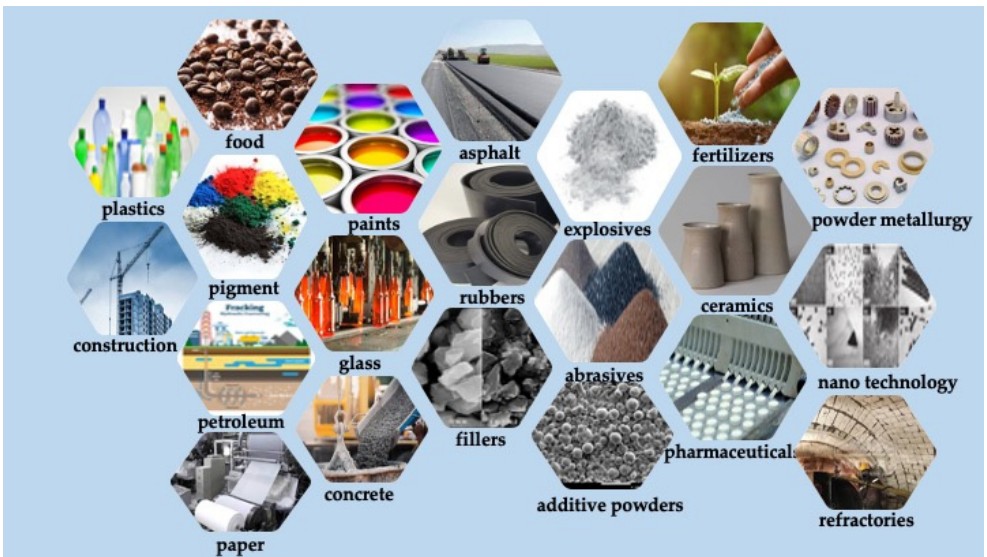

**Figure 1.** Particle-based industries.

The properties of powders and particulate materials are generally divided into two classes (Figure 2); primary properties resulting from the composition of the material, and secondary properties associated with systems of individual particle collections whose interior surfaces interact with a gas, usually air. Since the primary particle properties such as shape and density, fluid viscosity and density, along with the concentration and state of dispersion, control the secondary properties such as particle settling rate, compaction, flowability, sintering, powder rehydration rate, filter cake resistance, etc. [8], particle systems, which are related to several industrial processes [9], can not only be described by single particle properties but also by bulk properties and their impacts. In other words, the physical characteristics of individual particles, such as their sizes, shapes, and size distributions and also how they interact with one another, determine the bulk properties [10] such as mechanical, thermal, electrical, magnetic, optical, acoustic, and surface physicochemical properties [11]. Thus, individual particle properties, which are crucial to product characteristics [12], include not only physical properties such as size, shape, hardness, density, surface roughness, homogeneity, and density, but also include adsorption, biological, optical, thermodynamic, and dynamic properties [8,13,14].

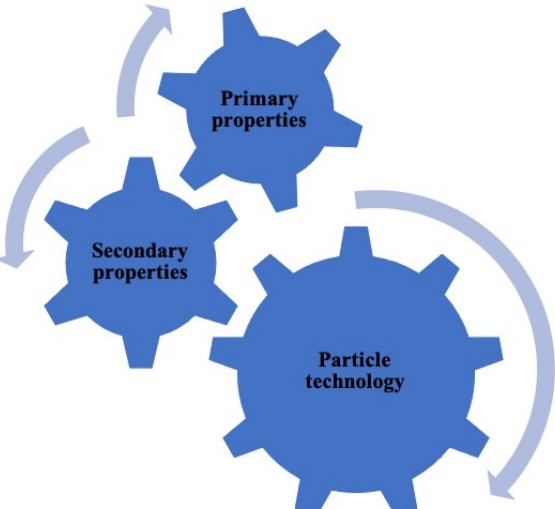

**Figure 2.** Properties of particulate materials.

From another point of view [15], particle properties can be classified into two categories: namely material and morphological properties, as given in Figure 3. Therefore, particle shape, as well as particle size, surface properties, mechanical properties, charge properties, and microstructure, are crucial physical attributes that play a role in measuring in terms of production and development.

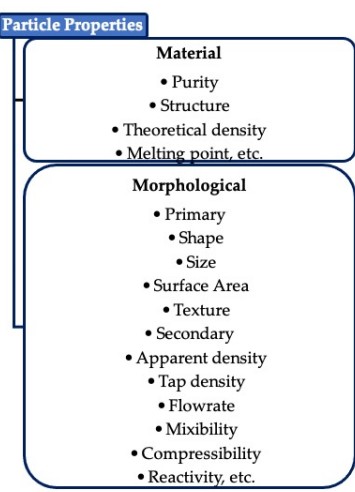

**Figure 3.** Categorization of particle properties (modified from [15]).

Thus, particle characterization, which means describing the primary properties of powders in a particulate system, forms the basis of all particle technology studies aimed at predicting secondary properties based on primary properties [14,16]. Moreover, powder or particle technology has emerged as a new engineering discipline, combining elements of microbiology, chemistry, formulation science, colloid and interface science, heat and mass transfer, solid-state physics, aerosol and powder science, and nanotechnology [17]. In other words, it is relevant to the systematic investigation of many processes involving the characterization, storage, transportation, mixing, fluidization, classification, and agglomeration of all particulate materials, whether in the dry state or suspended in a liquid. Since particle processes are a significant factor in the manufacture of various industrial products, advances made through the study of particulate systems by applying the principles of particle characterization have been beneficial in a wide variety of industrial processes (production of chemicals, batteries, catalysis, nanomaterials, pharmaceuticals, ceramics, and foods) [8]. Characterization of particle properties is needed for product quality management and process control as the quality of many products is governed by particle properties [16]. Thus, material characterization is the key factor for industrial research and development because it provides greater knowledge of the connections between the processing, structure, characteristics, and performance of materials. In addition, the evaluation of the behavior of powders during processing is made possible by a knowledge of process variables in conjunction with physical properties [18]. Therefore, the assessment of size, shape, and structure distributions is a crucial stage in process control and product specification since each industrial mineral particle shape confers benefits to the finished product [19–21]. Moreover, selecting the right mineral and optimization of parameters (particle size, shape, micro-structure, color, and hardness) is necessary for providing the precise formulation needs of the consumers [22]. Above all, the characterization of particle properties is of prime importance as it not only provides a better understanding of the link between process performance and product quality but also facilitates the improvement of production performance [16]. Thus, particle characterization has been routinely employed in many industries for better quality control and understanding of products, additives, and practices to enhance product efficiency, identification of fabrication and production problems, optimization of the performance of production processes, and improvement of the production or increase capacity. Today, many industries recognize the importance

of measuring particle shape as well as particle size for a full understanding of products and processes, since particle shape can influence many properties in various product materials, e.g., reactivity and dissolution in pharmaceutical active components, flowability and handling in drug delivery, sinterability in ceramic filters, abrasive performance in SiC wire saws, and texture and feel in food ingredients [23]. Moreover, the best utilization of processes relies on a complete understanding of particle science and technology, which gives crucial knowledge for the characterization, formation, handling, processing, and use of a wide range of particles [9]. For example, the fact that individual particle morphologies vary greatly has a direct effect on the mechanical and physical properties of collective particles of materials [24]. The behavior of bulk solids or multi-particulate systems is significantly affected by particle morphology because many of the physical and chemical properties of these systems rely on particle shape and surface geometry [25]. Since the behavior of particles and their sustainable use are significantly influenced by particle shape and particle size distribution (PSD) properties [26], applications in numerous engineering disciplines need knowledge of the particles' shape as well as particle size. There are numerous examples where particle shape has been incorporated in materials science and technology, as well as many other branches of science and technology, from aggregate to pharmaceutical powders [27,28].

Although the first shape characterization studies started with Wadell, [29] who studied rock particles in the 1900s, little attention has been paid to its effects on the process and the effects of particle size have attracted more research attention for many reasons. First, it is difficult to characterize the complex shape of real particles to measure the effect of particle shape, as there is no standard for shape characterization [30]. Second, when using non-spherical particles, it is difficult to evaluate the effect of the shape variable separately from other variables [31]. However, for industrial materials, very seldom are the particles spherical [32]. Although shape can be described as the external form, contours, or outline of an object regardless of its absolute size, the distinction between size and shape is not always obvious. In addition, the shape of the particles change as they become smaller, mostly coming out of the crushers and mills (Figure 4). In other words, not only particle size but also particle shape is used to best characterize raw materials. Since particles are irregular or deviate from the ideal sphere in various industries, the prediction of particle size and PSD of any particulate material is influenced by particle shape. Therefore, statistics based on one size value are insufficient to reflect the real performance of the particulate material [33], and particle size without shape analysis can lead to substantial underprediction of average particle size and inaccuracies in PSD [34] for non-spherical particles.

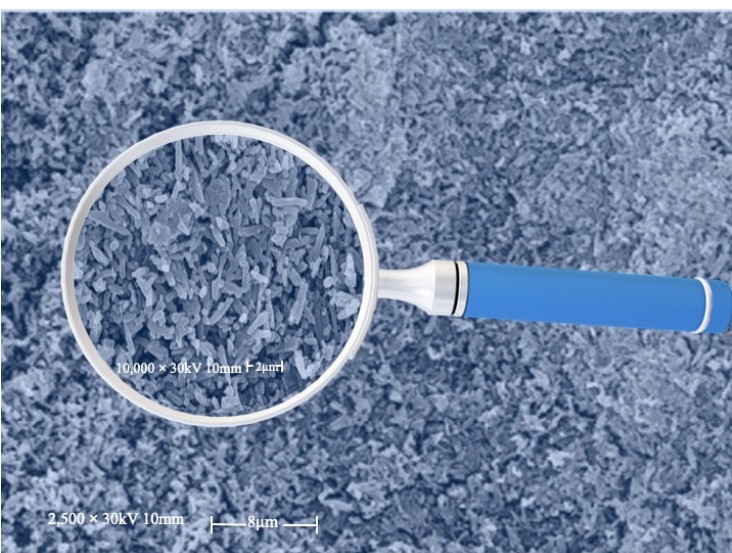

**Figure 4.** A $10^4$ times magnified image of aragonite particles by SEM (modified from [35]).

For example, Figure 5 shows the results of two different starch samples, which are also used as a filler in the paper industry and as a typical excipient for pharmaceutical applications [36], namely the PSD on the left (Figure 5a) and the aspect ratio distribution on the right (Figure 5b). As seen in Figure 5a, the median ($d_{50}$) value is nearly the same, although the distribution is not the same. On the other hand, when looking at the shape analysis results in terms of aspect ratio (Figure 5b), it is apparent that sample 1 comprises rounded particles, while sample 2 has a substantial number of elongated particles [37].

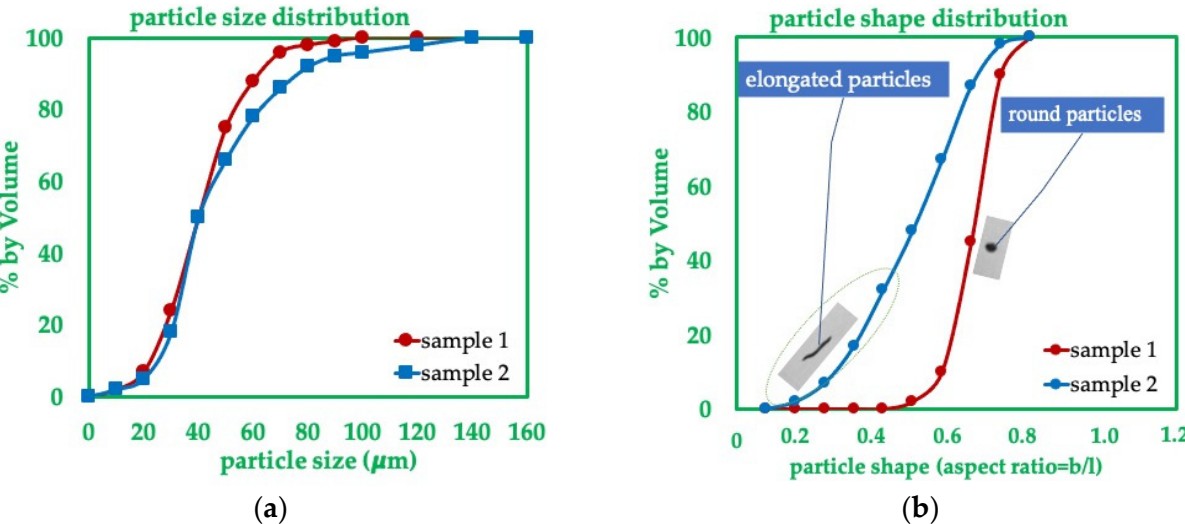

(a)　　　　　　　　　　　　　　　(b)

**Figure 5.** Two different starch samples have (**a**) similar $d_{50}$ sizes but (**b**) different aspect ratios (modified from [37,38]).

Another example can be given in Figure 6. According to the particle PSD histogram shown in Figure 6a, which assumes that all particles are spherical, it is seen that the median ($d_{50}$) value of the sample is 30 µm. Despite having similar spherical diameters (Figure 6a), particles #1 and #2 have significantly different shapes, as evident from Figure 6b. Therefore, they will likely behave differently in a subsequent process [34]. Since natural or engineered colloidal particles frequently have irregular or non-spherical shapes, and the theoretical assumption is based on ideal spherical particles, the shape of the non-spherical particles is expected to change the interaction between particle, fluid, and surface, thus changing the behaviors or properties of particles in the processes or applications [39].

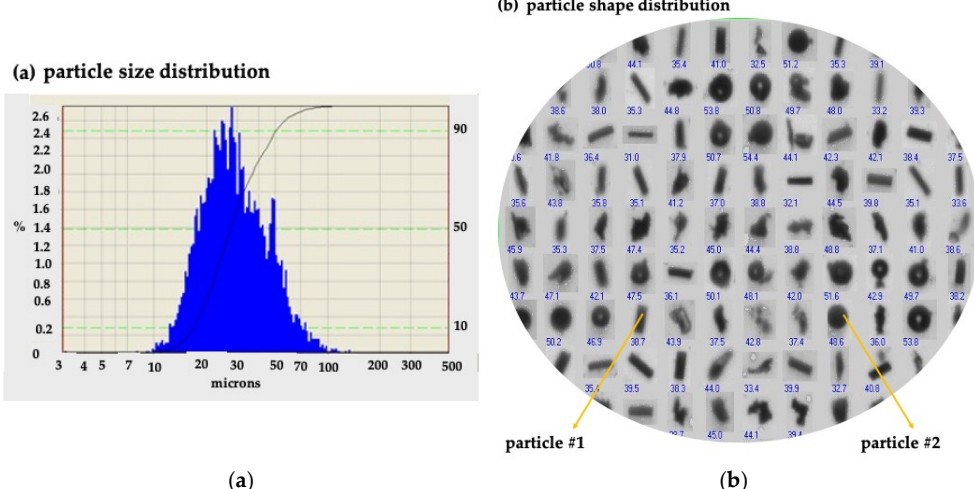

(a)　　　　　　　　　　　　　　　(b)

**Figure 6.** (**a**) PSD, (**b**) shape distribution, and selected particles have nearly the same size (modified from [34]).

A better comprehension of particle behavior depends on the systematic evaluation of particle shape, since the shape is a common variable influencing particulate material's behavior in the processes [40]. Therefore, there has been an increasing effort to understand and control particle processes by investigating the effect of particle size and shape on the behavior of granular materials in the process. Although it is obvious that particle shape is significant, there is still no standard approach for routine analysis of particle shape distributions in the field of mineral processing [41]. Moreover, the significance of particle shape in many industrial materials is frequently disregarded or undervalued. However, particle shape may play a key role in transforming a given industrial material into top-performing products. In addition, the inability to objectively describe the shape hampers the progress of involving the effect of particle shape in many technical analyses. Since particle shape determines lots of properties of industrial materials, manufacturers should also analyze and change particle morphology to reach the best degree of performance. For example, it has been reported that particle shape affects particulate materials' properties according to numerous works in the engineering field, e.g., aggregates' technical qualities in cement and concrete [42–44], properties of catalysis, drug delivery [45], pharmaceutical manufacturing [33,46,47], mechanical properties of powder metallurgy [48], soil mechanical properties [40,49–51], filler properties of industrial minerals for pigment, paint, plastic, rubber, paper, and coating [19–21,35], and nanomaterials [52–54], etc. Since it is essential to quantify fine material attributes to assure the physical characteristics of the products for the material utilized, particle shape acts as a vital factor that must be accurately described and reported as part of mineral characterization.

However, studies addressing the effect of changing particle shape properties highlighted in the examples above appear to be insufficient. Moreover, the effect of irregular particle behaviors on their properties in the processes is not yet well understood, although a clear understanding of particle behavior is of prime importance to effectively explore the potential applications of particles. Therefore, the importance of particle shape on the properties of various materials in numerous industries and applications is presented in this review by addressing how the shape is related to their properties. In other words, in this review, studies on particulate materials used in various industries, and on which particle shapes have better properties or which application areas or processes are more effective, are compiled. In addition, the aim of this review is not only to promote a deeper and more comprehensive investigation and application of shape characterization in shape-based material research, but also to help better understand and control processes in different applications of different materials with varying sizes in various engineering fields, from aggregate to nanomaterials, as well as to encourage optimization and the development of new processes. Thus, this review summarizes current research efforts in various engineering disciplines and also outlines future research needs. Additionally, it is anticipated that this review will serve as a springboard for fresh concepts and a starting point for researchers looking to investigate shape-based materials across various material science and development fields. Moreover, this review is expected to serve as a guide when choosing the right shape for a particular particulate material that exhibits the necessary functions, properties, or industrial applications.

## 2. Shape Parameters

According to the Oxford English Dictionary [55], shape refers to the external form, contours, or outline of something regardless of its absolute size, whereas International Standards [56] defines particle shape as the envelope formed by all the points on the surface of the particle. As seen in Figure 7, the particle shape of the powders can be very diverse, e.g., acicular, angular, flaky, lamellar, granular, irregular, spherical, modular, dendritic, needle-like, and crystalline [1,8,18].

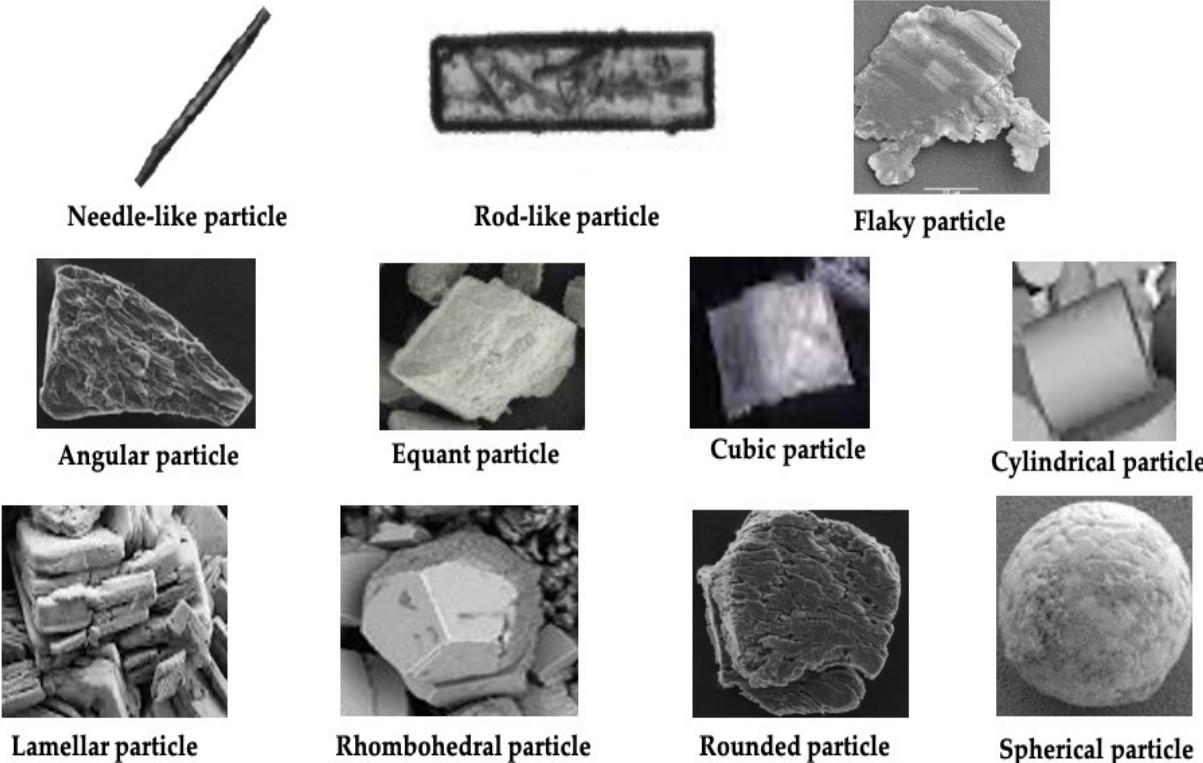

**Figure 7.** Industrial particles of various shapes.

While Podczeck [57] proposed that the geometry of particles can be defined by four main characteristics, such as size, shape, roundness, and roughness, which are conceptually different, Neal and Russ [58] claimed that the term shape relates to the geometry of the particles regardless of their absolute size. In many studies, shape and morphology are intertwined and are often used interchangeably. In fact, a geometric description of a particle's size, shape, internal structure, and surface qualities is described as morphology [17]. The nuance is that morphology also encompasses roundness and roughness in addition to shape.

Although basic facts regarding the shape of a particle can be categorized as qualitative (elongated, spherical, flaky, etc.) or quantitative, the latter is more important in engineering because of repeatability [59]. In addition, the shape of particulate material is frequently described by using shape descriptors, which are traditional combinations of particle dimensions such as length and width [57].

A common approach [50,60,61] describes particle morphology using two-dimensional (2D) projections, which can be classified as three distinct observation levels. The scale at which an analysis is performed determines the size of the corners and edges. Deviations are discovered and assessed by examining the circles (Figure 8) created along the particle's boundary. The first category is given the term of form or sphericity, which signifies the similarity of the particle's overall shape to a sphere at the macro-scale, and generally, on this scale, the particles are described as spherical, flat, and elongated. The second category is defined by roundness, which is the antonym of the term angularity and implies the particle's corners at the mesoscale. Finally, the third category can be represented by roughness or smoothness, which shows the surface texture of the particle at the microscale [59].

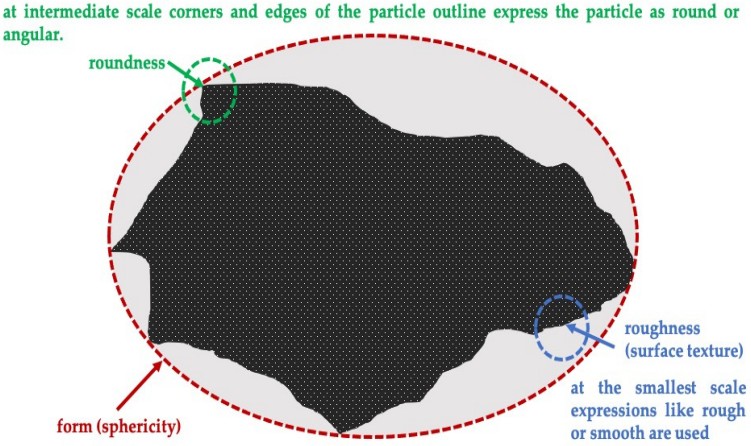

**Figure 8.** Description of a particle morphology based on three different scales (modified from [50,56,60,61]).

While the ratio of the longest to the shortest Feret diameter is known as the elongation ratio, the aspect ratio, which is calculated by the ratio of the greatest axis' length to the smallest axis' length, is a crucial factor for describing fundamental particle shape [30,62]. In fact, the aspect ratio is the inverse of the elongation ratio. Thus, both are sensitive to general geometry. Other shape descriptors are sphericity and convexity, ellipseness or ellipsoidness, since they can be affected by form, roundness/angularity, and texture [63]. In addition, circularity measures how far particle contour deviates from a circular pattern, while sphericity gives the identic information as three-dimensional (3D) shape factors. Sphericity and roundness frequently correlate with shape-related characteristics such as the volume and surface area of the particle [64].

Sphericity and roundness [29,65,66] are generally described in 2D based on particle projections or silhouettes [63,67]. The use of these 2D descriptors has become widespread since it is easy to characterize macro-scale form and medium-sized roundness [61,68]. However, there are further definitions for 3D data [69–73]. Since it is reported that only the convex areas of the particle are quantified for roundness, while the edges and corners are not taken into account in the concave areas, angularity is widely utilized based on 2D data to express the sharpness of a particle's edges and corners [74–77]. However, there are also various methods employing 3D data that are documented in the literature [74,78–80].

### 3. Measurement Techniques Used for the Shape Characterization of Particles

Since particles used in different engineering fields have unique properties depending on their size, shape, and PSD, it is important to accurately measure their properties to ensure that they are used to optimum benefit. Even though the influence of particle shape on behavior is recognized, there is no universal approach to shape characterization [81].

The measurement of shape-related features has been the focus of much research since 1935 through Wadell's [29,65,66] well-known works. Since virtually all fields of applied science and engineering might benefit greatly from particle shape characterization as an analytical tool, measurement techniques for the characterization of particle shapes have over time been developed within many areas of application, either for basic research or industrial purposes [82].

Different analytical techniques, which have different functionalities, capabilities, advantages, and limitations, from simple to advanced ones, are used to define shape properties of particles for several industrial materials in engineering disciplines from macro to nanoscale level. However, the sample being examined and the information required is what decides which technique is best. In this review, all the techniques used in shape characterization can be categorized and given as follows.

### 3.1. Classification Charts

Since the manual shape assessment of thousands of particles is laborious, historically, the chart consisting of reference particles developed by Krumbein [83], which is the initial method for recognizing particle shape, is considered the simplest way to visually assess the roundness of particles. As shown in Figure 9, the morphologies of the particles can conventionally be described by the sphericity and roundness parameters based on 2D projections [40,83–86].

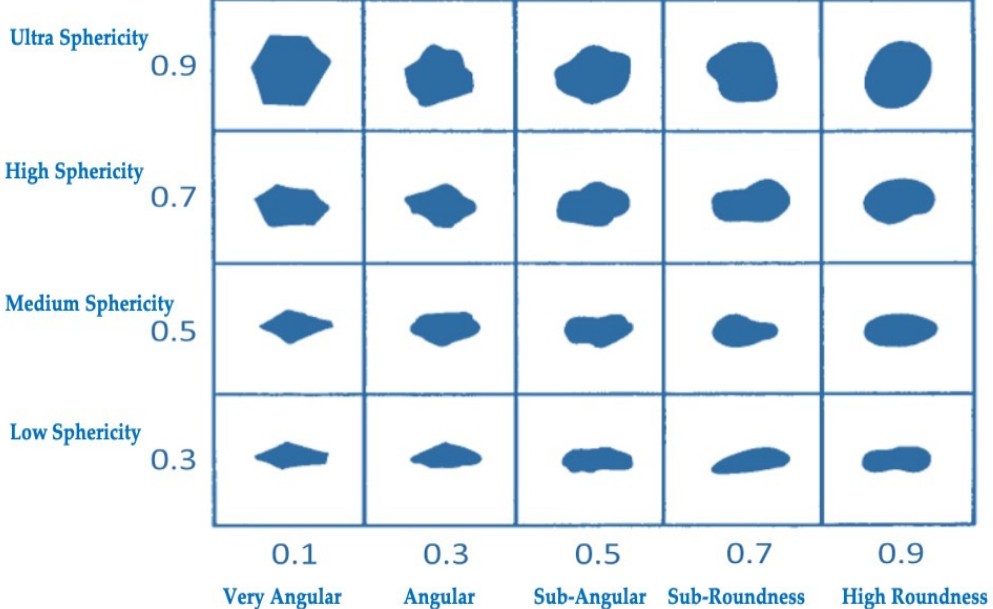

**Figure 9.** Classification chart based on roundness and sphericity of particles (modified from [40,84,87,88]).

This method is only based on the comparison of visual images of particles by microscopic evaluation, without any image analysis software, and the shape results of the particles examined do not rely on any quantitative analysis; therefore, it is considered insufficient for the evaluation of any particulate material's appropriateness for a specific application. The main disadvantages of this method are that it is boring, it is very sensitive to the individual perception of the user, and it is time-consuming. Moreover, it is also limited to larger-sized particles, since it is impractical to measure three dimensions of small sand particles [89]. Although the comparison charts for roundness determination have been widely recognized in various engineering disciplines, this technique has a high degree of operator subjectivity, which reduces its accuracy. Thus, quantitative measurements are advised to minimize subjectivity-related errors.

### 3.2. Two Dimensional Measurements

Two dimensional measurements in terms of angularity, roundness, and sphericity, have been extensively employed in shape analysis of engineering particles for many industrial fields due to the ease of image-collecting techniques such as photography [50,61,90] by using a microscope (optical, stereo, scanning electron, and transmission electron) or a camera. Many 2D approaches based on the roundness parameter have been created nowadays [67,91,92].

The data obtained are 2D, since the measurement is based on particle projection or silhouette, since the thickness, which is the particle's third dimension, is frequently neglected, eroding the precision of the method used to calculate the shape factor of real-world particles. The issue becomes even more apparent when measuring wet and deformable particles mounted and polished to uncover their shape [93]. These techniques highly effectively

computerize Wadell's particle roundness approach, even though they still operate in 2D space. In fact, particle roundness is a 3D concept by nature; therefore, 2D approaches frequently produce results that are inconsistent with reality, since a single projection contour cannot accurately capture the shape of the entire particle and the projection angle is chosen arbitrarily [71,72].

### 3.2.1. Microscope Techniques

Ever since the microscope revolutionized science with its invention in the 1600s, it remains the most used technique for analyzing subvisible particles by enabling a comprehensive examination of them at various magnifications. In this method, the particles under investigation are placed on the slide in a stable position to ensure that the 3D particles are systematically analyzed in 2D with the minimum possible centers of gravity. The ability to automate Wadell's technique has been made possible by enhancements in optical image capture, which have significantly improved the processing of particle projections. Distributions of different size properties, such as projected-area diameter, minimum, and maximum Feret size, and aspect ratios, can be calculated by microscopy after representative samples were collected, molded, and polished [62]. However, it has been difficult to obtain accurate information regarding particle shape because most microscopic techniques only provide 2D images, particularly in the case of platy particles.

### Optical Microscope Technique

The earliest type of microscope was the light microscope, commonly referred to as the optical microscope. It is a non-destructive technique consisting of light in the visible spectrum and a lens system to magnify small samples. It is a highly sensitive technique for particle image analysis that permits the observation and quantification of individual particles [18], since it can demonstrate the precise shape of each particle down to micrometer size. Moreover, optical microscopy, which is mainly utilized for routine studies [94], can capture images of the particles when coupled with a camera and transfer them to a computer with image analysis software using shape characteristics such as aspect ratio and circularity.

Optical microscopes using the bright field, polarized light, and dissecting/stereo techniques are widely preferred in the field of mineral and concrete research. For instance, it is common practice to use thin-section optical microscopy to examine the microstructure of hardened concrete. It was also used for 3D shape analysis in many works [95,96]. Semiautomatic analysis of microscope images is frequently used in industries ranging from food to pharmaceutical biomaterials [97–99]. Its advantage over other methods such as SEM is that it provides a wide field of view, permitting relatively simple numerical calculations [100,101]. Although relative shape data could be collected with adequate accuracy using an optical microscope, some images have to be manually adjusted to assure accurate particle isolation and binary color processing [89]. However, unless a statistically significant number of particles are examined, manual microscopy should only be used to provide a qualitative sense of particle size and form. On the other hand, it is challenging to obtain a representative sample because a very small mass of the sample is measured in a labor-intensive procedure [102]. Moreover, since the lower resolution limit for its application to colloidal particles must be above 200 nm, it can only be applied to aggregates and large colloids [103].

### Stereo-Microscopical Analysis

A stereo-microscope is a kind of optical microscope which uses visible light (reflected light from the object being studied) to see a 3D view of opaque, thick, solid objects. It provides a 3D image that allows the viewing of the texture of a bigger specimen, in contrast to compound microscopes which provide a flat image. Stereomicroscopy relies on two light channels passing through the objectives and eyeglasses to function. Every light path offers a unique viewing angle (10–12°) to each of our eyes. These two very similar, albeit slightly different-angled, images are processed by our brain into a three-dimensional

view [104]. Although it has lesser optical resolution power, with a typical magnification of 6 to 75 times, it is widely utilized for several purposes because it can show 3D objects [105]. It has been used for morphological analysis of many materials in the field of food, mineral, and concrete research, e.g., cocoa beverage powder by steam agglomeration [106], sand particles for determination of the cement paste content in the concrete [101], and chromite particle morphology in a shaking table circuit (Figure 10) of a Turkish chromite beneficiation plant [107].

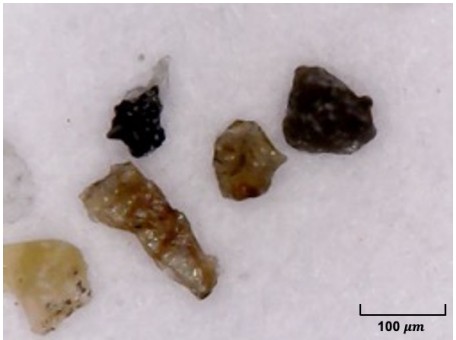

**Figure 10.** A stereo-microscope image of middling particles from a fine shaking table in a Turkish chromite beneficiation plant at 180 + 125 μm size fraction [107].

SEM-Based Techniques

Scanning electron microscopy (SEM), which is another technique often used to assess particle morphology, has extra properties, namely a broad depth of field and higher resolution compared to optical light microscopy [108,109]. Powder samples are sprinkled onto a carbon band before being inserted into the vacuum environment, which utilizes an extremely thin electron beam to inspect the specimen's texture while generating pictures from backscattered primary electrons and secondary electrons that are emitted [103]. The use of SEM in research in various scientific disciplines has been reported to have nearly exploded for a variety of materials, since a huge number of articles on the shape characterization of mineral particles have been documented in the literature. In other words, it has been reported that it can provide 2D particle shape characterization for thousands of particles used in various fields of science (Figure 11).

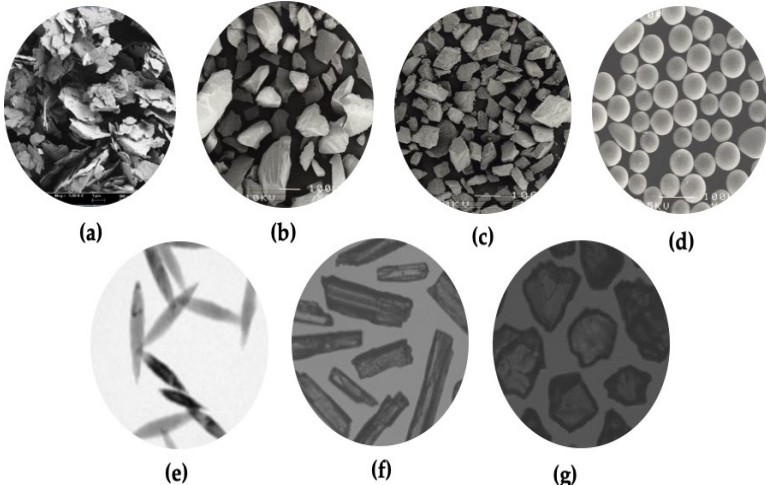

**Figure 11.** Particle shape characteristics of some mineral particles: (**a**) platy talc particles; (**b**) angular quartz particles; (**c**) irregular barite particles; (**d**) glass beads; (**e**) cigar-shaped particles [59]; (**f**) rod-shaped wollastonite particles; (**g**) equant quartz particles.

However, it has also been reported that shape characteristics are frequently determined by measuring a very small number (usually less than 100) of particles that are not always randomly selected, leading to higher statistical variation of results in various applications [110–112]. Furthermore, sample preparation is time-consuming, and images need to be manually adjusted to ensure each particle is clearly identified and not overlapped with other particles, reintroducing a subjective element for the quantification and making it less attractive than faster techniques [89]. In addition, some software tools such as COREL DRAW [113] can be used to digitize shape parameters by measuring the axes of non-overlapping particles (Figure 12) after each SEM image is taken and magnified by a certain amount, such as 400%. Assuming that the particle's projection has an ellipse-like shape [114], the area (A) and perimeter (P) can be calculated based on the measured length and width as shown in Equations (1) and (2) [115]. Some shape parameters, including elongation, flatness, roundness, and relative width, can be used to characterize particle shape by Equations (3) to (6) [114,116–119].

$$A = [(\pi LW)/4] \tag{1}$$

$$P = \pi/2[3/2(L + W) - \sqrt{(L.W)}] \tag{2}$$

$$\text{Elongation } E = (L/W) \tag{3}$$

$$\text{Flatness } F = [(P^2)/(4\pi A)] \tag{4}$$

$$\text{Roundness } R = [(4\pi A)/P^2] \tag{5}$$

$$\text{Relative width } RW = (W/L) \tag{6}$$

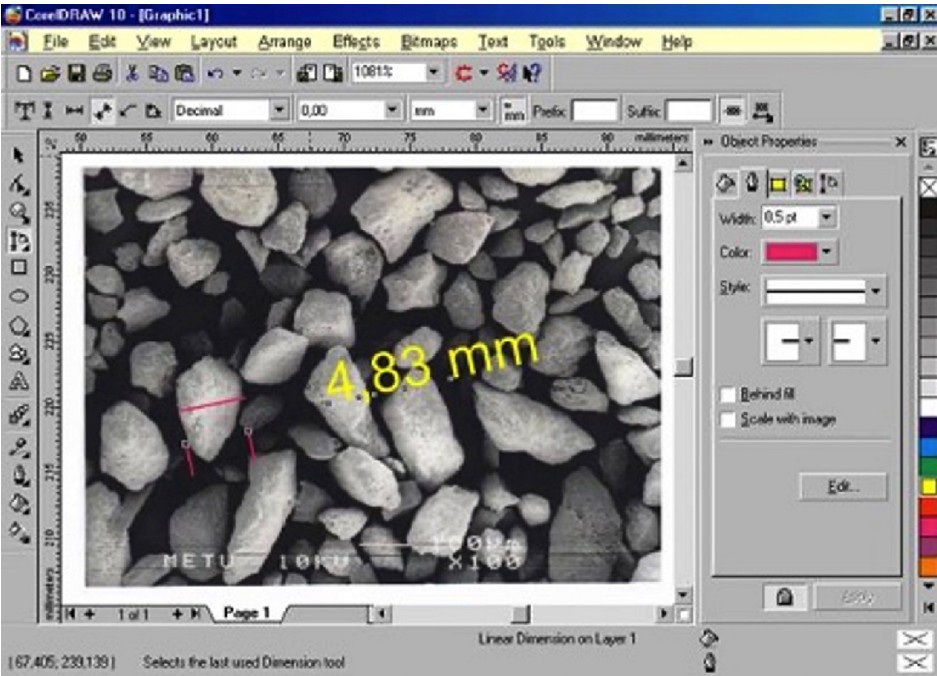

**Figure 12.** Measurement of axes of particles using software for the characterization of particle shape [113].

### 3.3. Three Dimensional Measurements

Although image-based techniques have been used to measure particle shape from 2D projections of 3D grains for practical purposes [40,90,91,120–122], the limitations of 2D particle projections to characterize 3D particle shapes have prompted researchers to develop 3D imaging techniques to better quantify particle shapes [123].

It has been reported that the direct use of 3D data provides a more complete answer than the previously proposed 2D methods [78], since the manual method was laborious and the visual comparison method was open to interpretation [123]. However, while 3D is the method that generates much more information related to particle shape, the resolution has also been reported to play a role in precision [59]. Therefore, it is possible to characterize shapes by analyzing 3D images, but it is thought to be difficult to obtain reliable data and reduce them to useful parameters [120,124,125]. Since the particle shape definitions were mostly put forward between 1920 and 1950, there were no methods available at the time to create 3D particle surface models and, for this reason, 3D particles had to be roughly represented by 2D projections for shape analysis. Even though 2D shape properties can be helpful for 3D shapes, the directions being projected have a big impact on them. Therefore, it has been reported that a great number of particles must be used for a statistically accurate analysis of roundness and sphericity [126].

Recently, Nie et al. [71,72] have suggested a novel approach for assessing the 3D roundness of actual particles quantitatively and objectively, based on the widely accepted Wadell's [66] definitions of particle corner and particle roundness. Once a triangle mesh was used to rebuild the particle, corner parts were identified by obtaining the surface vertexes with both a high local curve and a large relatively connected area. Then, corner parts were filled with spheres using a sphere-filling technique. Finally, 3D Wadell roundness was obtained by comparing the mean diameter of the filled spheres to the diameter of the greatest inscribed sphere.

### 3.4. Permaran Technique

A Permaran instrument (manufactured by Outokumpu Oy, Tapiola, Finland) measures airflow resistance (R) through a tightly packed bed of uniformly sized particles. Considering that resistance to airflow is related to the permeability of the sample in the container, the airflow resistance of ore samples and glass beads can be measured based on equal volume and compactness. Since the resistance to airflow of particles in a porous bed of samples relies on their acuteness properties, the Permaran technique can be used to measure the sharpness of the particles based on airflow resistance imparted by particles in a porous bed of samples [127–129]. The greater resistance values displayed on the scale of the Permaran instrument imply a greater acuteness degree as shown in Figure 13. In other words, sharpness can be considered as a deviation from roundness. As a result, the ratio of the flow resistance of ground particles to the flow resistance of reference particles (glass beads) is used to determine the acuteness degree of particles (Ac), while pressure drop ($f(\Delta p)$) throughout the sample bed is measured by a created liquid pressure gauge.

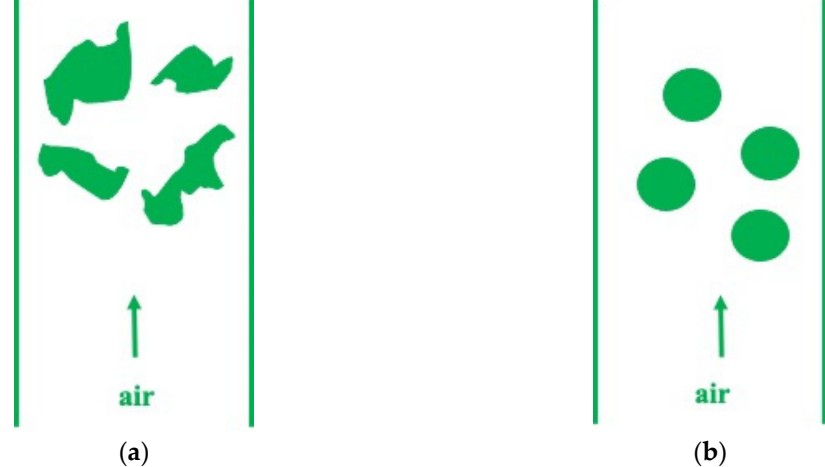

**Figure 13.** Concept of acuteness by Permaran technique: (**a**) more acute particles; (**b**) less acute particles (modified from [128]).

The empirical formula below (Equation (7)) is used to determine the sample bed's airflow resistance when the sample and bed volumes were maintained at constant throughout the measurements for comparison results of two different particulate materials [130];

$$R = [(0.01)f(\Delta p)^2 Rn] \tag{7}$$

where Rn is the normal flow resistance and is specified in the handbook as 1690 (g/cm$^4$.s), $f(\Delta p)$, is a factor of airflow resistance and can be read directly on the manometer scale of the Permaran instrument. Hence,

$$R = [(16.9)f(\Delta p)^2] \tag{8}$$

The acuteness degree of particles can be calculated based on Equation (9) [128];

$$Ac = (R/Rr) \tag{9}$$

where R and Rr refer to the measured air flow resistance value of the sample, and the airflow resistance value of reference glass beads, respectively.

### 3.5. Brunauer, Emmett, and Teller (BET) Technique

A well-known technique for estimating the surface area of a few grams of particulate material using nitrogen gas adsorption is Brunauer, Emmett, and Teller (BET) [108,131,132].

In the last decade, number-based estimation of aspect ratio for materials whose grain shape deviates from spherical (especially for highly elongated, flat, rod-like, disc-like, fibrous, or acicular grains such as talc or kaolin) can be possible thanks to the Hohenberger model given in Equation (10) [133]. This method, which is based on the measurement of specific surface areas and PSDs, has been a readily available, reproducible, and independent method.

$$\rho = \left[ (\delta m - 4).\varepsilon_{BET}.\, d_{50}.\, e^{-ln^2(d_{84}/d_{50})} \right] /2 \tag{10}$$

where $\rho$ is the representative aspect ratio of the platy particles, $\varepsilon$BET is the specific surface area measured by nitrogen gas adsorption (BET), $d_{84}$ is particle size at which 84% of a sample's mass is comprised of smaller particles, $d_{50}$ is particle size at which 50% of a sample's mass is comprised of smaller particles, and $\delta$m is the density of the mineral (t/m$^3$) [133].

It should be noted that Gantenbein et al. [35] have also used the model for non-porous (aragonite) and porous (palygorskite) particles. However, it has been reported that the application of this model is limited when samples are only available in small amounts for nitrogen gas adsorption [108,132].

### 3.6. Image-Based Techniques

Image analysis technique is defined as the process of obtaining information from pictures, photographs, or images using light, electron, or scanning force microscope. The measurement of the shape and size of a large number of particles can be made thousands of times faster, and more accurately and reproducibly than by manual techniques by computer analysis of digital images. An image analysis system can consist of a type of microscope linked to a photographic camera or video camera, along with a computer transforming the analog image into a digital one based on algorithms for the characterization of the shape of the particles defined in the image by determining the color or grey-shade of each pixel as well as estimating the particle perimeter and area [46].

The first image analysis techniques were developed in the late sixties by Moore [134] with the introduction of the computer image analysis approach, which offers a comparatively quick way of automatic analysis. Initially, it was not feasible to record or freeze images, requiring measurements to be performed "semi-manually" directly on the com-

puter. However, sample preparation, measurement, and data analysis were all carried out manually because automation for image analysis was not developed until recently. Therefore, it is frequently tedious and time-consuming, and does not have remarkable statistical reliability since the number of particles analyzed was very small [47]. Today, image analysis, which also allows the simultaneous measurement of huge amounts of data, is performed by saving the images in the fast image memory of the computer and measuring them instantly [135]. It has been reported that the configuration with advanced high-resolution imaging power is a more reliable and accurate tool for characterizing the particle shape [136,137].

By recording the images of the particles on the screen, the geometrical parameters of the particles, such as area, perimeter, and length, among others, can be obtained by the binary image processing algorithm (Figure 14). Algorithms, which may include several steps such as capturing an image, normalization of grey levels, segmentation, scrap and fill, contours [138] filtering, overlapping particle segmentation, border killing, hole filling, and debris removal, improve the image. Then, a straightforward grey-level thresholding procedure is used to choose the particles and transform the image into a binary image. While hole filling is used for obtaining the silhouette of a particle, separation of overlapping particles is achieved by segmentation. Removing any object in contact with the image edges to become invisible is accomplished by border killing. On the other hand, deleting unusual objects based on grain size or shape in the image is carried out by residue removal [139]. In a simple segmentation operation, greyscale thresholding is performed for the identification of each pixel lighter than a predetermined threshold value as background and the rest as particles. Before every measurement, calibration is carried out to make sure that the physical unit being used and the pixel size are correctly matched. Figure 15 illustrates how the image is captured and analyzed from 2D silhouettes of particles.

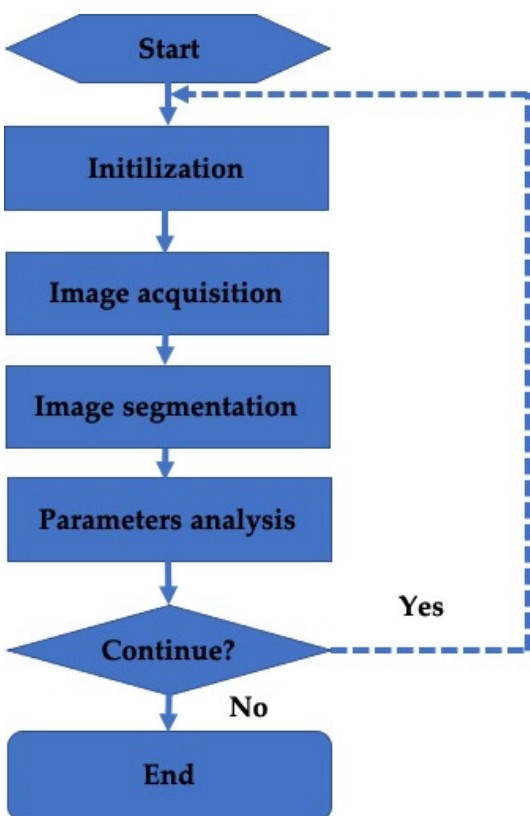

**Figure 14.** The typical algorithm used for image analysis (modified from [140]).

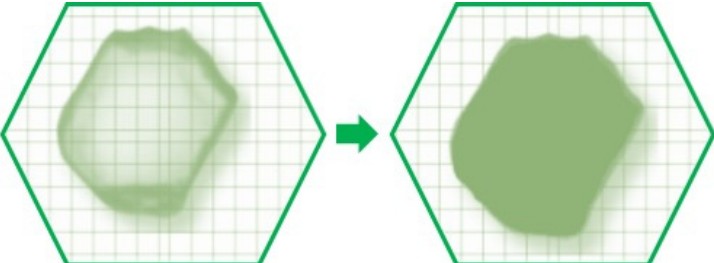

**Figure 15.** Image capture and analysis (modified from [141]).

The threshold value should be adjusted (Figure 16) so that various shape parameters can be calculated from the particle outline obtained [141]. Then, the image is converted to a binary (black and white) image based on the threshold value. Using the pixel intensity values, which typically vary from 0 (black) to 255 (white), and the threshold value, which varies between 0 and 255 (typically about 75% of the background value), all pixels darker than the threshold are considered as particles, and on the other hand, pixels lighter than the threshold are considered as background.

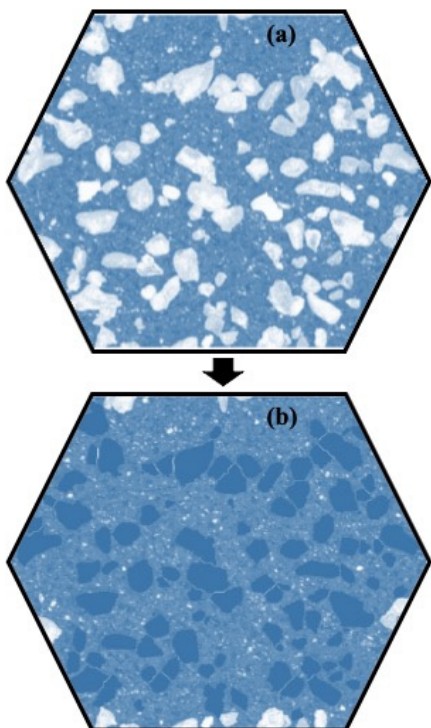

**Figure 16.** Effect of thresholding: (**a**) before thresholding; (**b**) after thresholding (modified from [142]).

The most commonly used technique to describe the particle shape is undoubtedly imaged analysis [139], due to its advantages such as low time consumption, reproducibility, and accurate and reliable results [143]. On the other hand, the main problem with the approach of digital image processing is that only the 2D projection of the particles is recorded and quantified, and the third dimension, namely particle thickness, cannot be acquired directly from the digital image processing findings. Furthermore, digital image processing results are more difficult to interpret as they have to be expressed in area fractions instead of mass fractions, and most researchers are more accustomed to measuring quantities in terms of mass [144]. Therefore, new parameters that are better suited for direct measurement of the particle geometry may need to be included when employing the digital image processing technique [43]. In addition, the resolution in the image

analysis must be taken into account as it affects the perimeter and other measurement parameters [59]. Recently, extensive research on the influence of particle morphology by image analysis has increased, e.g., on asphalt binder-aggregate interfacial interaction [68] and the characterization of morphological characteristics of filler particles [145,146].

Image analysis techniques can be applied in two ways, static and dynamic.

### 3.6.1. Static Image Analysis (SIA)

Static image analysis (SIA) techniques generally utilize an optical microscope for the characterization of particles scattered on a slide (Figure 17a), which is moved by an automated platform. The slide travels to show a different area (such as a grid), a digital camera takes a picture, and a software routine runs different operations to separate contacting particles, distinguish the particles from the background, and determine shape properties. SIA, which complies with ISO 13322-1, can analyze only dry particles that are stationary (2D). Since SIA originated as a microscope technique, just a few are photographed at a time. The prolonged analysis time associated with slide preparation activities is another reason why SIA is seen as a time-consuming technique [38]. Moreover, the fact that it can analyze only a few hundred particles reduces the statistical reliability of the results. Today, SIA systems are widely used in the characterization of active pharmaceutical compounds.

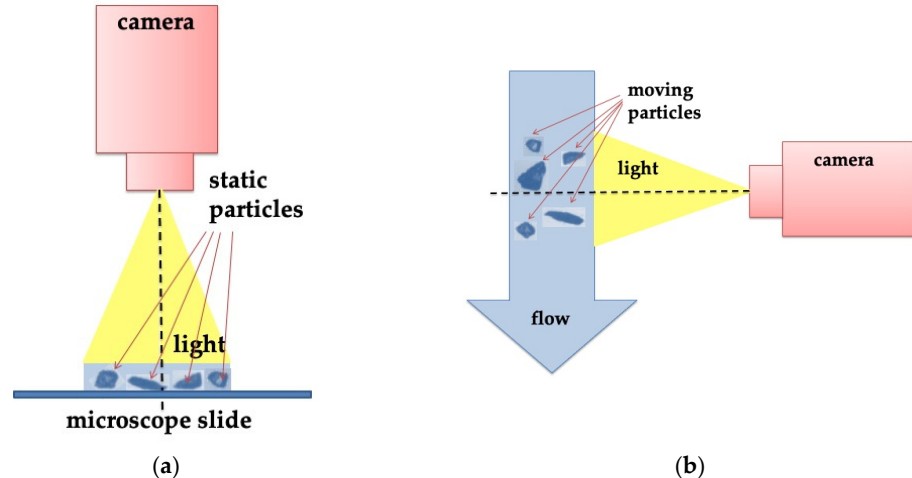

(a)    (b)

**Figure 17.** Schematic illustrations of two different image analyses: (**a**) SIA; (**b**) DIA.

### 3.6.2. Dynamic Image Analysis (DIA)

Dynamic image analysis (DIA), which complies with ISO 13322-2, enables contact-free and non-destructive [147] measurements of particles in motion (Figure 17b) [148]. It operates similarly to a microscope, i.e., a camera captures magnified digital images of the particles, which are then examined by software to calculate the size and shape descriptors of each particle. Therefore, DIA, which has become a popular technique since it has the capability of providing additional comprehensive information on the shape of particles in random orientation (3D) [38], can measure ten thousand images per minute and it is simple to use. For particles ranging in size from 0.5 microns to 30 mm, high-performance automated analysis of dynamic images is enabled, with the capability of capturing up to 500 images per second [102]. Due to increased resolution (more than 1 μm), it has outstanding accuracy and reproducibility over a very wide-measuring range for the quality control of particulate materials [149]. The quickness of the DIA approach allows for real-time monitoring and adjusting of the production parameters. This would imply that production might be streamlined to just match the requirements, thereby lowering costs and improving product homogeneity. Thus, various industries (such as mining–metallurgy (Figure 18), cement, ceramic, and pharmaceutical, etc.,) have already recognized the potential of the DIA method and have started using this technology in research and quality control in many applications [37]. However, it has disadvantages such as a lower

size limit, around 2 μm, and most systems are only capable of wet analysis, which involves suspending particles in a fluid medium to easily adjust the particle flow rate and, as a result, lessen motion blur when taking images [47].

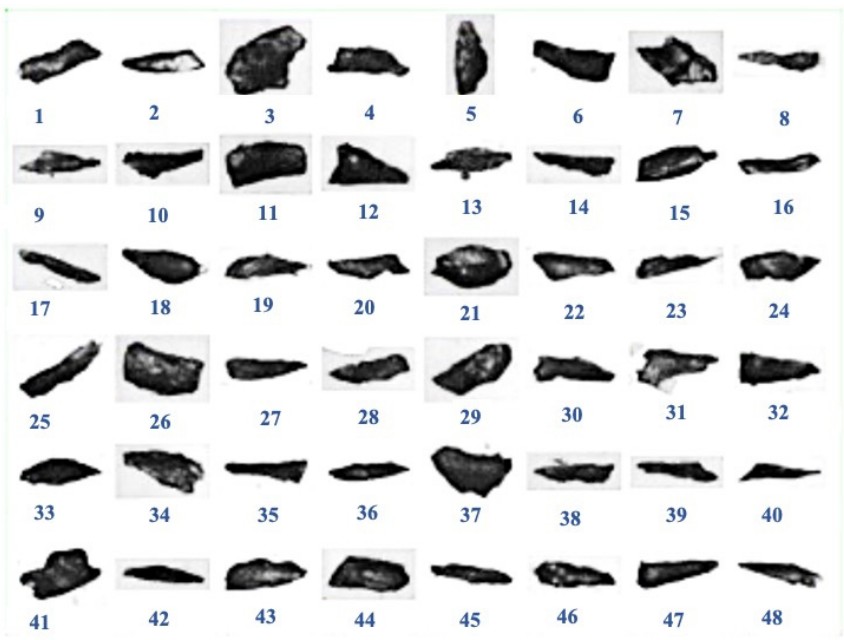

**Figure 18.** DIA images of ball-milled quartz particles.

### 3.6.3. Automated Image Analysis

Systems where images of particles are automatically and objectively captured, measured, analyzed, and classified for inclusion/exclusion, which greatly reduces any operator subjectivity, are called automated image analysis (AIA). In comparison to manual microscopy, automated imaging devices can often measure tens to hundreds of thousands of particles in the same amount of time. They can rapidly analyze particles and give actual precise data. For each measurement, individual photos of every particle are recorded and may be easily viewed with the sample's size and shape details. This offers strong visual evidence of the findings, such as determining the presence of undesirable particles or agglomerates in the sample dispersion. In addition, it offers high-resolution detailed information (particle size, shape, transparency, and chemical identity) by individually measuring each particle in the sample. AIA, which increases the measurement's statistical reliability, has become popular since it is widely used in the quality control of particulate materials with desired attributes for a specific application.

However, in any routine quality control practice, the number of particles that need to be measured is an important issue as it affects analysis time and accuracy [94]. It has been reported that the statistical variation decreases when more particles are subjected to AIA; e.g., when 50,000 particles are compared to 100 particles, it is at least ten times better. Using AIA, the relative spread in results for the mean area parameter is reduced to less than 0.30% as compared to 6% for a typical SEM analysis based on 100 particles [150].

Particle shape and size measurements of various materials such as minerals [150,151], aggregates [43,78,135,152], pharmaceuticals [47], wastewater treatment [148], organic crystalline chemicals [102] and root zone sands [89] were investigated by using AIA.

### 3.6.4. Advanced Techniques with Sophisticated Equipment

Automated SEM (Auto-SEM) technology, which has become a useful tool for process mineralogy and has delivered significant value in numerous operations [153,154], enables data to be produced in digital form when cooperated with an energy-dispersive X-ray energy spectrometer (EDS) and backscattered electrons (BSE). Despite being frequently used

for mineralogical investigation, automated scanning electron microscopy (Auto-SEM-EDS) also offers straightforward, quantitative, and mineral-based shape characterization [41]. Auto-SEM-EDS, where samples are mounted in resin, segmented vertically, and then reassembled into blocks, has been reported to have higher resolution than cameras, and shape information can be easily generated from historical information gathered through routine mineralogical assessments [155].

In addition, comprehensive mineral-based shape characterization can be possible by using a new instrument called a "Quantitative Evaluation of Minerals by Scanning Electron Microscopy" (QEMSCAN®) [156]. The first commercial QEMSCAN® was developed in the 1980s to analyze particles' characteristics rapidly by using Backscattered Electron Intensity (BEI) and X-ray images generated by the SEM. It provides information about particle shape as well as particle properties such as detailed compositions, particle size, and elemental mapping. The benefits of QEMSCAN® include being statically durable, and having accurate mineral detection, round-the-clock operation, high efficiency, use in various applications with high levels of flexibility, various measurement modes, and integrated quality-control programs [157].

Furthermore, imaging/video imaging systems are used to analyze aggregate shape characteristics [158–160]. Although these approaches are typically quick, effective, and offer further advantages of automation that remove the bias connected with the old-fashioned manual procedures, it is challenging to assess the shape qualities in terms of mass or volume because the majority of these techniques only create a 2D image of the aggregates and present only 2D information about the geometry of the aggregate particles [161].

As 3D imaging technology develops, several methods, including stereo photography, photogrammetry, profilometry, X-ray tomography, neutron imaging, laser scanning, white light scanning, and interferometry, are used for characterizing 3D particles. Furthermore, compared to a conventional 2D perimeter, the 3D surface contains a large number of points. This will grow processing loads remarkably. Hence, another challenge in creating a 3D algorithm may be increasing computing efficiency [123].

Nowadays, digital 3D particle geometries can be obtained by various 3D imaging techniques, such as stereo photography, X-ray CT, neutron imaging, 3D laser scanning, white light scanning, and interferometry [67,126,162].

The majority of automated devices analyze cross-sections or projected areas of particles, but, more recently, 3D image data have been obtained by using X-ray computed tomography (XCT) and X-ray micro-computed tomography (micro-CT) [41]. XCT can be utilized to better understand the 2D shapes that DIA and SEM measure, since it is the only technique among many techniques that provides complete 3D particle size and shape information, including interior pores, for each particle within the resolution of the scan [163,164]. Micro-CT, which is the latest image-based technique of 3D imaging, provides the high-resolution examination of the shape characteristics of particles smaller than a few microns [163–166], and provides a non-destructive method for viewing properties inside opaque solid materials to collect digital data on their 3D shape [163,167]. It has been reported that XCT is an ideal technique to provide the full 3D particle morphology for aggregates [163], green iron ore pellets [168], iron ore granulation [169], and additive manufacturing powders [163,164]. However, it has a high initial cost and needs an expert technician to operate and maintain it. In addition, it can only scan a tiny specimen due to resolution and field of view limitations. The usual dimensions of scanned specimens described in the literature were 12 mm in diameter and 24 mm in height. Moreover, processing XCT images requires high-performance computing resources and is computationally laborious and time-consuming [123]. However, it necessitates strict safety and radiation control rules for X-ray equipment. In addition, the resolution and aggregate particle size affect the scanning process' duration. Long scanning times are expected for high resolution and large particle sizes. For instance, it has been reported that the overall scanning time was about 25 min for all the aggregate particles scanned, whilst the average scanning time for a ballast particle was 50 min [161].

Recently, the 3D laser scanning technique gained great interest as a method for assessing aggregate form properties as a more practical and affordable technique compared to imaging and XCT [170,171]. Various 3D laser scanning approaches have been utilized for the assessment of surface area factors of crushed and uncrushed natural aggregates for asphalt mix designs [161,172]. Laser scanning-based digital modeling of gravel particles might be fast, reproducible, and useful for assessing the characteristics of ballast material [173,174]. Nevertheless, these systems can only retrieve particle shapes or the upper half of the particle surfaces visible to the camera [123]. Due to their simplicity of use and low cost, 2D image processing methods are still used for large-scale measurements of particles even though 3D laser scanning and CT scanning are evolving progressively [24].

Atomic force microscopy (AFM) has been the high-resolution non-optical imaging technique for surface analysis of particles at the nanoscale and sub-nanoscale level, since its discovery in 1985. AFM, which can be used in vacuums, air, and liquids, has benefits such as simple sample preparation, accurate results, and 3D imaging property. On the other hand, it is limited to imaging objects with a maximum height of 10–20 μm and a maximum scanning area of 150–150 μm. Another limitation of AFM is its scanning speed compared to SEM. Therefore, the use of AFM for shape characterization is limited because large numbers of particles must be measured to obtain statistically strong results. Therefore, this technique is most suitable for detailed surface characterization such as surface roughness, rather than particle shape.

Moreover, the transmission electron microscope (TEM), which is a microscopy technique that uses a high energy electron beam (rather than optical light), can be used for nanomaterials. TEM has the benefit of being able to magnify specimens up to 10,000 times, enabling researchers to view incredibly, extremely small structures. The TEM, which produces extremely effective high-quality images with exceptional clarity, is simple to learn and operate. However, sample preparation for TEM is very tedious, since it uses thinly sliced samples. In addition, TEMs can only provide 2D analysis for a 3D specimen [175].

The latest study reporting the estimation of shape parameters of sand particles using a deep learning-based network containing a CNN and a regression layer is an indication of how far one can go in this area [176].

## 4. Effect of Particle Shape on Industrial Applications of Various Materials

Engineering applications and industries deal with a variety of particulate materials ranging in size from millimeters to nanometers, since particles come in a variety of shapes based on how they were generated and their mechanical characteristics [177]. Aggregates, metals, coal, ceramics, pigments, toner, soil, minerals, proppants, pulp, paper, synthetic fibers, petrochemicals, glass, plastics, rubber, and drugs and pharmaceuticals are just a few of the examples that involve granular systems, which are key components of a wide variety of industries including mining, mineral processing, powder metallurgy, chemistry, petrochemistry, energy, pharmaceutics, nanotechnology, and advanced materials [9]. It is generally known that, in particle processing, the behavior of particulate material relies not only on equipment characteristics but also on their physical properties [178].

Since organic/mineral particles are used for many engineering applications, their physical properties, such as the size and shape of the particles, affect their usability in various industries. Thus, particle shape properties find a wide range of industrial applications from aggregates to nanomaterials. Understanding the shape of granular materials and their response to various properties and functions is important in many engineering applications, such as advanced manufacturing, including additive manufacturing and powder metallurgy, battery and energy storage, building materials such as cement and asphalt, mining and minerals, metallurgy, nanomaterials, oil and petrochemicals, polymers, plastics, fibers, agriculture sciences, medical sciences, chemistry, and applied chemistry, and pharmaceutical sciences [44]. Therefore, in this review, particles and their shape effects on various engineering fields from macro to nanoscale, depending on desired properties in specific applications, were given as sub-categories.

### 4.1. Aggregate Particles

The strength of the aggregates and the longevity of materials such as concrete, asphalt, and railroad aggregate are influenced by particle shape [152].

#### 4.1.1. Ballast Aggregate Particles in Railway Roads

Railroad ballast is a mixture of uniformly graded coarse aggregate materials poured between and immediately below the ties to provide drainage and structural support for the loading exerted by trains [179]. A strange ballast problem occurs due to the number of times the overload occurs under the train. Asymmetric rail settling and ground irregularity are triggered by the fracturing of sharp ballast corners, continual grinding and wear, as well as the crushing of weaker particles under strong cyclic loadings. It is widely known that the performance of asphalt and concrete pavements, as well as the structures of railroad tracks, is influenced by aggregate and ballast shape, which must comply with the specifications at the same time. Since aggregate/ballast particles have irregular and non-ideal shapes and variable surface textures, aggregate selection should be improved for construction purposes on daily bases for quality control and quality assurance [161]. The desired property for ballast aggregate according to the American Railway Engineering and Maintenance of Way Association (AREMA) [180] is an open grade of hard, angular-shaped grains that yields a minimum of flat and elongated fragments, sharp-edged cubic pieces. In other words, the best performance in the track structure is provided by uniformly graded angular, crushed hard rocks with no dust aggregates. Since highly angular crushed aggregates are also preferred due to the excellent interlock they provide [179], angularity is a not only desirable property for the bond between the bituminous binder and aggregates in asphalt and spray seals, as well as the bond between cement and aggregate in concrete, but also for the ability to interlock to avoid shearing and deformation in unbound materials [161]. It has also been reported that asphalt mix stiffness and volumetric qualities are both impacted by flaky particles [181] and are undesirable in constructing railway tracks, as flaky particles rapidly break when subjected to repeated train loads. This is due to their tendency to lie flat, thus causing the planes to slip, lowering the interlocking and durability of pavements [161].

#### 4.1.2. Aggregates Particles in Concrete and Asphalt

The foundation of our modern life is made up primarily of cement and concrete, which are the most often used construction materials due to their plentiful resources, simplicity of usage, durability, and adaptability [182,183]. Concrete is typically known as a three-phase composite containing additions in the type of aggregate particles and air voids contained in the cementitious material (matrix). As the volume of aggregates constitutes approximately 80% of the total concrete volume, their properties have a big impact on how well fresh and hardened concrete performs, as well as how much concrete costs [184]. It is reported that aggregates significantly affect the concrete mixture and hardened concrete characteristics, and they will have different compressive strengths, modulus of elasticity, and shrinkage properties relying on the proportion of coarse aggregates, sand, and paste. Numerous research has been conducted to assess the characteristics of spherical and non-spherical particles and how they influence the performance of finished aggregates, especially concrete and asphalt blends, since PSD, particle size, particle shape, surface texture, strength, absorption, resistance to freezing and thawing, specific gravity and bulk density are the main characteristics of aggregate particles used in concrete [185–191]. It has been reported that the shape of the aggregate grains significantly affects certain properties of concrete, both fresh and hardened [192], because it is well-known that particle shape and size are very important parameters in managing ensemble-level mechanical behavior including strength, hardness, and packing fraction friction angle. In addition, it has been revealed that the shape of the aggregate particles used in concrete is a criterion that is taken into consideration in deciding many important concrete properties, such as the rheology of fresh concrete and early-age mechanical properties [193,194]. In essence, it is a well-known fact that void content is higher in flaky, elongated, angular, and poorly graded particles

compared to cubical, rounded, and well-graded particles [195], and the stiffness, fatigue behavior, tensile strength, workability, and durability of concrete have all been linked to the shape of the aggregate particles [196]. Thus, it has been reported that not only the finishability, bleeding, pumpability, and segregation of fresh concrete [193,194,197,198], but also the strength, stiffness, shrinkage, creep, density, permeability, and durability of hardened concrete are impacted by shape, texture, and grading of aggregates. There have been some studies showing a strong correlation between some shape indices of aggregate and the compressive strength of concrete. While longer and flaky particles tend to make the concrete mix less workable, which could reduce durability over time, spherical particles were found to be more suitable for high compressive strength, ultrasonic pulse velocity, unit weight, and slump values of concrete [196]. Similarly, it has also been reported that the workability and strength of concrete can be impacted by the qualities and changes of aggregate particle shape in terms of sphericity and roundness [199].

Particle shape also influences concrete [200] and the production of dust [201], since wear resistance of asphalt concrete arises from heavy traffic and environmental loading depends on the durability of aggregate forming asphalt [75]. The durability of the asphalt mixes in the asphalt layer will be decreased by aggregate particles that fracture during production and construction, resulting in structural damage, flip, and holes [158,161]. As good management of the physical properties of concrete and asphalt products will be possible by quantifying the fine material properties, and this is crucial, the effect of particle shape of mineral filler on the asphalt binder-filler interfacial interaction was investigated. It has been found that granite filler, which has better shape properties than limestone and basalt fillers, has relatively apparent angularity and surface texture features [202].

## 4.2. Proppant Particles in Hydraulic Fracturing for Oil and Gas

Hydraulic fracturing, which is known as the process of extracting hydrocarbons from massive unconventional geo-resources, requires the high-pressure injection of fracturing fluid into the wellbore to create and reinforce fractures in the rock formations [203]. In other words, the well is stimulated by fracturing (Figure 19) to produce more oil and natural gas, which are the foundations of the global energy system. It has been reported that fracturing techniques, in comparison to other conventional methods such as vertical drilling of the wells, can often boost a well's overall production rate from 5 to 15% and increase the yield of hydrocarbons by 1.5 to 30 times. It is well known that settling rate increases will hinder hydraulic fracturing by decreasing well production, requiring the use of chemical additives. Thus, proppants, which are substances used to form fracturing fluid, keeping the cracks open, are essential to release the desired hydrocarbon materials, because the fracture closes quickly after the fracturing energy is reduced [204,205]. When the pressurized fluid injection is discontinued after hydraulic stimulation, the proppants are allowed to keep the crack open. The physical characteristics of both natural and artificial proppants, in addition to availability and affordability, are of the utmost importance because it is well known that the porosity and permeability of the proppant pack affects the fracture conductivity, which means an indicator of ultimate production.

Since silica sand proppants were first employed in hydraulic fracturing in 1947 [206], a wide variety of materials such as ceramics, resin-coated sand, sintered bauxite, kaolin, and natural sand (Figure 20) have been reported as proppants [206–208]. It is reported that approximately 300,000 and 4 million pounds of proppant are used in gas–oil and shale gas wells, respectively [209,210].

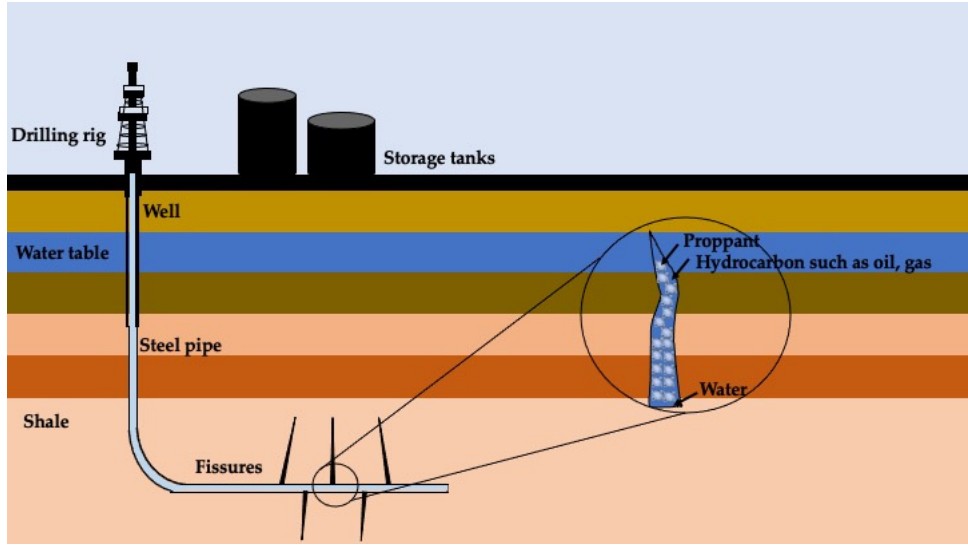

**Figure 19.** Hydraulic fracturing, modified from [209,211].

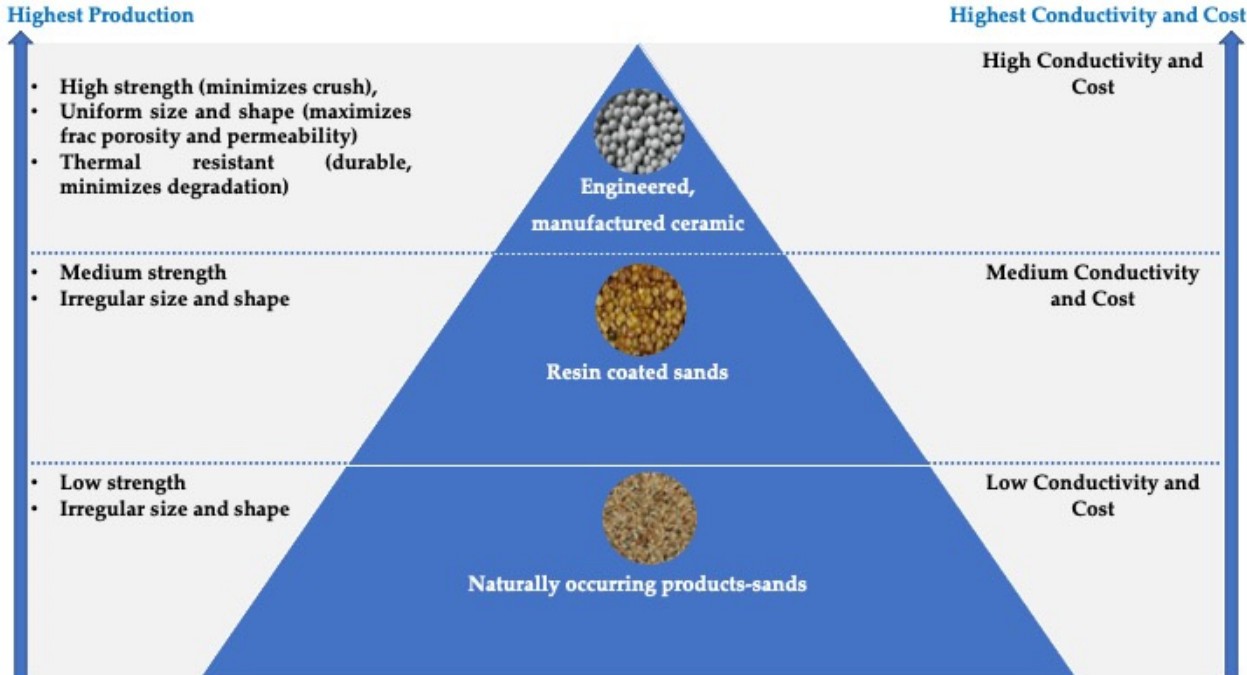

**Figure 20.** Proppants and their features (modified from [212]).

In practice, it is reported that the structural properties of proppants are evaluated according to some standards, such as conductivity, acid solubility, shape, crush resistance, solubility, and turbidity. Proppants, which vary in terms of size, shape, and strength, are delivered to the fractures by fracturing fluid and serve a crucial role in sustaining conductive pathways [213–215]. Therefore, the proppant must possess specific properties (concentration size, density, and shape of the proppant, the viscosity of the carrier fluid) to be able to endure the high pressures present inside the fracture. As a result, these qualities influence how well proppants function generally in fractures [212].

The shape of the proppant is thought to be just as important as its size, as it will change the ultimate permeability through the fracture. A wide range of particle sizes and shapes will reduce conductivity and permeability, resulting in a compact packaging configuration. It is thought that the preference for more uniform size and spherical-shaped proppants

can increase conductivity. In other words, it has been reported that the optimum use of proppant is crucial from an economic standpoint, as it decreases settling, which is one of the main problems with hydraulic fracturing, and then increases the well's conductivity, thereby improving the process' effectiveness. Studies have shown that there are correlations between settling rates and proppant shape, and under the same conditions, particles with higher sphericity generally settle more quickly than particles with more irregular shapes. As a result, it has been reported that the use of irregular particles during fracturing may decrease settling. However, it has also been noted that this may result in decreased porosity and more clogging. Therefore, it is also thought that a balance is necessary between ensuring proppant packing and lowering settling [216]. It is well known that choosing the right proppants to provide structural support and a permeable flow path for the reservoir fluid when designing hydraulic fracturing is crucial. Thus, a wide variety of proppants such as resin coating, high and ultra-high-strength ceramics, rod-shaped proppants, and self-suspended proppants have been developed [217].

In recent studies, it has been suggested that it is possible to improve the conductivity of hydraulic fracturing to a level that cannot now be achievable with spherical proppants, by completely altering the conventional shape of proppants employed in hydraulic fracturing. That is, studies have revealed that the packing of non-spherical particles differs significantly from that of sphere-shaped particles [218,219]. In other studies, it has been reported that when the shape of the proppant changes from spherical to rod-shaped, the porosity changes in the range of 24–45%, and the conductivity of the fractures increases [220–222]. Thus, a new proppant, which has an elongated shape and high strength, has been developed to improve the ultimate fracture conductivity, as the biggest issue when changing the particle shape is the effect on the packing behavior and failure modes of the packed bed. It has been reported that these newly developed cylindrical proppants outperform spherical proppants in terms of conductivity and flow-back portion [223]. In another study, investigating the effect of aspect ratio on packing porosities by simulations using cylindrical particles with different aspect ratios (0.5, 1, 2, 3 and 4) at the closing stress pressures of 15 MPa, 25 MPa and 40 MPa, it has been reported that cylinders with aspect ratios of 3 and 4 in comparison to spheres exhibited larger packing porosities [224]. Other research has also reported that particle shape affects the mechanical response of the proppant pack in fractures regarding unconfined pack bed height, packing porosity, restricted modulus, and coordination number [225]. In another recent study, it was found that rod-shaped proppants can yield cracks up to three times the conductivity of spherical proppants, due to the increased porosity and permeability of the proppant pack as the aspect ratio increases [226].

### 4.3. Glass Bead Particles in Highway Road Paints

Road markers, which consist of a base (paint) layer and a retroreflective layer [227], are known to be one of the most fundamental but powerful safety elements on most asphalt roadways. The base layer of these distinctive industrial maintenance coatings can be made from a variety of materials, whereas the retroreflective layer is always made of partially embedded glass beads [228]. While the light is randomly reflected in all directions when paint lines without beads are used, the reflection of light back to the light source when round reflective beads are added is called retroreflectivity in the industry. Glass spheres added can be seen by drivers at night by providing retroreflectivity [229], and are also known to protect the base layer against abrasion [230]. In other words, retroreflectivity, which is measured by the coefficient of retroreflected luminance and expressed in millicandela per square meter per lux (mcd/m$^2$/lx) [231], is the primary factor that determines the quality and service life of road markers. Retroreflectivity is essential for road safety because, although there is much less traffic, accidents happen more frequently and with greater severity at night than during the day [232,233].

Since the chemical and physical characteristics of the beads, such as their size, refractive index, clarity, and roundness, have a significant impact on how well they reflect light,

one of the manufacturing variables that can be controlled for retroreflectivity and durability is the roundness of the beads. As seen from Figure 21, the size and shape of glass beads used for highway paint play a critical role in reflection properties. Beads must be rounded to be retroreflective. Light can only be reflected back toward the light source by spherical beads. In other words, more round particles give more reflectivity back to the source. It is suggested that the particle shape of the glass beads also determines their strength since round glass beads were found to be less likely to break than angular glass beads [234].

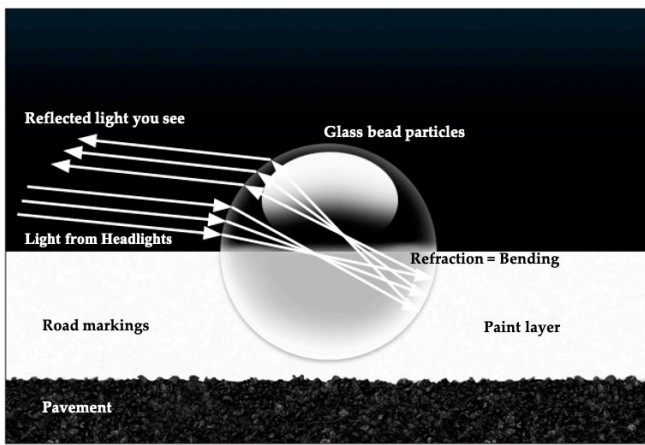

**Figure 21.** Retroreflective mechanism of glass beads used in highway road paints (modified from [235]).

### 4.4. Viscosity and Rheological Properties of Particles in Suspensions

It has been reported that suspensions of particles in a viscous liquid can have a major impact on the rheology, drastically resulting in considerable non-Newtonian behavior, and also in dilute suspension [236]. It has been reported that the viscosity of the suspension depends on the particle shape, so the particle movements are affected by its rotational and translational abilities. It has been found that uniform spherical particles can rotate more freely around their own axis, while platy or rod-like particles require an area equal to the length of their longest axis. Since plate-like or rod-like particles do not have this sufficient spacing, it is thought that the interaction resulting from the contacts between the particles will enhance the viscosity. In other words, it has been reported that the intrinsic viscosity of the solution increases quickly as the shape of the particles changes from spherical to elongated, as shown in Figure 22 [237–239].

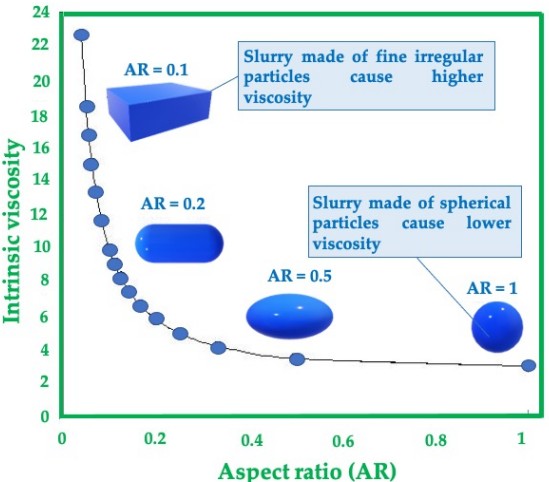

**Figure 22.** Variation of the aspect ratio of particles by intrinsic viscosity of the solution (modified from [240]).

In many industries, the flow characteristics and rheology of colloidal dispersions are of great importance. Numerous industrial applications make use of non-spherical colloidal dispersions to take advantage of their distinct elastoplastic rheological behavior [241]. Moreover, it has been reported that there are numerous instances [242,243] of industrial fluids in which rheology has been altered by combining colloids of various sizes, shapes, and charges [244]. For example, anisometric colloids, particularly natural clays, are extensively employed in a variety of industries, from oil well drilling to personal care goods [243]. These make use of one or more of the various clay particle-specific capabilities. It has been reported that they can produce suspensions with comparatively high viscosities at low volume fractions (and low shear rates), because rotating anisotropic particles such as platelets and needles are very effective at filling up empty space. This considerable yield stress or static elastic (storage) modulus is typically present in conjunction with this high-low-stress viscosity [244].

According to recent studies, it has been reported that the viscosity properties can be increased by modifying the morphology, e.g., when the morphology changes from blocky round particles to platy particles, it has been found that the viscosity of aluminum trihydroxide (gibbsite) is improved [245]. Similarly, it has been reported that the addition of colloidal particles of various sizes and shapes can greatly improve the rheological characteristics of these fluids, in a study using a model mineral-colloid systems whose shape varies gradually from a plate-like aluminasol (gibbsite) through a lath-like smectite clay (hectorite) to a rod-like aluminasol (boehmite) [244]. In a recent study, the rheological behavior of mixed anisometric colloid systems based on pure components (such as mixtures of lath + rod, lath + plate, and lath + sphere using cationic alumina-coated silica spheres) have been investigated. It has been found that the magnitude of the effect of dispersion rheological properties depends on the shape of the grains, shear modulus, low-stress viscosities, and effective yield stresses. Moreover, it has been reported that the magnitude of these properties increases in the order of additives rods < platelets < spheres [241].

### 4.5. Settling Properties of Particles in Gravity Sedimentation

The separation of particles from fluids while under the influence of gravity is known as gravity sedimentation. Particle aggregation, known as a complicated process formed depending on several factors including physical, chemical, and hydrodynamic, has been of interest for research, analysis, and development in some systems such as water treatment [246]. Since materials with a particle size of a few microns settle very slowly in gravity precipitation, it is desirable to agglomerate, or agglomerate into relatively large clusters called floc, which settle more quickly [247]. It has been reported that particle size, particle shape, surface characteristics, density, the proportion of solids in suspension or pulp, viscosity, and density of the liquid all play a role in how quickly solid particles settle out of the water [248]. Thus, a thorough understanding of how a particle's shape affects its settling behavior is helpful for the prediction and design of separation processes [249].

In a study investigating the relationships between transport and growth behavior, it has been suggested that improvements can be made to the description of aggregate properties and behavior in aqueous solutions using a fractal-based analysis, rather than relying on the common approach of assuming aggregates to be impermeable spheres. Since the solid sphere assumption will have some disadvantages in the aggregation theory in case of high porosity, it has been reported that basic geometrical properties of aggregates in suspension can be studied by generating features such as fractal dimension using an image-based analysis. Therefore, it has been shown that not only fundamental properties such as density and porosity but also fractal dimensions are size dependent. The drag coefficient postulated in the equation, which evaluates the aggregate settling rate from a standard force balance between gravity, buoyancy, and drag, arises from the formula for a spherical particle in laminar flow. However, it has been reported that it could be more accurate to presume the drag coefficient depends on fractal dimensions, which define overall aggregate shape including the effect of various shapes on settling properties [246].

In another study, the influence of sphericity and the capacity to forecast the drag coefficient and settling velocity of a non-spherical grain has been investigated by using ellipsoidal pebbles settling in glycerine with a viscosity 1000 times greater than water [250]. Similarly, it has been reported that a grain's settling velocity in a fluid decreases when it deviates from a spherical shape. It has been reported that, from two particles of the same particle weight, the plate-like or needle-like one precipitates more slowly than the spherical one, but the density and viscosity of the medium must also be taken into account [247]. It was also stated that the deviation in the settling velocity of a spherical particle of the same weight increases with the degree of non-sphericity [250]. When the impact of particle shape on the settlement tendency of sand particles under the gravity effect in an initial stagnant water environment has been investigated, it has been found that the terminal settling velocity tends to decrease with an increase in the aspect ratio [251]. In other words, the settling rate of round particles is the greatest of all shapes measured, whereas an ellipse has the least settling rate, as can be seen in Figure 23.

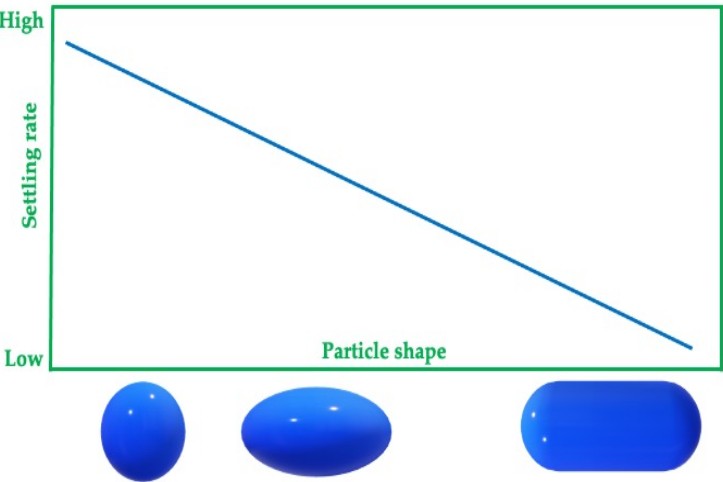

**Figure 23.** Settling rate versus particle shape (modified from [250,251]).

### 4.6. Mechanical Properties of Bulk Solids in Conveying

The primary properties of the individual particles that make up a bulk solid dictate a number of their physical-technical properties such as density, flowability, bulk density, tap density, compressibility, angle of repose, cohesion, dispersibility, permeability, compressibility, green strength and how the solid will behave when it is handled and conveyed. For a specific powder composition, these properties may be influenced by the granulometric composition, particle shape, particle morphology, particular surface, moisture content, etc. [18]).

### 4.6.1. Density Properties Related to the Packing of Particles

Since particle packing, as well as effective confining stress, control how mechanically a soil responds to changes in applied load, the configuration of consolidation equipment and the extent of the press motions necessary for packing and compacting the loose powders are dictated by the term called apparent density of powders. It decides the real volume allotted by the weight of loose powder and is formulated as the weight of a unit volume of loose powder. It has been found that apparent density, which depends on PSD, reduces as the particle shape deviates from spherical [18]. In other words, particle shape affects the apparent density of the powder feed stack, which directly influences the ultimate density of the formed product [252]. Tapped density, which is another variable that can be used to predict how powders will pack and flow [16], is defined as the weight per volume of powder when the volume of the powder inside the container stops decreasing [253]. It has been reported that, since PSD, particle shape, and surface roughness all influence tap

density, particle shape significantly affects how granular materials behave mechanically, e.g., the packing density [40,254–256]. It has also been reported that it is always greater than the open flow apparent density, and the level of increase from apparent to tap density highly relies on the size and shape of particles [18]. Thus, smaller densities, larger porosities, and higher void ratios are tendencies that are brought about by irregular particle shapes. Another density term related to the packing of particles is the bulk density of a material, which refers to the total weight of a material for a certain volume contained, and covers the fluid-filled (such as air or water) interparticle spaces. It depends on the method of packing, and a system of packed particles can become denser with time and the settling process. The limiting value is known as the tapped or packed bulk density [9].

It is known that the coordination number and the relative packing density (packing fraction) are the two primary metrics used to quantify particle packing. Packing density, and thus compaction properties, has been reported to decrease when the shapes of the grains change from small flake ones to larger spherical aggregates as a result of grinding time or the presence of surfactants [257–259]. According to a correlation analysis investigating the influence of shape descriptors on packaging behavior [63], it has been found significant because the particle shape affects physical and behavioral properties such as packaging and fluid interaction [260]. It has been reported that packing densities increase as the shape of the particles approaches spherical, gradually decrease as the aspect ratio values decrease, and even lower packing densities are achieved with spongy particles. Moreover, under vibration, irregular particles are thought to have a high packing density due to their predictable orientation [48,202].

### 4.6.2. Properties of Flowability and Angle of Repose

Flowability is another important property for designing and operating particulate systems, such as pneumatic conveying and multiphase reactors. The relative motion of a collection of particles among nearby particles or along the surface of a container's wall is known as powder flow [261]. The efficiency of powder conveying, blending, and packaging rely on the flow properties of powders, which are crucial in many handling and storage conditions found in bulk material processes in the agricultural, ceramic, food, mineral, mining, and pharmaceutical industries [262].

It has been reported that the initiation of powder flow is due to the incipient failure where the forces acting on the solid produce stresses in excess of the strength of the solid. It has been also reported that the rupture of powders depends on the mechanical properties of bulk solids, including the angle of internal friction, the kinematic angle of internal friction, the kinematic angle of wall friction, bulk density, unconfined yield strength, major principal stress, and cohesion.

Flowability, which is defined as the ratio of the difference between tapped density and bulk density to tapped density, improves with decreasing compressibility [253]. It has been reported that the geometry of the system, particle shape, PSD, and particle surface qualities all have an impact on a powder's capacity to flow [14]. It has been shown that compressibility mainly depends on particle size, whereas flowability is influenced by particle shape. In other words, round-shaped particles compared to irregular particles showed greater performance regarding flowability and lower porosity [252]. As seen in Figure 24, parameters such as particle size and shape have a direct impact on bulk density and flowability, though not entirely. In addition, it is known that smooth or regular particles typically have better flowability as compared to rough or irregular particles [263], as the flowability is decreased, since rougher surfaces cause intergranular friction and irregular particles are more likely to engage mechanical interlocking. In addition to packing spherical particles more effectively than irregular ones, greater bulk densities have also been reported to be produced [264].

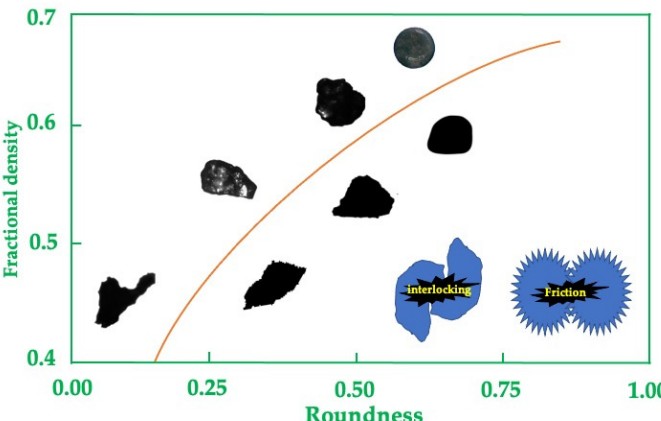

**Figure 24.** Bulk density as a function of particle roundness, (modified from [265].

Generally, a coarse powder with a lower PSD and aspect ratio is known to flow better. On the other hand, it is suggested that high aspect ratios and irregular particle shapes can block the free movement of particles. Compared to spherical particles, needle-shaped crystals have been reported to flow less effectively. In another study, it has been reported that larger, rounder or blockier particles tend to flow more easily than smaller, irregularly shaped ones, such as glass beads and corn flour [16]. It has been reported that flow issues caused by the consolidation properties of the powder can be resolved by altering the powder density.

The fluidization properties and flow regimes can also be used to characterize the flowability of dry particles. When discharged out of a transport system, such as a hopper, other crucial flow qualities, such as the angle of repose, angle of fall, compressibility, and dispersibility, provide information for the flowability of granular materials [9].

When a powder cone is generated by freely dropping powder on a plane under specified circumstances, the angle of repose can be defined as the angle between the free surface of the cone and its horizontal plane [16,266].

It is known that the degree of the angle of repose is a measure of the flowability of powder, i.e., the greater the flowability, the smaller the angle of repose [253]. The angle of repose and the dilatancy behavior of granular materials are reported to be influenced by particle size, PSD, and shape [18,267].

It has been suggested that powder internal forces are governed by particle shape, which also affects the angle of repose. For instance, it has been reported that there is a clear distinction between particles with spherical and angular shapes, such as glass particles, or crystalline and spangle-shaped particles, such as those found in sugar. Glass spheres with a low coefficient of friction roll on a conic heap that has already been produced, remaining very flat. On the other hand, angular particles pile up to widen the prepared heap, which enhances the angle of repose [14]. It has been reported that, as particle shape deviates from sphericity, flowability declines. In other words, asymmetrical forms are less likely to generate friction and flow issues than irregular, dendritic, and flaky particles. On the other hand, it has been reported that inter-particle sliding is limited by spongy particles at a greater angle of repose [253,268].

### 4.6.3. Pharmaceutical Powders in the Compaction Process

The most popular pharmaceutical dosage forms, tablets, and capsules are made by compacting powders and granules into solid masses of a specific dosage unit. Making tablets requires a powder to compact when pressure is applied, hence this property is crucial [269,270]. While the term compaction is used to define a bulk solid's capacity to form coherent compacts, it is also used to measure how process variables such as maximum compression pressure affect how strong the resulting compacts are mechanically [16]. The compaction of a powder by tapping or mechanical compression is referred to as com-

pressibility. These processes may happen accidentally during handling or transportation, or deliberately when making tablets or agglomerates [8]. It has been reported that compressibility provides important information for identifying densification and compaction processes, especially for exploring the effect of bulk density of particulate materials on the applied pressure [16]. It has been experimentally found that grain properties, including roundness, sphericity, and smoothness, impact the potential compressibility of materials [44]. In addition, the effect of particle shape on their behaviors during tapping has also been studied [271–273]. For example, the effect of particle shape on a void fraction in a tapping particle bed have been investigated by using certain materials having various particle shapes, sizes, and densities [274]. In another study, the random packing of non-spherical particles has been simulated to examine how particle shape and aspect ratio affect packing density and microstructure [254]. Another study focused on the influence of particle shape on packing density as well as on the strain mechanical characteristics of sandy soils [40]. It has been reported that when particle irregularity increases, stiffness decreases but stress sensitivity is increased, leading to an increase in compressibility under zero-lateral strain loading. It has also been found that irregular particles with a large surface area and sharp edges are more reactive than coaxial particles with a smooth surface. In other words, it has been reported that the highest reactivity (i.e., chemical reactivity, oxidation, adsorption, explosiveness, catalysis efficiency) was observed in flaky grains, and decreased in the order of granular powder, spherical atomized metal powder, and spherical atomized metal powder, respectively [275–277]. On the other hand, it has been reported that spherical particles can be compressed more than irregular ones because they can be reorganized more readily. It has also been stated that the internal porosity decreases compressibility because of internal pores in spongy particles' resistance to filling up [18,48,258,259,269,274,277].

### 4.6.4. Discharge Rates of Particles in Storage Systems

Since silos, hoppers, and standpipes are widely used for storing and delivering granular materials in the food, pharmaceutical, chemical, agricultural, manufacturing, powder, and fuel industries at a controlled rate [9], it is necessary to know how efficiently the bulk material can flow in many industrial applications, such as placing it in a container or discharging it from a silo or a hopper [16], as well as to predict accurately granular material's discharge rate from these storage systems. It is well known that the rate at which a granular material is discharged from a hopper can vary depending on some factors, including particle shape. Therefore, particle shape is considered one of the key parameters affecting the discharge rate of granular material from a hopper [278]. Since it is well known that mineral powder packaging properties in bins and hoppers are strongly influenced by particle shape, it is considered crucial to be aware of the size reduction parameters that will give rise to various intended shapes to fulfill the needs of various specific application areas [22].

Despite conflicting results, many studies have suggested that the particle shape and angularity significantly affect the flow behavior of bulk solids. For instance, Sukumaran and Ashmawy [267] observed that the flow rate reduces as the particle shape factor and angularity increase in a study of the flow behavior of a variety of sands and spherical glass beads in the size range of 0.30 to 0.50 mm. When the discharge of 0.027 and 0.110 cm spherical glass beads, 0.055 cm rounded yellow sand, and sharp grey sand fractions in the size range of 0.020 to 0.053 cm from cylindrical flat-bottomed hoppers was investigated, they found that angular materials had lower discharge rates than spherical ones [279]. Similarly, it has been estimated that discharge rates for elongated or angular particles would be roughly 30% smaller than those for spherical ones based on simulations using the Discrete Element Method (DEM) [280].

On the other hand, it has been found that 6.5 mm hexahedron corn particles were discharged from a rectangular hopper at a greater rate than 5.7 mm spherical soybean particles. In addition, it has been reported that flow rates for disc-shaped cocoa beans with a diameter of 19.22 mm were 20–30% greater than those for spherical aniseed particles with

a diameter of 14.74 mm [281]. It has been also suggested that disc-shaped particles had 40% higher discharge rates than circular ones by DEM simulations [282]. Nevertheless, it has also been found that, for non-spherical materials with an aspect ratio close to unity and/or with high roundness, the predicted discharge rates deviated from the measured values by only about 10%. However, it has also been reported that, as the aspect ratio increases and/or the roundness decreases, the discrepancy between the measured and predicted discharge rates increases [267].

### 4.7. Mechanical Properties of Soil Particles

Sands, which are crucial components of many technical products such as concrete blocks, composite materials, glass, and ceramic production, are employed in a variety of sustainable engineering disciplines such as material engineering, mechanical engineering, hydraulics, and geotechnical engineering [28]. Sands are utilized in agriculture because of their applicability as soil with strong permeability properties to make heavy clay soil lighter, while in earth engineering, chemical engineering, and petroleum engineering they can also be used as a filtering substance. They are also frequently employed in the construction of temporary and (seldom) permanent roads [283].

It is known that the behavior of sand and its sustainable use in various engineering disciplines are significantly influenced by the shape and PSD properties of sand particles. For instance, permeability, characterized by the porous media structure such as porosity, pore size, shape, and curvature, is thought to be a key factor used in the design of models for fluid flow and mass transfer in porous media. It has been reported that even in the basic instance of binary packing with grains of different particle sizes, the impact of porosity and related pore crimps on bed permeability should be taken into account [284]. The relationships between several particle shapes and fractal dimension factors, as well as data about the relative density and permeability of soils, were investigated by several researchers [285,286].

It is known that particle shape also influences mechanical parameters such as strength, stability, and compaction level in soil geomechanics [40,50,287,288]. Furthermore, it is known that some properties of granular materials, such as modulus of deformation [289,290], shear strength [291–294], and critical state, [293,295] are closely related to grain morphology [40]. When the impacts of particle shape on the different strain mechanical characteristics of sandy soils were investigated, it has been reported that fine powders with a high shape factor showed a rapid increase in tensile strength with a slight increase in packing percentage because of strong frictional forces. It has been reported that, compared to spherical powder compacts, oblate and flaky compacted powders have a smaller deformation ability [40,48,277,296–298].

It is well known that sand's particle size, shape, and packing have an impact on its stress–strain and stress path properties. It has also been reported that the particles' size and morphology significantly impact the sand's undrained shear strength [299]. In other words, the size and shape of the sand grains are among the main factors affecting the performance of the soil when used for various purposes [286]. It is suggested that irregular particle shapes lead to lower densities and larger pore and void ratios [28]. It has also been reported that minimum and maximum void ratios are influenced by particle shape [49]. The general view in published studies on the effect of the uniformity coefficient and packing density of sand samples is that as the sphericity and roundness of the grains increase, the minimum and maximum void ratios also decrease [28]. It has been observed that, when initially poured from a funnel to establish the maximum void ratios condition, angular particles with low roundness are not easily packed since they do not rotate and slip like spherical particles. It has also been stated that, as the particle shape deviates from the spherical, it provides a broad range in the fabric between minimum and maximum porosity. Furthermore, it has been reported that, while diffusion, stiffness, and residual friction angle are reduced by platy particle shape, roughness and angularity control the evolution of stress-induced anisotropy, reduce small-strain stiffness, and enhance high-strain strength.

It has been concluded that reduced densities, greater porosities, and higher void ratios are tendencies that are brought about by irregular particle shapes [44].

### 4.8. Metal Additive Particles

Metal additive manufacturing technology, which has been embraced by the aerospace, energy, automotive, medical, and tooling industries, has effectively evolved in recent years from a prototyping tool to a still-emerging but well-established and financially feasible option for component production [300]. The creation of ceramics utilizing additive manufacturing methods offers the possibility of producing near-net products, which is not possible with traditional production methods. Thus, it is impossible to ignore the importance of powders in the additive manufacturing of ceramics [301]. In other words, it is known that the performance of additive manufacturing is influenced by the physical properties of a metal powder, such as packing density and flowability. It has been reported that powders with good flowability can spread uniformly and smoothly across a bed to form a homogeneous layer with no air spaces, while powders with high density and consistent packing are linked to the creation of components of consistent quality and fewer faults [265]. It has also been reported that powder bed formation, melt pools, and microscopic homogeneity in additive manufacturing depend on powder size, shape, and chemistry [302], and the quality of the components manufactured by additive manufacturing is clearly influenced by the particle shape and PSD of metal powders [164]. For instance, spherical and smooth grains have been found to have better flowability and higher packing densities in additive manufacturing than irregular and rough grains. This caused more uniform powder flow through the dosing mechanism, more optimized and uniform layer density, and a consistent and predictable quality achievement in the obtained final built parts [303]. It has been reported that the smoother, spherical particles can flow more easily and thus produce higher apparent densities, due to a bulk material property associated with the formation of high-density layers [304]. However, additive manufacturing powders are generally not perfectly spherical, as they are generally produced by gas or plasma atomization of molten metal, where some particles are welded together and form elongated shapes [164]. Furthermore, it has been reported that spherical particles have the best conditions for melting and spraying and increase feedstock flowability; on the other hand, irregular and spongy powders have greater drag coefficients and thus higher deposition performance [305–308].

### 4.9. Bulk Powder Properties of Particles in Powder Metallurgy

Powder metallurgy, a cutting-edge technique for producing structural and functional materials, is known to have the advantage of better utilization of raw resources, reducing manufacturing costs since almost all common metallic materials can be successfully used to create components using this technology. In addition, powder metallurgy is known to provide significant material and energy savings by enabling the production of parts with extremely complicated shapes with no need for final machining. Above all, it is also argued that the quality of materials created by powder metallurgy often outperforms those of adequate materials created using other processes. Products from powder metallurgy are mostly employed in the oil and gas, mechanical engineering, aircraft, and automotive industries [309].

Sometimes, during mechanical alloying, powders are ground with a wet media; this is known as wet milling. Wet milling, as opposed to dry milling, can create smaller particles with a larger powder volume while avoiding the aggregating effect of the powders. Due to its simplicity and applicability, it is a widely used and effective process for creating tiny particles, and even nanoparticles [310]. Since understanding the fundamentals of powder metallurgy requires research into metal powders used as raw ingredients, particle shape is also among the general properties of metal powders, such as particle size, technological properties or bulk density, and compressibility [309]. Therefore, it is well known that the performance of the process and, consequently, the quality depend greatly on the size and shape of the powders used in additive manufacturing [164].

The uniformity of mixing two or more substances depends on how similar their particle properties are. Powder blends tend to be consistent when the components have an identical particle size, shape, density, and PSD. Metal and ceramic components, as well as medicinal and detergent tablets, are frequently manufactured using the powder compaction process. Even though a solid's bulk demonstrates that it can be compressed effectively, this does not mean that a coherent compact will follow. Particle shape is known as one of the metal powder properties such as particle size, PSD, flow rate, compressibility, apparent density, and purity [165] and determines the performance of powders during processing. In other words, metal powder characteristics such as particle size and shape are crucial for powder metallurgy processing as they affect the quality and functionality of finished products [265] because the shape (from spherical to flake) of the grains might differ greatly according to the process used to create powders (mechanical, atomization, reduction metal oxides, electrolysis, gas phase precipitation, spray granulation) [18]. It is also known that particle shape is important in determining bulk powder properties, the use of particles as an abrasive, and in applications involving packaging in powder compacts, slurry rheology, and flowability. Other physical and behavioral properties that are influenced by particle shape include packing and fluid interaction, angle of internal friction, powder flow rate, apparent and tap densities, and deformation behavior [260]. It has been reported that particle shape affects the efficiency and final properties of powder metallurgy products by influencing powder properties such as sinterability, packing density, compressibility, mechanical behavior, reactivity, and flowability [48]. For example, it has been suggested that spherical powders are desired in powder metallurgy processes such as hot isostatic pressing because they have high packing densities and low specific surface areas, and low oxygen content [311]. Since the bulk packing and flow properties of a granular feedstock are known to be influenced by the shape of the powder particles, it is thought that spherical particles would be packed and arranged more effectively than irregular particles because, in powder bed systems, spherical-shaped powders flow more freely, resulting in the creation of more homogeneous powder layers. It has been reported that, as the shape of the grains deviates from sphericity, the flowability decreases; that is, irregular, dendritic, and flaky particles are more likely to interfere with flow and friction than symmetrical ones. The inter-particle sliding is limited by spongy particles at greater angles of repose [268,269,277,311,312]. As the particle shape gets closer to being equiaxed, especially if it is spherical, packing density rises. Conversely, it becomes lower as the aspect ratio becomes lower. Even lower packing densities are achieved with spongy particles. Under vibration, irregular particles may have a high packing density due to their systematic orientation [275,276,311,313–315].

Compressibility is described as the ratio of initial powder volume to the volume of a compressed component according to particle characteristics such as size, PSD, and shape. It has also been stated that spherical particles are more compressible than irregular ones because they may reorganize more readily due to the internal pores in spongy particles' resistance to filling up, and internal porosity decreases compressibility [18,258,259,269,277,312,316]. Due to the large frictional forces, it has been reported that the tensile strength of tiny particles with a high shape factor increases quickly with a little increase in the packing portion. Spherical powder compacts are claimed to offer greater deformation capacity than oblate and flake-like powder compacts. However, plastic deformation is possible with spongy particles [277,297,298,317]. Since compressibility is known to influence the compact's green strength, it has been reported that the green strength of particles with irregular shapes is greater than that of spherical ones due to the stronger mechanical interpenetration of surface irregularities. However, it has been stated that, as surface oxidation of irregular particles is more likely to occur, their green strength decreased [18,277,305,316,318].

It has been found that the reactivity of irregular particles with a greater surface area and sharp edges is higher than equiaxed particles with a smooth surface. While the reactivity (chemical reactivity; oxidation; adsorption; explosiveness; catalyzing efficiency) decreases

the most for flake powder, it decreases gradually for granular powder, spherical atomized metal powder, and spherical atomized metal powder, respectively [18,277,315,319,320]. On the other hand, irregular particles with angular shapes and relatively high aspect ratios were found to have higher abrasive properties [275,306].

### 4.9.1. Thermal Spraying

Thermal spraying is a method that, in principle, employs a unit to melt and spray a substance onto a surface where it quickly solidifies, generating a coating that adheres well [321]. It has been reported that spherical particles are the best option for thermally sprayed coatings since they flow well and create the greatest conditions for particle deposition [306], yet irregular and spongy powders have a greater drag coefficient and thus a higher deposition performance [306–308].

### 4.9.2. Gas Atomization

It has been reported that the smooth surface and spherical shape prevent the development of cold compaction's green strength. It has also been suggested that pre-alloying before atomization also improves particle hardness and strength, which lowers compressibility. Due to the intrinsic constraints of gas-atomized powders, techniques for high-density consolidation at extreme temperatures, such as hot extrusion and hot pressing, can be applied. It has been reported that it is necessary to modify the shape of the particles away from sphericity by accelerating the cooling rate through a reduction in particle size. Additionally, some components that change the surface tension values may be added (Li, Mg, Si, Ca, Mn, etc.) [18].

### 4.9.3. Sintering Behavior

Granulation, the initial stage of the iron ore sintering process, produces granules of various shapes and sizes. The permeability sinter strand grate is filled with granules, which are mostly made of fine iron ore, fluxes, and coke breeze. Next, the top of the bed is heated using gas burners while air is drawn through the permeable grate. Following a brief period of ignition, the small combustion zone goes down through the sinter bed, gradually raising the temperature of each layer to between 1250 °C and 1350 °C. A sintered product results from partial melting and subsequent bonding between various grains after cooling. Then, molten iron is created using this sintered material [93].

It has been reported that the open porosity and permeability of porous metal must be maximized while maintaining a balance between material characteristics and sintering. However, it has also been reported that permeability and material characteristics such as strength and ductility are typically inversely connected, and the best balance can only be found in a relatively narrow space. Therefore, it is well known that the sintering temperature is chosen according to the type of material, the powder PSD, and the shape of the powder particle [18]. It is known that spherical particles sinter more slowly than irregular ones, and particles with higher specific surface areas typically show superior sinterability. Nevertheless, it has been reported that spherical particles are desired for loose powder sintering [202]. The impact of particle shape on the structure of the packed bed was predicted by Hinkley et al. [322]. They discovered that, while irregular particles are frequently likely to inhibit compaction and improve the permeability of the packed bed, rounded particles cause fewer particle-to-particle interlocks than angular particles and so enable easier compaction.

It has been reported that non-spherical particles are frequently used in conventional powder metallurgy despite their low flow rate because of their favorable pressing and sintering characteristics. It has also been stated that, because of their irregular shape, they have more points of contact with neighboring particles throughout pressing, which increases the likelihood that they will interlock and strengthen the green compact before sintering [323]. In other work, the mechanical properties, compressive strength, elastic modulus, and yield strength of compacts were compared through compression testing of

cylindrical samples using two differently shaped (atomized spherical particles and angular particles) Ti64 alloy powders. It has been found that, as powder shape deviates from spherical, compressive strength (Figure 25a), elastic modulus (Figure 25b), and porosity properties decrease [297].

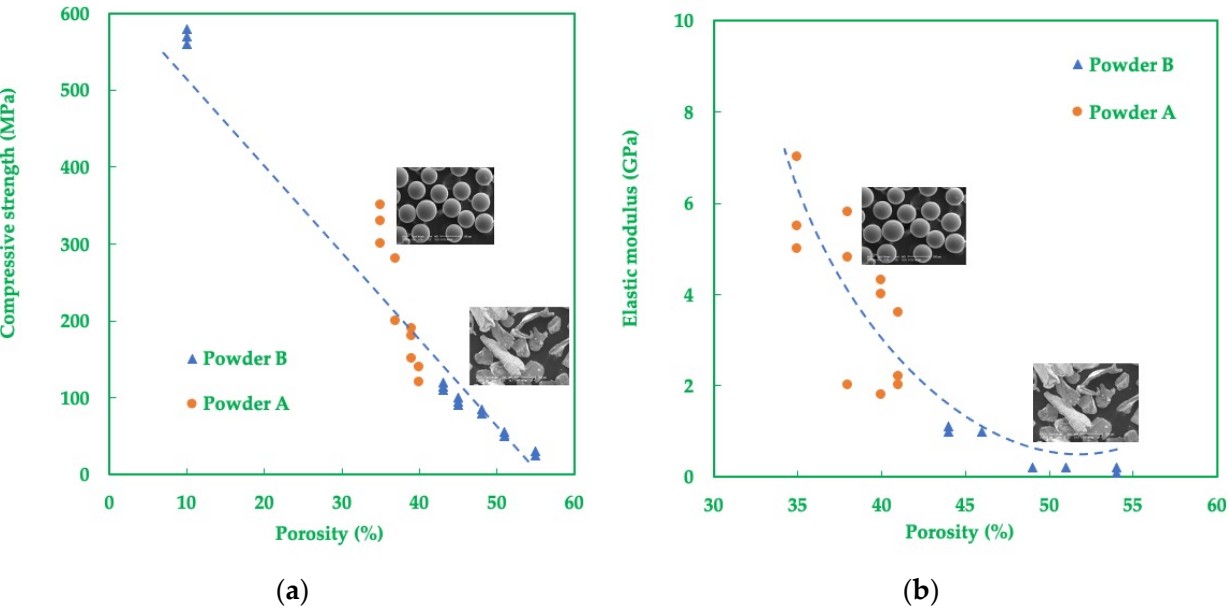

**(a)**                                                            **(b)**

**Figure 25.** Effect of porosity on (**a**) compressive strength and (**b**) elastic modulus for round and angular particles (modified from [297]).

In a recent study, the sintering activity and compressive strength of partly sintered porous alumina using variously shaped alumina particles (sphere, rod, and disk) have been examined. The spherical particles of sintered porous alumina were found to have a higher compressive strength than rod-shaped and disk-shaped particles. The applied stress was uniformly distributed across numerous established grain boundaries, which increased the compressive strength of the spherical particles. It has been reported that the packing density of rod-like particles is low, but these particles are tightly sintered with grain growth, and their shape is more spherical. Disk-like particles were also found to have a high packing density, but heating had no discernible effect on either the relative density or the microstructures. It has been stated that the spherical particles displayed a high packing density and were densified more quickly at a low heating temperature with little discernible grain development. The low Young's modulus and poor compressive strength of the alumina compacts made from the disk-shaped particles were an indication of the minimal establishment of strong grain boundaries during heating. On the other hand, it has been reported that the relative density of the spherical and rod-like particles increases with increasing Young's modulus and compressive strength. It has also been noted that the fracture occurs at low compressive stress because the applied force is localized on a small number of grain boundaries of disk- or rod-shaped particles in the porous compacts. It is concluded that the applied stress is uniformly distributed across several spherical particle grain boundaries, which enhances the compressive strength of the partly sintered porous alumina. It has been reported that porous electrode-supported solid oxide fuel cells and membrane filters employed at high gas or liquid pressures can both benefit from the aforementioned pore structure's strong mechanical qualities [324].

### 4.9.4. Optical Properties

It has been reported that the optical properties such as band-gap and photoluminescence of particles have both been affected by the variation in shape and size of zinc oxide nanoparticles synthesized by the polyol method [325]. In addition, it has been stated that

the shape, PSD, and mean particle sizes of the pigment all have an impact on light scattering of a coated layer of paper [326]. Furthermore, it has been reported that smoothness, homogeneity, optical characteristics, and other aspects of appearance are also influenced by the size and shape of the mineral particles. Mineral fillers' color, particle size, and shape attributes are all considered very important since they have an impact on the final products' characteristics [296].

*4.10. Mineral Particles in Ore Processing*

Before most ores may be transformed into consumable metals or end mineral products, they must first undergo some physical or chemical processing methods such as mineral processing or extractive metallurgy [327]. Furthermore, in the digital age we live in, it is inevitable to separate and recycle metals and plastics in order to prevent the waste generated by all electrical and electronic vehicles from harming the environment and to bring them into the economy.

Although particle shape is known to potentially alter classifier performance and PSD measurements [41,328,329], very few efforts have been made to measure, correlate, and characterize ground minerals and ores with regard to processes. Therefore, their importance for the beneficial use of particle shape in the mineral processing field is not fully understood [330]. The role of particle shape in various mineral processing unit operations can be given as follows.

4.10.1. Communition (Crushing and Grinding)

It is well known that, in geotechnical engineering, the particle shape of granular materials strongly influences their crushing strength. It has been found that the order of the particles' average peak stress levels recorded is cubic particle > spherical particle > natural-morphology particle [331]. It has been reported that particle shape changes with load frequency due to fracture. Laboratory impact-load tests on single- and multiple-size ballasts have revealed that, under impact loading, the angularity and concavity of the ballast particles eventually tends to decrease and they tend to become spherical. It has been found that small ballast particles are more likely to produce tiny particles, whereas large ballast particles primarily degenerate with sharply shattered corners. Interestingly, it has been reported that the flake-like particles produce more fine particles during the degradation process [332]. When the effect of different crushers employing different forces such as impact, centrifugal, vibrating, and compression on the particle shape of the crushed products of dolomite material have been studied, it has been found that impact crushers created mostly regular particles whereas centrifugal crushers provided more elongated particles [333]. When a basic shape parameter (angularity) from 2D images of the particles have been used to show how the shape of the particle changes after crushing in 1D compression, isotropic compression, and triaxial tests by using laser light diffraction technique, it has been found that angular sand specimens with a higher uniformity coefficient demonstrated less crushing than rounded sand specimens. The 1D compression tests of comminuted sand reveal that rounded sands become more angular, whereas angular sands become more rounded. It has been reported that particle morphology has a bigger impact on loose samples compared to dense samples. On the other hand, it has been found that randomly packed assemblages with excessively elongated particles may have higher shear strength and more dilatation [334]. It has been reported that the higher particle angularity would facilitate breakage and result in soil with reduced stiffness and substantially higher compressibility. In another study, it has been found that shape also influences the crushing pattern of particles regarding the breakage mechanism. It has been reported that spherical particles frequently break, with a major splitting in 80% of observed incidents [335].

It has been reported that particle shape is also important in the grinding process, which is known as the finer stage of size reduction in mineral processing. It has been reported that, while flake or irregular particles, which have the greatest frictional resistance to flow,

are likely to be held between the approaching balls, spherical particles, which have the least friction, are generally thrown from the colliding balls [18].

### 4.10.2. Classification

It is known that the properties of the sieved material such as PSD, particle shape, bulk density, moisture content, abrasiveness are crucial for an efficient sieve selection and design, since it has been reported that particles with different shapes (such as long, round, oval, or cubic) will have different screening characteristics and will make a big impact on the sieve opening selection [8]. In an interesting study, it was found that the particle shape had a significant effect on sieving efficiency for circular and rectangular sieve openings, but this effect was very small for square sieve openings [336]. However, it has also been reported that the particle shape plays an important role in preventing screen blinding [337].

It is now a well-known fact that air classification, as well as hydraulic classification, which is a widely used method for separating granular materials according to their settling rate in ore processing, is highly dependent on particle properties such as size, density, and shape [8,14,338,339]. For instance, hydrocyclone, which is a versatile classifier that sorts particles according to size, density, and shape, gives cut sizes down to a few microns and is commonly utilized in mineral processing plants not only for classification but also for de-sliming, de-gritting and thickening. It has also been reported that the performance of hydrocyclones is influenced by several factors, such as inlet velocity, solid content, liquid phase viscosity, particle size, and shape [328,340–342]. For example, it has been reported that platy particles such as mica, even if they are relatively coarse, often tend to be discharged as overflow [342]. In another study, investigating the effectiveness of a hydrocyclone for separating spherical and flaky (plate-like) aluminum particles with the same PSD, it has been reported that spherical particles tend to be separated more efficiently compared to flaky particles, as shown in Figure 26 [342].

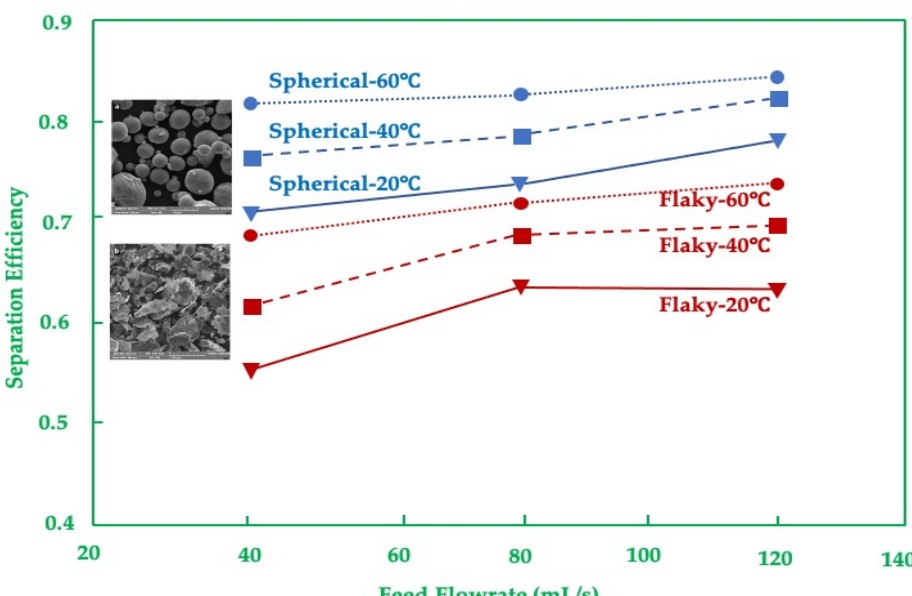

**Figure 26.** Comparison of spherical and flaky particles according to their separation efficiency and feed flow rate (modified from [342]).

### 4.10.3. Separation of Valuable Minerals

Although it is widely acknowledged that comminution, which is necessary for the separation of liberated valuable minerals from gangue, can result in modifications to particle shape, little attempt has been made to quantify, correlate, and characterize ground mineral particles [125]. Since the characterization of the particle shape distribution of feed, concentrate, and tailing streams in the mineral separation units will provide information

on the connection between the recovery of minerals and the particle shape, equipment and circuits can be designed for the best efficiency [32]. Therefore, the value of particle shape has also been investigated experimentally for physical concentration methods such as magnetic separation, electrostatic separation, gravity separation, and flotation [330,343–346].

Magnetic Separation

Magnetic separators are generally used not only for mineral and coal beneficiation but also for purifying solid and liquid materials. Although several previous studies have highlighted the importance of particle shape, which is thought to be one of the characteristics influencing the feedstock for magnetic separators such as dry high-intensity magnetic separators, induced roll magnetic separators (IRMS) [347], and Rare Earth Roller Magnetic Separators (RERMS) [348], empirical evidence has yet to be found.

In a study investigating the effect of particle shape on the forces in magnetic separation, by magnetic roll separator, it was found that the magnetic moment changes by up to 5% from that of a sphere of similar volume [349]. In another study, it has been stated that elongated, isolated particles (with w > 1) magnetize more easily than disk-shaped particles (with w < 1) [350].

On the other hand, it is seen that the magnetic properties of nanoparticles with different shapes are very important in nanotechnology. When CoNi nanofibers and CoNi nanospheres are prepared by the reduction of ions in a polyol solution under the application of magnetic fields of different intensities, using electron microscopy, $N_2$ adsorption, energy dispersive X-ray spectroscopy, X-ray diffraction, and magnetometry were characterized, and it has been found that the strong effect of fiber orientation on the demagnetization field is due to the shape of the nanofibers and nanospheres when the magnetization of the compacted powders is achieved [351].

Electrostatic Separation

Electrostatic separation is a widely used beneficiation method that exploits the differences in conductivity between different minerals to achieve the separation of valuable minerals, such as monazite, spinel, sillimanite, tourmaline, garnet, zircon, rutile, ilmenite, coal and fly ash [352]. Recently, it has been considered an efficient and clean method for recycling metals such as copper, aluminum, lead, tin, and iron from electric and electronic waste materials, which can have a harmful effect on our environment [353–358].

It has been reported that the shape factor affects the purity of conductors and non-conductors, electrostatic separation performance, and process variables [345]. For example, the effect of different particle shapes (sphere, cylinder, and flake) on their movement in Corona-electrostatic separation has been investigated by using three different shapes of metal materials from scrap printed circuit boards to establish theoretical models based on shapes. Results have shown that the motion behavior of three differently shaped particles is different. Spherical grains could be moved farthest, followed by cylindrical grains, and flake particles were thrown nearest [359].

In other research, which focused on computing electric field and the analysis of forces on the different particles such as acicular, flat, and spherical, the charging value of the particles was correlated with their surface area, since shape factors influenced the separating results of mixture metal particles [360].

In other studies, it has been proposed that the route of conducting particles is determined by the particle's characteristics (dielectric constant, dimension, shape, mass density) and how the electric field intensity is distributed at the active zone of the Corona electrostatic separator. Therefore, it has been concluded that an accurate calculation of the trajectory of the conducting particle from different materials or with varying process variables affecting electrostatic separation is crucial for the maximization of the application range of the sorting and for controlling the optimal metal recycling process [361].

In recent works, the separation efficiency of conductors in roll-type Corona electrostatic separation by simulation of the particle trajectory as a function of the shape, air drag,

and various charging situations has also been investigated. It has been found that the effectiveness of the electrostatic separation depends on the particle size and shape. In other words, smaller and flatter particles are easier to separate using the electrical and mechanical forces in a plate-type separator [359–362].

Gravity Separation

Although numerous attempts were made to optimize the impact of operating variables on the performance of gravity separation for the recovery of heavy minerals, experimental works using physical separation techniques in the last ten years have also demonstrated the value of particle geometry, such as particle size and shape [333,338,339,343–346,363].

For example, it has been reported in previously published studies that, although a shaking table separates particles mostly according to their density, it also does so to some extent depending on their size and shape [364–367]. Furthermore, it has been suggested that separation was mostly due to the hydraulic displacement of particles and that the shape of the particles affected their motion, making separation more complex, especially since flat particles were less likely to roll [368]. In addition, it has been reported that the velocities of the particles flowing in water will vary depending on the resistance forces in the medium, and the particles of different shapes moving in the water medium will be exposed to different resistance forces [369]. For example, it has been reported that the $-0.212 + 0.074$ mm mica mineral can be easily separated from feldspar due to its laminar shape on the shaking table [370]. Moreover, it has been suggested that optimization of the operation parameters of the shaking table and deck structure will improve the separation after the lamellar shape of vanadium particles has been produced by the milling process [371]. In another recent work quantifying the particle shape of all products in the shaking table circuits of a Turkish chromite beneficiation plant by using dynamic image analysis and visual techniques, namely scanning electron microscope (SEM) and stereo-microscopy, it has been revealed that tailing product particle shapes were more elongated than concentrate product particle shapes, which were more rounded (at a 95% confidence level). It has been stated that the difference in the particle shapes of the products is related to the rolling and sliding motion of the round and long particles on the table surface, and particle shape is one of the parameters affecting the separation by gravity [107].

Jig enrichment takes place when the suspension of particles is moving vertically and pulsing. The particles are stratified into groups according to their density and geometrical (particle size and shape coefficient) characteristics after a period of such motion. It is known that these characteristics have an impact on the settling velocity of the particles, which is the primary parameter of jig separation. It has been reported in previous studies that the shape factor is a crucial variable for optimizing the separation outcome of jig separation [343,346]. In a study investigating the impact of particle shape on the separation efficiency by computing distributions of settling velocities of spherical and irregular particles for a narrow (8–10 mm) coal size fraction, it has been found that the irregularity of particles influences the separation efficiency, which was measured by the imperfection change. In other words, the decrease in the settling rate of the irregular particle shape increased the probable error as a result of the decrease in the rate of non-homogeneity of the velocity distribution in the sample [372].

Nowadays, it is becoming increasingly vital to separate post-consumer plastics with various polymeric compositions to reduce waste and increase recycling [373]. When the impact of particle shape on the dynamic dense medium separation of plastic was investigated, it has been found that, using salt solutions as a medium, single particle sizes and shapes may be separated with great precision in a Tri-Flo separator [374]. It has been reported that the results provide a clear insight into how particle size and shape affect the cut density and the probable error since the shape variability resulting from the grinding of plastic scraps relies on the mill type. It has been reported that shape factors, combined with a mathematical description of the partition curve, helped in the process modeling, selection, and design of dense medium separation systems for recycling plastic [373].

Flotation

Flotation, which revolutionized the industry with its introduction in the early 1900s, is the most widely used fine mineral concentration technique today. It has been reported that more than two billion tons of minerals and fine coal are processed annually using only flotation worldwide [327,375]. It is known that flotation efficiency is determined by several possibilities, such as particle-bubble contact, particle-bubble attachment, transport between pulp and froth, and froth collection into the product launder [376]. Since flotation is a 3-phase process involving liquid, gas, and solid phase, particle properties (size, PSD, shape), the degree of mineral liberation, slurry properties, slurry flow rate, electrochemical parameters/potentials, chemical reagents and their addition rate, pulp levels in cells, air flowrates into cells, froth properties, mineralogical composition of the ore, mineral concentrations in the feed, concentrate and tailings and froth wash water rate, are the well-known important variables in flotation, which is a complex process [377,378]. In other words, it is known that the particle shape, which is one of the important properties of a solid, is one of the parameters affecting flotation as shown in Figure 27 [379], since particle characteristics such as wettability, size, and shape influence flotation by causing various sorts of interactions with the liquid–air interface [380].

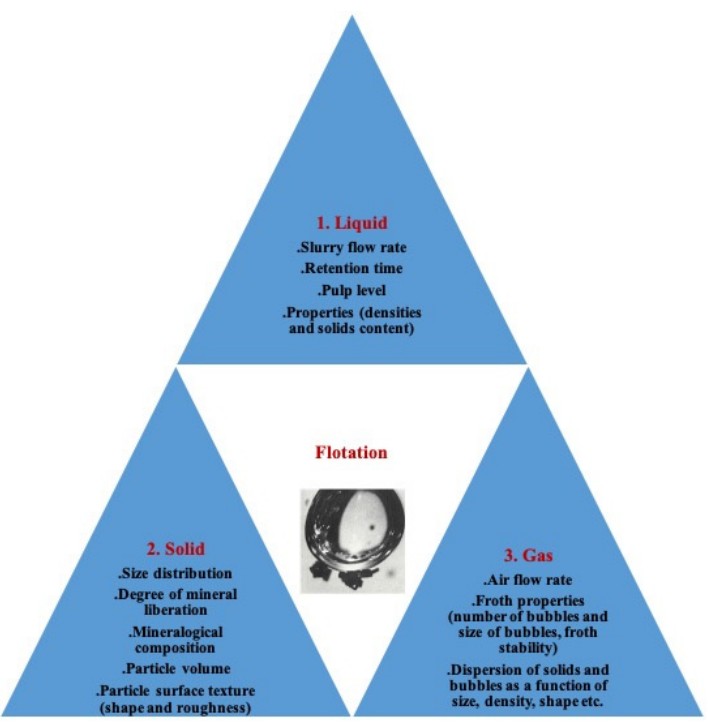

**Figure 27.** Phases and related factors affecting flotation, modified from [379].

The shape of comminuted mineral particles is an important physical criterion for the bubble-particle attachment in flotation because the shape of mineral particles produced by comminution equipment is typically non-spherical [381]. It has been reported that key factors such as surface roughness, heterogeneity, particle shape, particle size, and contact angle play a role in the wettability of real surfaces [382]. Despite the different opinions, it has been reported in several investigations all around the world that irregular particles have higher flotation recovery, flotation rate, contact line and area, flotation kinetics, advancing water contact angle, and bubble-particle angle than those of spherical particles [383–387]. It has also been suggested that there is a correlation between the flotation performance of mineral particles of different shapes created by various mills and the proportion of crystalline surfaces exposed to the mineral particles [388,389], since the physicochemical properties of exposed crystal surfaces affect the adsorption of collectors on mineral surfaces [390–393]. It has been found that regular and irregular particles behave

differently in the attachment of particles to a bubble for flotation since it has been reported that the edge shape of particles and their wettability both influence the adhesion force in flotation [394]. For instance, it has been found that particles with higher elongation and less roundness had higher flotation recovery than particles with lower elongation and more roundness [113,395]. In another study, it has been revealed that ground glass bead particles have higher flotation rate constants than spherical glass bead particles (Figure 28) [383]. When the variation of the induction time between spherical and irregular particles at various approach velocities was investigated, it was found that the particle shape also remarkably influenced the induction time. It has been shown that the induction time between a bubble and an irregular particle is significantly less than for a spherical particle [385,396]. In a recent study where the 3D Discrete Element Method (DEM) simulation model of particle–bubble interaction behavior and six types of irregular shape particle models were used, it was found that the critical induction time of irregular particles is lower than that of spherical particles [24]. It has been demonstrated that, in the flotation system, since the prismatic particles adhere to the bubble surface more easily than the round particles [397], sharp edges can successfully cause the liquid film to rupture and increase flotation recovery [24,383,398]. According to Ma et al. [386], during particle–bubble adhesion, the angular shape can easily tear the water film, which shortens the duration of attachment and speeds up floating. Additionally, compared to non-spherical particles, spherical particles have a smaller contact surface for the attachment of the particles to the bubble, which increases the likelihood that particles will detach from a rising bubble. Non-spherical particles are more likely to float because they have a greater chance of attachment and a lower chance of detachment.

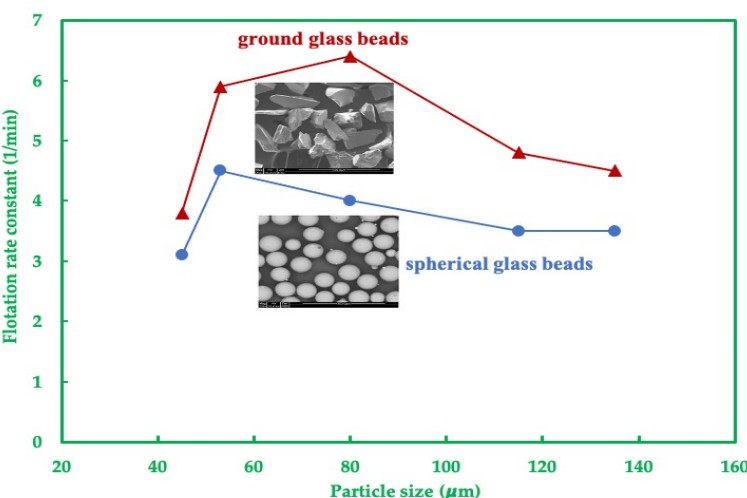

**Figure 28.** Effect of particle shape on flotation rate constants for ground and spherical glass beads at various sizes (modified from [383]).

Entrainment in Flotation

It is well known that the presence of too many fine gangue minerals (slime) in the separation of valuable minerals from gang minerals by flotation technique has a detrimental effect on the process [399,400]. Mechanical entrainment occurs because the hydrophilic gangue minerals are easily transported by the water to the froth concentration [399–402]. Moreover, it is widely known that gangue minerals' entrainment behavior is greatly influenced by the particle shape, in addition to particle size and density. It has been found that particles with different shapes exhibited different entrainment properties [403,404]. For instance, it has been reported that particle shape affects entrainment (Figure 29) by using detailed chromite mineral-specific shape characterization with QEMSCAN during the flotation of UG2 ore on South African platinum [155]. It has been found that round particles exhibit higher entrainment than angular, elongated particles, especially in the size range below 45 μm [155]. In another interesting study, it has been reported that the

entrained mineral recovery is linearly related to water recovery, and for minerals with different particle shapes there are significant differences in degrees of entrainment due to differences in settling rates that play a dominant role in entrainment [405]. Despite the dominance of their particle sizes and densities, it has been discovered that the particle shapes also had an impact on the differences in the degrees of entrainment for the three minerals [155,406]. It has been concluded that it would be wrong to estimate how particle density and size affect the entrainment of all other minerals without considering particle shape [404,406,407].

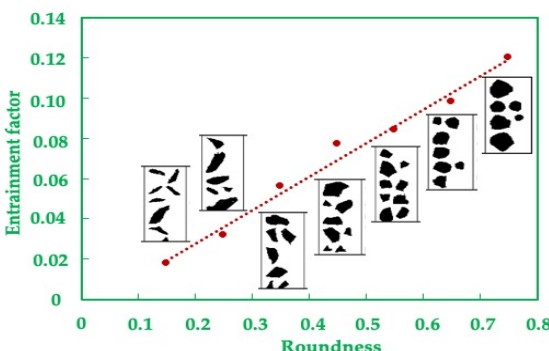

**Figure 29.** The relationship between entrainment factor in flotation and particle roundness (modified from [155]).

*4.11. Mineral Filler Particles Used in Various Industries*

Minerals are used in most of the major industries in a variety of forms, such as abrasives, ceramic raw materials, chemical, construction materials, consumer products, electronic and optical components, electrical, fertilizers, fillers, filters and absorbents, fluxes, food, foundry sand, gems and jewelry, glass, medical, pharmaceuticals, pigments, refractories, and oil well drilling fluids. It has been reported that the estimated total value of nonfuel mineral production in the U.S. alone in 2020 was USD 82.3 billion, while the total value of industrial mineral production is USD 54.6 billion. It has generally been reported that the value of mineral products after processing is 10 times greater than the value of mineral products mined [408].

Various mineral particles are employed as fillers, in coatings, as ad- and absorbers, or in catalysis in numerous industrial applications. Mineral fillers, which are finely ground mineral materials used in a variety of industrial applications such as paint, cosmetics, asphalt, and concrete [135], are inexpensive additions to give finished products certain desirable characteristics such as flow, fire resistance, density, heat conductivity, color, brightness, opacity, hardness, brittleness, impact strength, deformity, viscosity, softening, and electrical conductivity [409]. They are also known as mineral extenders as they improve and extend the qualities of the products in which they are utilized. Since the beneficial effects a filler can provide are, of course, linked to its physical properties, particle shape, which affects both processing and final properties, is often one of the criteria (as well as hardness, color, density, refractive index, and chemical inertness) to consider when choosing a filler. It has been reported that the majority of filler minerals fall within the elongated (or needle-like, flaky, platy, blocky) shape classifications [19–21,402,410–412] such as talc, mica, kaolin, attapulgite, precipitated calcium carbonate (PCC), and carbonyl iron particles.

It has been reported that the most used fillers in current research are at the nanoscale, and particle morphology has become the most important feature of the filler structure. Nanofillers, which are typically smaller than 100 nm, can be classified based on their dimensions as 3D (e.g., carbon black, fumed silica, or titanium dioxide nanoparticles), 2D (e.g., carbon nanotubes or cellulose whiskers) or 1D (e.g., graphene) [245].

### 4.11.1. Rubbers and Plastics

Rubbers are widely used because of their relatively poor polymer-filler adhesion potential, such as poor abrasion and tear resistance, and their low cost, and they can be used at very high loadings with little loss of compound softness, elongation, or flexibility. Since functional fillers help the tough, hard mineral absorb applied stress from the rubber matrix, the filler frequently has a big impact on the ingredients used to change the qualities of rubber products. The majority of rubber fillers provide a great value that enhances the rubber product's usability or ease of application. It is known that particle size, particle surface area, particle surface activity, and particle shape all affect the attributes a filler will add to a rubber compound, and particle aspect ratio is equally as useful for flat and needle-shaped fillers as particle size. This is because it has been reported that as the aspect ratio increases, Mooney viscosity, modulus, and hysteresis increase, flexibility and extrusion shrinkage decrease, and inclusion time increases [413]. Additionally, it has been reported that when these particles have a needle-like, fibrous, or platy shape they will more effectively block the stress from spreading through the matrix. It has also been reported that particles with a planar shape have more surface available for contact with the rubber matrix than isotropic particles with an equivalent particle diameter. It has also been found that, in contrast to a spherical filler, layered-shaped fillers with an aspect ratio greater than one can transfer stress more effectively, because Mica's non-spherical shape and silica's high filler–filler interaction are known to make them unrivaled materials with the best reinforcement properties [414]. It is known that the properties of ground natural products required for rubber are low aspect ratio, low surface area, and low surface activity. For example, the type of calcite that is most frequently used in rubber compounding is PCC with isometric prismatic particles. Similarly, finely ground barite is used in rubber, where its weight, inertness, isometric particle shape, and low binder demand are advantageous [19]. Thus, rubber reinforcement is known to demand granular particles, as elongated particles tend to generate a grain pattern that could cause the rubber to tear more readily along the direction of their alignment [22]. When calcite mineral particles are fewer than 20 microns in size and rounded, they can be employed as filler material in tire (wheel) and plastic manufacturing [415].

On the other hand, it has been reported that plastics prefer elongated reinforcements such as fibers over granular shapes since they need high impact strength rather than high tensile strength [22]. Moreover, platy talc is added to thermoplastics to improve dimensional stability, decrease creep in molded components, enhance molding cycles, and raise heat deflection temperature [409,410]. Similarly, asbestos, which is the most fibrous mineral filler, has been reported to be used in plastics and floor tiles since its effects on reinforcing and viscosity control [409].

### 4.11.2. Pigments and Cosmetics

Pigments, which are known as organic or inorganic compounds, either natural or artificial, that may absorb a portion of the light spectrum and then emit a portion of this light that corresponds to the color seen by the eye, are utilized in a broad spectrum of products such as inks and paints as well as in many industries such as textiles, food, and cosmetics [416]. Thus, it has been reported that the pigment quality is greatly influenced by the shape, morphology, and aspect ratio of the particles, i.e., particle shape is the basic feature of the powder influencing packing and bulk density, porosity, permeability, cohesion, flowability, caking behavior, attrition, interaction with fluids and the covering power of pigments [260]. It has been reported that the characteristics of the paint's flow and surface failure are influenced by the shape of the pigment particles. In the study conducted on pigments with similar particle size by precipitation [417], it was reported that as the shape factor of the clay particles increased, the in-plane tensile strength of the coating layer increased, while the elongation at break decreased [418].

It has been reported that cosmetics containing flaky particles exhibit better covering power and optical properties than those containing round ones [22]. For instance, in talc, only soft platy or lamellar ones with very high purity and brightness are preferred [19].

Dolomite fillers are considered one of the raw material options for fiberglass manufacturing [419]. For example, recent research has reported that low-aspect ratio dolomite fillers are less likely to affect modulus and strength than high-aspect ratio ones, but perform better at increasing impact strength [245].

### 4.11.3. Paints and Coatings

It is generally accepted that pigments (Figure 30) with a greater aspect ratio typically provide better properties such as coverage and enhanced gloss [20,420]. In addition, it has been reported that inks and paints with flaky particles show better covering power and optical properties compared to those with round particles [22].

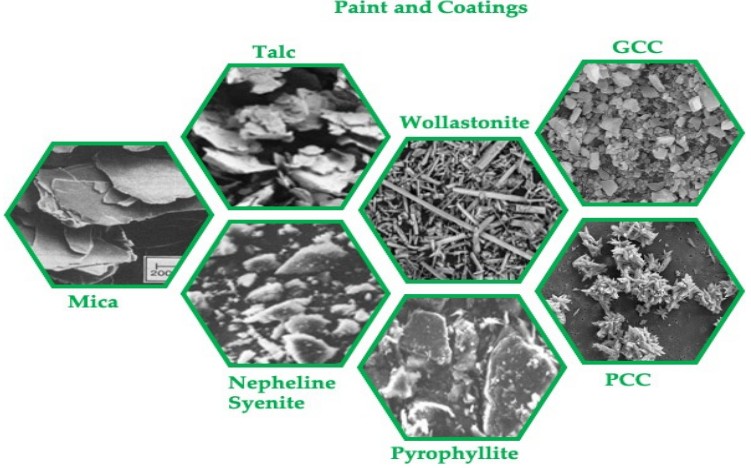

**Figure 30.** Effect of particle shape on paint and coatings (modified from [21]).

For instance, fine, wet-milled, flat mica mineral particles are known to serve as the foundation for special-effect pigments and exterior latex paints for tint retention and weatherability [21]. It has been reported that the use of mica-reinforced polymers are expected to increase significantly, especially in the automotive sector [409], due to the highly platy particle shape that creates layers parallel to the paint coat [421].

Another example is the use of platy talc in coatings due to its high oil absorption, softness, and brightness to help in gloss control, $TiO_2$ spacing, anti-settle, inter-coat adhesion, and corrosion resistance [21]. It has been reported that talc not only reduces the cost of paint recipes in interior decorative paints, but also improves performance because of its surface hydrophobicity and lamellar particle shape. Moreover, it has also been reported that platy talc, which is an indispensable additive for paints with low moisture permeability, provides a satin and matte finish in various paints [245].

Similarly, platy kaolin clay (aluminum silicate mineral) has been reported to be preferred for interior primers and flat paints. It is also well known that pyrophyllite, a platy aluminum silicate, can be used as an interior primer, an affordable flat paint, a texture paint, and as a mica substitute in joint compounds [21], while platy novaculite, which is a kind of silica material, provides extra properties such as mar, wear, and weather resistance. In fact, it has been reported that one of the reasons flat product ingredients such as pyrophyllite, kaolin, and mica are preferred is that they assist in film drying and develop resistance to film cracking while promoting proper dispersion by preventing pigment settling [21].

It is well known that wollastonite, which is a calcium silicate mineral with an acicular (needle-like) particle shape, is preferred in paints because it shows low oil absorption [21]. It has been reported that it is also utilized in coatings as a flash- and early-rust-resistant extender pigment, especially in water-based DTM primers due to its corrosion resistance,

exterior latex paints due to its tint retention feature, indoor flat paints due to its resistance to scrubbing, and powder coating as finely ground products [21]. Its acicular structure has been reported to give texture paints mud-crack resistance as well as great scrub resistance and film longevity. It is also well known that fine-ground and micronized grades offer smooth flow, water resistance, enhanced wet adhesion, and high gloss in epoxy powder coatings [21].

On the other hand, tremolitic talc (magnesium silicate mineral), because of its prismatic shape, has a lower oil absorption rate and is known to offer easier dispersion, greater loading levels, less flattening, and better dry hide than flat talc. It is also stated that it makes exterior and traffic paints more durable [19].

GCC, which is formed in various crystal habits in different shapes such as blocky (chalk), scalenohedral (calcite), and short needle (aragonite), has been reported to be widely employed in all types of paints and coatings, especially in interior and exterior architectures, due to its high gloss and low oil absorption [21].

Another example used as a paint filler mineral is Nepheline Syenite, which is silica-free and has an irregular shape, and is used in a variety of paints and coatings because it gives flat paint strong scrub and excellent exterior weatherability, color and gloss retention and resistance to chalking and frosting [21].

### 4.11.4. Ceramics

It has been reported that nepheline syenite, aluminum oxide, bentonite, or fly ash are used to make ceramic spheres, which have much higher densities than glass or polymer beads, yet because of their thicker walls, they are less expensive, more robust, and mechanically durable. They also have high packing density and enhanced flow properties as all spherical materials due to the ball-bearing action. Moreover, they enhance the crack resistance of speckling compounds and decrease the dielectric constant, warpage, and shrinkage [245].

It is well known that both natural and synthetic platelet and short-fiber systems are widely used in ceramic science and other fields. While the tabular forms of numerous oxides, nitrides, and carbides ($Al_2O_3$, BN, SiC), kaolinites, various clay minerals (montmorillonite and hydrotalcite) and phyllosilicates (talc) are all present in platelet systems, SiC (whiskers), $Al_2O_3$ fibers, and wollastonite are all present in short-fiber systems that are utilized in ceramic technology [62].

### 4.11.5. Abrasives and Explosives

It is very well known that particle characteristics also affect how well abrasive minerals work in cutting and polishing equipment. For instance, it has been reported that the abrasive coefficient (Ka) used to indicate the abrasive qualities of dust relies on the hardness, particle shape, density, and particle sizes [18]. It has been reported that there is a strong correlation between angularity and the rate of abrasive particle wear [306]; therefore, angular particles are favored for wear applications. Similarly, it has been reported that abrasive characteristics are greater for irregular particles with an angular shape and a relatively high aspect ratio [275,306]. It has also been reported that angular particles are preferred over flat particles in the abrasive industry because they tend to slide rather than roll and therefore have a low chance of imparting their sharp properties to the surface to be abraded. On the contrary, it has been reported that smooth spherical particles are preferred in the explosives industry because irregular particles, especially those with sharp edges and corners, cause the initiators to become chemically unstable [22].

### 4.11.6. Paper

It has been known that fillers used for papers can significantly alter the structure of the paper, impacting the way it performs in its final application. Therefore, the shapes of the filler particles, and how those shapes might influence the material's apparent density, strength, optical qualities, and other crucial characteristics, are important [296,422–426]. In

other words, it is known that the different shapes and sizes of mineral filler particles give today's modern papermakers a variety of choices for modifying the characteristics of paper. Thus, it has been reported that the size and shape of the mineral particles can provide a benefit for mechanical qualities such as stiffness and tensile and impact strength in the paper sector [296], e.g., it has been reported that particles that resemble plates can be useful for paper goods with a high apparent density [427]. Furthermore, it has been known that platy fillers offer papers with greater gloss and less porosity, improving their suitability for printing [20,296]. For example, high-purity, micronized platy talc is the favored ingredient for pitch adsorption and paper coating and filling [427]. It has been also reported that the typical particle shape and aspect ratio (length/width) values of ground calcium carbonate (GCC) for paper-making fillers are blocky and 1 to 1.5 [411]. On the other hand, in a study examining mechanical, optical, and rheological properties of mineral-filled polypropylene, and filled and coated paper, it has been reported that the desired shape of the grains is close to spherical in flowability, ease of transportation, processing, packaging and mixing processes [296].

*4.12. Graphite Particles Used in the Battery Anode Material*

Nowadays, the development of electric vehicles and renewable energy sources makes it crucial to store a significant amount of energy in electrical form. Because of their unique benefit of having a high energy density, batteries, particularly lithium-ion batteries (LIBs), are the most extensively used efficient energy storage technology for electric vehicles, as well as portable electronic devices including laptop computers, cell phones, and camcorders [428–431]. In other words, LIBs have been utilized extensively in a wide range of applications because of their advantages of having some unique properties such as long storage life, no memory effect, and a low self-discharge rate. Since the main reasons for choosing a battery are its operating time, safety, cycle life, power, energy density, and costs, the four main elements that make up the battery, the cathode, anode, separator, and electrolyte, are important. For the fabrication of batteries, and particularly for the production of electrodes, a variety of powders are employed. While many studies concentrate on the chemical characteristics of the trio anode/electrolyte/cathode, it is known that the characteristics of the powders used for the batteries have a significant impact on the production of the electrodes and, consequently, the quality of the batteries. A comprehensive characterization of these powders used in the battery is therefore vital to make the best battery possible. Thus, cathode materials and anode materials for LIBs have undergone extensive research to address the growing demand for affordable, long-lasting, and energy-density efficient energy storage devices [432]. Therefore, it has been reported that a lot of interest is being paid to electrode materials, particularly anode materials, e.g.,various high energy density materials (nitrides, tin oxides, tin-based alloys, silicon, and their composites) and enhanced carbon materials (petroleum coke, mesocarbon microbeads, and natural graphite).

Natural graphite, which can take and release lithium ions between the layer structure in charge and discharge processes, is regarded as the most beneficial material for the anode in LIBs among these options due to its higher degree of graphitization, which results in reduced production costs and greater capacity, superior charging efficiency, energy density and cell capacity [433–435]. Therefore, the use of graphite with high structural quality and purity, the right particle size, and the best morphology for efficient lithium-ion intercalation chemistry is the foundation for anode enhancements that have tremendous industrial potential growth [436], since more than 95% of the anode material used in all commercial battery technologies (LFP, NMC111, NMC532, NMC811, and NCA) produced today is graphite, indicating that it is a major raw resource and one of the primary materials used in anode production [437]. Thus, the creation of anodes with higher velocity capacity and energy density, lower first-cycle irreversible capacitance loss, longer cycle life, and better safety performance depends on optimizing the morphology of the graphite particles [438]. Therefore, particle shape, which is reported to be of typically under-appreciated or overlooked significance in battery electrode materials [439], must

also be understood and optimized by manufacturers to reach the optimum level of battery performance, as the packing density, porosity, and homogeneity of the electrode coating are all impacted by particle shape on the slurry rheology [440] because particle shape, as well as the particle size of raw materials, has been reported to have an impact on the tapped density, i.e., it has been reported that the tapping density and circularity properties of particles are positively correlated [441].

Due to the structural instability of natural graphite used as an anode material during the loading/unloading process, such as poor performance as a result of swelling and crack formation along the particles through repeated volume changes during cycling [442], significant initial irreversible capacity loss, short cycling life, and unsatisfactory rate performance in practice, modification such as particle size change, morphological change, and the interface chemistry become vital for long-term cycling performance [221,443–448]. Therefore, it has been reported that raw natural graphite flakes are transformed into spheres to improve their electrochemical performance by impact milling and thermal vapor decomposition. In other words, it has been reported that natural spherical graphite is often manufactured through impact by milling small natural graphite flakes, which are then bent concentrically and formed into a spherical shape [449–451]. Since spherical particles that are well rounded have a low specific surface area, which reduces capacity inefficiencies, it has been reported to have various benefits over conventional flake graphite when utilized in LIB applications: more effective packing, anode's high tap density, and high volumetric energy capacity. It has been reported that the produced spherical graphite samples exhibit good performance in terms of high-rate capacity, high reversible capacity, high coulombic efficiency, and low irreversible capacity [450].

### 4.13. Microparticles in Magneto-Rheological Fluids

When a magnetic field is applied to multiphase systems with at least two phases, known as magnetic suspensions, it has been reported that there is a serious transformation in the rheological, magnetic, electrical, thermal, acoustic, and other mechanical and physical properties of the material. In some circumstances, the solid phase is the component that acts on two important magnetic fields, such as magneto-rheological fluids and ferrofluids [452]. While the last mentioned of these liquids, namely the concept of ferrofluids, will be included in Section 4.14.2, the first mentioned concept is discussed in this section.

Magneto-rheological liquids are described as liquids prepared by dispersing spherical micro-particles that can be magnetized in a generally non-magnetic medium. It has been reported that the structuring obtained by an external field can support shear stresses by providing huge-field-dependent viscoelastic modules and yield stress due to the enormous-magnetic multi-field-particle size [453–455]. It has been reported that the magneto-viscous and the magneto-rheological effects are produced when a magnetic field is applied, and magneto-rheological fluids polarize multi-domain magnetic particles and cause them to aggregate into agglomerates [456]. It has been reported that these outcomes are significant for practical systems. In moderate magnetic fields, these effects are manifested as a relative increase in viscosity and yield stress of approximately three orders of magnitude. It has been reported that the removal of the magnetic field that induces the magnetic moment of the particles will result in the suspension returning to its original conditions, and the functional viscosity alteration and yield stress phenomenon are reversible. As a result, it has been reported that the basic processes in typical magnetorheological fluids are linked to the potential for utilizing mild magnetism to influence or regulate their architecture and, consequently, their rheological function. It has been reported that, since particle sizes used in magnetorheological fluids compositions typically range in the micrometer range, the particles become strongly magnetized because the volume of the particles affects their field-induced magnetic moment [457]. Since magneto-rheological fluids exhibit enhanced rheological behavior that includes a viscoelastic response, shear thinning, and yield stress, all of which may be controlled externally using magnetic fields, it has been reported that they have applications in three distinct fields such as colloids with tunable rheological

properties, patterned anisotropic self-assembled materials, and sensors for monitoring mechanical vibrations [457].

It has been reported that conventional magneto-rheological fluids are used in applications for active vibration control and torque transmission in a variety of engineering sectors such as civil (semi-active dampers for building and bridge foundations), automotive (clutches, brakes, and magneto-rheological dampers used to reduce vibrations transmitted to passengers), household applications (washing machine dampers), sports applications (gym equipment dampers), military equipment, and robotics due to their ability to be controlled and tailored [454,457–465]. They can also be found elsewhere, including in chemical sensors, biomedical (artificial joints with integrated dampers), polishing techniques, sound transmission, thermal energy transfer control, magnetic field sensors, instruments with optic bi-stability, and light valves [466–476].

In studies on particle shape, it has been demonstrated that poly(p-phenylene-2,6-benzobisthiazole) rod-like particles improve both the dielectric interaction between particles and their mechanical strength [477,478]. Progress was then made in calculating the rheological characteristics and orientation patterns of particles in a heavily diluted colloidal suspension of ferromagnetic spherical cylinders [479,480].

The next development in this area has been the incorporation of magnetic needle-like particles of $Co-Fe_2O_3$ and $CrO_2$ into the formulation of conventional magneto-rheological fluids [481], which have been found to improve stability versus quick sedimentation [31]. A recent study showing that particle shape significantly affects structuring under an external field reported that elongated magnetic particles perform significantly better magneto-rheologically under small-amplitude shear and simple steady shear flows [482]. When the effect of particle shape on magneto-rheological performance has been investigated by using different-shaped (sphere, plate, and rod-like) iron particles suspended under the presence of magnetic fields, it has been reported that magneto-rheological fluids containing rod-like particles exhibited higher storage modulus and yield stress compared to their sphere and plate equivalents [31]. This was attributed to the fact that non-spherical grains are more readily magnetized when their long axis is parallel to the external field.

### 4.14. Nanoparticles

Nanoparticles are currently investigated and employed extensively since their nanoscale dimensions offer superior features in them, which may be very different from those of the relevant bulk material. It is known that nanoparticles have different properties when compared to the same material at the macro scale. Therefore, these novel physical, chemical, and biological characteristics appear when materials are reduced to sufficiently small sizes, often fewer than 50 nanometers (a few molecules), opening up possibilities for new uses. Thus, it has been reported that benefits arise in the form of increasing the surface-to-volume ratio and increasing the quantum effects that determine the optical, magnetic, and electronic properties of the material by means of molecular modification and also optimizing these surface properties for specific purposes [483]. In other words, the advantage of being at the nanoscale is that it enables the design of nanoparticles with multiple functions [484]. Nanoparticles have become popular for various applications. For example, they are used as inert chemical additives such as fillers in advanced composite materials and UV absorbing pigments with high-refractive-index in cosmetics and other consumer goods, as chemically active particles in the biomedical, catalytic, and biotechnological industry, and as drug delivery agents in the pharmaceutical industry [485–487]. Furthermore, it was suggested that silver nanoparticles synthesized using a fungal extract of Aspergillus fumigatus have promising antimicrobial capabilities and might be useful for a wide range of biological applications [488]. In addition, it has been suggested that endophytic-mediated ZnO nanoparticles are biocompatible and have a great deal of promise for use in agriculture [489]. Furthermore, the wide application possibilities of CdO nanoparticles in extensive biomedical applications are demonstrated by their high antibacterial activity [490]. It has also been reported that nanoparticles have the advantage

of interacting with biological systems in unexpected ways since they share the same size range as the majority of components in living cells, such as proteins, nucleic acids, lipids, and cell organelles.

Moreover, they are used in the production of nanomaterials that are fundamental to today's technological developments, since nanoparticles have special characteristics (physical, chemical, and biological properties such as melting point, wettability, electrical and thermal conductivity, catalytic activity, light absorption, and scattering) depending on their size, composition, origin, shape, and PSD [491]. Thus, it is critical to be able to quantify these characteristics effectively and precisely, to make sure their properties are used to their best potential [492]. The importance of nanomaterials morphology, which is defined as the shapes, sizes, and structures of nanostructured materials and their distribution in composites, has started to be understood more recently. It is significant for identifying features that may greatly vary from the recognized bulk properties [54], as morphology determines physical and chemical features [52,53].

The use of nanotechnology enables the creation of a variety of classes of particles with a wide range of properties including size, shape, density, and surface chemicals. For example, they can be manufactured for specific medical applications such as drug delivery systems [62]. Nanomedicine, which spans the fields of chemistry, material science, engineering, and medicine, takes advantage of the special properties of nanoparticles to provide more effective anticancer therapies [493,494].

While the majority of early-generation particles were spherical, more recent developments have taken advantage of nanoparticles' ability to be engineered to give them specific geometrical, physical, and chemical properties. For instance, nanostructures such as flat and rod-shaped ones (nanorods [495], nano chains [496], and nano worms [497,498]) have been created that are ideal for biomedical applications. It has been reported that the physical properties of nanoparticles, such as density, size, and shape, play a key role in their delivery to tumors [499]. Thus, this section, which describes the effects of nanoparticle shape, covers main areas such as superparamagnetic materials, ferrofluids, pharmaceutical manufacturing, drug delivery, disease diagnosis and treatment, advanced immunotherapy, drug dissolution, and catalysis.

### 4.14.1. Nanoparticles' Super-Paramagnetic Properties

It has been suggested that many materials show enhanced physical properties when built to a specific structure or morphology [500]. For instance, the magnetic characteristics of the nano chains are reported to exhibit a superparamagnetic mode at ambient temperature, considering that they are made up of microscopic nanoparticles as their building blocks. The measured magnetic parameters demonstrate that the magnetic moment of the nanoparticles in the nano chains is greater than that of the individual nanoparticle clusters, e.g., it has been reported that bulk maghemite ($\gamma$-$Fe_2O_3$) turns into a ferrimagnetic substance when the Curie temperature reaches 950 K [501]. Furthermore, it has also been reported that $\gamma$-$Fe_2O_3$ can display super-paramagnetic characteristics after reduction to individual particles smaller than 15 nm [502–507]. Then, magnetite nanoclusters were synthesized and allowed to self-assemble into 1D nano chains via a magnetic field-induced assembly method [508]. Thus, it has been reported that the transition from the superparamagnetic state to the ferromagnetic state is made possible by the formation of the 1D nanochain structure. As a result, it has been reported that the samples exhibit super-paramagnetic capabilities and that the magnetic moments of the nanoparticles in nanochain constructions were amplified compared to those of their spherical building blocks. It has also been shown that the magnetic characteristics of the constructed maghemite nanoparticles rely on the structure's shape [500].

### 4.14.2. Nanoparticles in Ferrofluids

It has been reported that nanoparticles can offer cutting-edge technologies capable of customizing the thermal properties of the heat transfer fluid by controlling the particle

size, shape, and other factors in heat transfer fluids. Thus, it has been reported that these findings can be used to forecast the activities of fluids concerning heat transfer, temperature, and velocity profiles to create a more efficient and environmentally friendly process [509]. Since ferrofluids often have particles that are a few tens of nanometers in size, it has been reported that they show a continuous magnetic moment as a result of being magnetic mono-domains, where the Neel relaxation effect predominates [457]. Moreover, the carrier fluid, temperature, particle size, shape, loading, magnetic properties of the particles, and the applied magnetic field have been reported to affect the behaviors of these fluids [509]. For example, in a study discussing the effects of particle shape on heat transfer rate and nanofluid flow, it has been reported that the heat transfer rate can be improved by taking different particle shapes [510]. According to the results of a theoretical study on ferrofluid using spherical nanoparticles, the heat transfer increase, which was 7.86% in the absence of a magnetic field, increased to 8.73% in the presence of a magnetic field [511]. In another study, examining the effect of the shape of different iron nanoparticles (spherical, flat ellipsoid, and prolate ellipsoid) in the flows on the velocity and temperature profiles, convective heat transfer coefficient, and radial and transverse shear stress due to a highly oscillating magnetic field on a stretchable rotating disk, it was also found that nanoparticles with different shapes of the same size also have effects on the velocity profiles due to their effects on the physical properties of the fluid. It has been reported that the highest velocities were observed in prolate-shaped particles, followed by oblate and spherical particles, respectively [509].

### 4.14.3. Nanoparticles Used in Pharmaceutical Manufacturing

The use of nanotechnology in biomedicine has attracted a lot of attention. The pharmaceutical sector is one application of particle technology that is extremely new and challenging, e.g., the uses of manufactured particles in biomedical research, disease diagnosis, and even therapy are endless, especially with the recent advancement of nanotechnology. Since many pharmaceutical products are available in powder or pellet form, particle technology is vital for developing, adopting, and optimizing pharmaceutical processes [9]. As the drug contains several active components, the physicochemical characteristics, bioavailability, and other pharmacokinetic phenomena of these components may be influenced by their particle sizes and shapes. Therefore, employing several of the sizes and shape analysis research techniques is becoming more popular and is being evaluated for many novel drug preparations currently being developed [512]. It has been reported that the manufacturing and terminal features, such as slurry filtration, compression, mixing, flowability, and bioavailability, are affected by the size and shape of active components and excipients [513]. Moreover, many research works have reported that the selection of processing [514] and operational factors, including solvents [515], supersaturation [516], temperature and agitation rates [517], or the existence of impurities [518], also depends on the particle size and shape [94]. In other words, controlling shape and crystal form is considered crucial in the pharmaceutical industry as they have an impact on downstream processing steps such as filtering, drying, and milling. In addition, some distinguishing properties such as deformability under compression, flow under certain conditions, and compressibility to some extents have been the subject of research efforts to understand the behavior of powders [8]. For example, it has been reported that the final encapsulated product's quality depends heavily on the size and shape of the particles in spray coating or fluidized bed processing [519]. It has been found that better encapsulation is generally proportional to the degree of particle sphericity, as sharp edges can protrude from the applied coating surface and become vulnerable to release, and a thicker coating is obtained by reducing the amount of surface area required for coating. As a result, before using this method, it has been reported that the shape of the irregularly shaped particles can be improved by modifying their structure [520]. Since particle shape can assist in recognizing and characterizing the many sub-components of a final product based on shape differences [521], it has also been reported that the active pharmaceutical ingredient (API) and excipient particle size and

shape must be studied and controlled when formulating and scaling up processes for both innovator and generic pharmaceuticals [270].

### 4.14.4. Pharmaceuticals Particles in the Dissolution Process of Drugs

Since particle size and shape are the critical quality characteristics of materials used in the manufacturing of many pharmaceuticals, it is known that the dissolution behavior of many particles used in applications including pharmaceuticals, nutraceuticals, cosmetics, and food where controlled release is needed, the bioavailability of active drug ingredients, and other pharmacokinetic phenomena can be affected by the particle sizes and shapes of these components. When a multicomponent controlled-release pharmaceutical is dissolved, a good understanding of the size and shape of the particles is required during formulation and production, as initially spherical or granular particles will gradually change shape as they are disintegrated [512]. Since it has been well known that the particle shape affects the physical and chemical properties of API, such as solubility and dissolution rate [522], many studies have reported that non-spherical particles such as elongated spheroids, discoids, and dumbbell-shaped colloids display different movement and migration characteristics compared to the homogeneous spherical particles, which are generally used as models [523–527].

For instance, when the effects of both particle shape and size and their combined effects on the dissolution rate of poorly soluble pharmaceuticals such as Griseofulvin, barium sulfate, oxazepam, and glibenclamide were investigated, it was reported that the decrease in the dissolution rate as the level of flakiness and disorder increased for particles of the same size is due to the increase in the mean hydrodynamic boundary layer thickness [528]. In another study examining the effects of particle shape by simulation and comparison on the mass diffusion-controlled dissolution process using five different shaped particles with the same initial volume and maximum cross-sectional area, it has been found that the variation of dissolution time for these five distinct particle shapes increases with the Reynolds number, and the particle shape has an impact on the dissolution time regarding the surface area, the convective intensity, and the intensity of flow recirculation. It is also thought that, when there is high recirculation behind a particle, a sharp corner would form during the dissolving process [529].

### 4.14.5. Nanoparticles Used in Drug Delivery

Drug delivery is another area of the pharmaceutical sector where particle technology is used extensively [9]. In the pharmaceutical industry, if the drug loading in the dosage form is high (usually 50% w/w and more of the dosage form weight corresponds to the drug weight), and the drug particles exhibit good flow, have an ideal shape (spherical), and have a density comparable to the other ingredients in the dosage form, it is said that the uniformity of the content of the drug is typically good [270]. In recent research, it is well known that pharmaceuticals can be delivered to specific locations under controlled conditions using micro- and nano-carriers. Nowadays, the unique effects of nanoparticles in drug delivery have changed the paradigm and revolutionized the field of pharmaceutical and drug delivery. It has been reported that improved bioavailability [530], site-specific delivery [531], decreased toxicity [532], and in vivo stability of macromolecules such as proteins [533] and nucleotides [534,535] are just a few benefits of encapsulating pharmaceuticals into nanoparticles. It is widely known that after being treated in vivo nanoparticles have a specific pharmacokinetic and biodistribution behavior that is controlled by the complicated interaction of their size, shape, surface charge, and surface hydrophobicity [45]. It has also been reported that the flow qualities of blood (such as flow rate, red blood cell deformability, and hematocrit, which is the volume fraction of red blood cells), vessel size, and particle characteristics (such as size, shape, and deformability) all play an important role in the cross-sectional distribution of micro- and nanoparticles. In other words, the size and form of drug carriers are crucial factors in how well they adhere to surfaces and can pass through biological barriers (e.g., internalization). Previously reported experimental

studies have demonstrated that drug delivery is influenced by particle shape, which affects colloidal processes at nano- and micro-scales [536,537]. Thus, the efficacy of nanoparticles in vivo has recently been linked to a new physical parameter known as particle shape, which has a significant impact on cellular uptake and biodistribution [45]. Furthermore, the manufacture of carriers in a variety of shapes, including spherical, prolate, oblate ellipsoidal, and rod-like geometries, is made easier by advancements in micro- and nanoparticle fabrication [538].

Moreover, it is also known that the cellular uptake of nanoparticles, which can be defined as interacting with various cell types according to the target region by in vivo administration, plays a crucial role in the intracellular delivery of therapeutic agents. Since the vital role of particle shape on cellular uptake, uptake kinetics and mechanism, intracellular distribution, and cytotoxicity of nanoparticles has been acknowledged, key research in this area has been motivated by the sharp influence of particle shape on cellular internalization of nanoparticles employing a variety of cells, such as macrophage, epithelial, endothelial, and immune cells [45]. According to a recent study it has been reported that ellipsoidal particles have slower rotation dynamics in blood, which makes them ideal for drug delivery [539].

### 4.14.6. Nanoparticles Used in Disease Diagnosis and Therapy

A promising method for the early diagnosis and treatment of diseases such as cancer is also thought to be the use of targeted micro and nanocarriers for the administration of imaging agents and medications [540,541]. Thus, recent breakthroughs in nanotechnology have led to the development of nanoparticles of various sizes and shapes that have been demonstrated to possess special qualities suited for biological applications, including cancer therapy and imaging. The importance of nanoparticle shape in the development of vascular-targeted carriers for disease therapy and diagnosis is increasingly recognized [542].

For instance, vascular-targeted carriers are known to provide highly localized therapeutic/imaging agent delivery, which presents a new chance to enhance disease diagnosis and therapy. The common geometry for vascular targeted carriers has been reported to be spherical with diameters on the nanometer scale, irrespective of the kind of carrier (for example, polymeric, liposomes, dendrimers, or micelles). However, recent studies have indicated that disorders affecting larger vessels with substantial blood flow, such as atherosclerosis, may not be best targeted with nanospheres [543,544]. The performance of vascular targeted carriers may be enhanced by departing from the traditional spherical shape, according to recent literature. Particularly, it has been suggested that macrophages are less effective than spheres at engulfing elongated particles of adequate volume, which may result in decreased clearance and lower accumulation in organs that are not specifically targeted [545]. It has also been demonstrated that different shapes of microparticles with equal volumes exhibit varied in vivo biodistribution profiles in tumor-bearing mice [523]. Additionally, it has been reported that drug-loaded filamentous micelles circulate much longer than spherical micelles in mice [546], and ICAM-1-targeted elliptical discs have a longer circulation half-life and better targeting specificity than their spherical counterparts [547]. It has been reported that ellipsoidal grains show a preferential horizontal drift in shear flow close to a wall (similar to a cellular layer) [548,549]. In another study, it was hypothesized that, due to their streamlined shape and increased contact surface with the endothelium, disk-shaped particles would more effectively stick to endothelial cells in the presence of flow-induced shear stress [550]. Recent research has also shown that particle elongation can enhance particle adhesion to protein-coated surfaces in microchannels [524,527]. According to an in vitro model examining the impact of the main physical properties such as particle shape, size, and density on a nanoparticle's inclination to marginate toward the vessel walls in microcirculation, varied kinds of nanoparticles, including liposome and metal particles, with a range of diameters (60 and 130 nm), densities (1–19 g ml$^{-1}$), and shapes (spherical, rod), were found to exhibit dissimilar deposition patterns as a result of varying margination rates. As a result, it has been reported that

smaller-sized and oblate-shaped particles are favored, owing to their greater margination rates [499].

4.14.7. Nanoparticles for Advanced Immunotherapy

To cure conditions such as cancer, immunological or metabolic system disorders, dementia, etc., significant effort has been put into developing nanogels, nanoparticles, and nanofibers. For these possible applications, research on cellular defenses against nanoparticles becomes crucial [551]. Cancer immunotherapy is projected to be an efficient and profitable method of treating cancer, alongside surgery and radiotherapy. By addressing many of conventional immunotherapy's drawbacks, such as ineffective drug delivery, it has been reported that nanoparticle-based advanced immunotherapy has the potential to increase the effectiveness of immune agents. In other words, it has been reported that multifunctional nanoparticle-mediated immune therapeutic agents can make it possible to track the progress of treatment, facilitate intracellular penetration, specifically targeting cancer cells, and increase the immunogenicity of antigens [552].

When creating nanoparticles, particle shape is another important consideration since it controls how well nanoparticles move through blood flow channels, trigger immune reactions, and behave inside internalized cells [553]. Nanoparticle asymmetry, which produces particle comminution and rolling towards the blood vessel wall underflow, has also been reported to promote nanoparticle penetration and distribution in solid tissues and tumors. It has also been suggested that nanoparticles of the same size with different shapes may interact differently with cell membranes and accumulate in cells differently [554]. Since the particle shape is considered to be one of the important physical factors that have a great influence on the interactions of nano biomaterials and cells [555,556], the most recent advancement in the field is the use of non-spherical particles to increase the therapeutic efficacy of anticancer drugs. It has been reported that antigens delivered in non-spherical nano vehicles also changed the immune reaction of nanoparticles. Therefore, some researchers have focused on techniques for creating non-spherical particles and examining how particle shape affects cellular absorption, biodistribution, tumor targeting, and the elicitation of immune responses [45]. Furthermore, it has been demonstrated that particle shape is key for the margination of nanoparticles. Unlike spherical particles, oblate-shaped particles are subject to torques resulting in rolling and rotation [548,549], and this complex dynamic causes translational and rotational motions. In a comparative study examining the effect of particle shape using a gold nanorod and a gold nanosphere of similar size, it has been found that the nanogold rod exhibited approximately eight times more precipitation than the nanogold sphere, in agreement with previous theoretical and experimental studies [549,557]. When the impact of particle size and shape on the margination efficiency and, consequently, on their adhesion potential has been investigated using simulations, it has been concluded that ellipsoidal particles should adhere more effectively than spherical carriers, since ellipsoidal particles have a greater surface area for adhesive contacts and slower tumbling motion within the red blood cell free layer. Nevertheless, because of their slower rotational dynamics close to a wall, ellipsoidal particles are believed to have better adhesion efficiency than spheres [539]. In studies suggesting that uptake, distribution, and cellular processes are also affected by particle shape [553,558], it has been stated that elongated nanoparticles often exhibit a higher uptake compared to spherical ones because of their capacity to attach to cells more strongly [559], due to the fact that spherical particles' curved surfaces offer fewer binding sites for contact with the surface of the cell. It has been reported that the elongated nanostructures, on the other hand, have a greater surface area to volume ratio, ensuring an efficient interaction with the cell surface [560,561]. For instance, in a study with the rod, discoid, cylindrical, triangular sharp-shaped, and semi-ellipsoidal nanostructures, it was reported that the particle shape is more effectively internalized by cells as it deviates from spherical [562,563]. It has also been reported that disc-shaped nanoparticles show more efficient internalization than rod-shaped ones [564]. Another study found that spherical nanostructures showed lower cellular uptake compared to

cylindrical ones [565]. It has been reported that differently shaped nanoparticles have a crucial effect on their transport behavior under blood flow in the microcirculation [499]. Moreover, the production of a series of shape-controlled $Cu_2O$ nanoparticles with the same dimensions but initially different crystal planes [cubes (100), octahedra (111), and dodecahedra (110)] is considered an important development in the field [556]. It has been reported that, as the shape of nanoparticles deviates from spherical shape, their efficiency increases, their circulation time increases, and their uptake performance changes according to the order of rod > cubic > cylindrical > spherical [562]. Another study found that spherical nanoparticles with faster blood circulation and reduced margination effects also have a high processing speed after internalization, whereas rod-shaped nanoparticles penetrate solid tissue tumors more easily than other shapes while processing speeds slow down after internalization. In addition, it has been reported that spherical nanoparticles give a Th1-based immune reaction, while rod nanoparticles give a Th2-based immune reaction [552,566]. It has been found that the presentation of antibodies on the surface of non-spherical particles increases antibody specificity as well as avidity towards their targets in another work investigating the specific and nonspecific uptake in three breast cancer cell lines (BT-474, SK-BR-3, and MDA-MB-231) using spherical, rod, and disc-shaped polystyrene nanoparticles and microparticles and trastuzumab as targeting antibodies. The fact that rod-shaped particles perform higher specific uptake and lower nonspecific uptake than spherical ones in all cells has been attributed to the unique role of their shape in deciding the binding and unbinding of particles to the cell surface. Trastuzumab-coated rod particles were found to not only show increased binding and uptake but also greater reductions in BT-474 breast cancer cell growth in vitro to a degree not possible with soluble versions of the antibody. It has been reported that by using pure chemotherapeutic drug nanoparticles manufactured from camptothecin instead of polystyrene particles, the impact of trastuzumab-coated rod particles on cells is significantly increased. The remarkable advantages for therapeutic and diagnostic forms of particulate antibodies were made possible by the fact that trastuzumab-coated camptothecin nanoparticles inhibit cell growth at a dose 1000 times lower than needed for similar growth inhibition using soluble trastuzumab, and 10 times lower than using BSA-coated camptothecin [567].

4.14.8. Nanoparticles Used in Catalysis

To achieve sustainability and lessen environmental worries while maintaining a good standard of life, catalysis is a cornerstone of green chemistry and a key technology. It is thought that future sustainable processes at the intersection of food, water, and energy will be built on better catalytic materials that exhibit higher activity, selectivity, and stability. It has been noted that making better heterogeneous systems for various catalytic applications requires a fundamental understanding of how catalysts work. It has been reported that well-defined catalytic structures are crucial for the growth of fundamental information about catalytic processes that will permit the efficient design and modification of catalytic materials. It is argued that the first and most important step in better understanding catalytic systems is to find and identify active sites that can be facilitated and accelerated by materials with well-defined shapes. It has been reported that once the active sites have been identified, materials can be developed to maximize their density by creating enhanced systems with suitable morphologies for the best catalytic results [568]. Therefore, it is known that the shape of the particle can also have an impact on the characteristics of particulate materials used in catalysis and environmental management. For instance, a study demonstrating how catalytic selectivity may be tuned by changing the particle shape reported that it may have important practical consequences to be able to adjust the selectivity of olefin isomerization reactions employing catalysts with well-defined particle morphologies [569].

It has long been known that structural variables can affect heterogeneous catalysis [570–573]. Since the majority of solid catalysts are made up of a finely distributed active phase, often a pricey transition metal onto a porous support with a high surface

area, the created microscopic particles are expected to exhibit a wide range of shapes, leading to numerous surface structures, as well as high surface-to-volume ratios, which are necessary to maximize their utilization. It has been stated that different surface planes can exhibit significantly different chemistries in studies using model systems, particularly single crystals, and ultrahigh vacuum conditions. For instance, it has been reported that the orders-of-magnitude variations in nitrogen activation rate affect different planes of iron surfaces [574,575]. It has been reported that these variations might offer a strategy to enhance catalytic selectivity by adjusting the surface structure [569].

In addition, it has been reported that the morphology of exposed facets that are involved in a catalytic reaction is determined by the shape of the particle. In other words, it has been reported that as the coordination number and shape of the atoms in the two facets change, the adsorbate binding modes and, hence, the variable reactivity change, resulting in different facets exhibiting various atom binding structures, leading to different catalytic behavior under the same reaction conditions. It is thought that the ultimate performance of catalytic materials can therefore be greatly influenced by the control of particle shape. Today's synthetic methods for creating nanocrystals are reported to be able to precisely control the shape of particles and consequently their exposed facets, with significant catalytic implications. It has been reported that metals are more likely to exhibit shape control, as there are many different ligands that can bind to metal surfaces and affect their growth. It has been shown that molecules take different shapes as a result of chemical adsorption on metal particles [568].

It is well known that hydrotreating of heavy crude oil is performed for removing undesirable impurities (metal, sulfur, nitrogen, aromatics, etc.) to comply with tougher fuel standards, especially those that limit the concentrations of sulfur and nitrogen compounds [576]. However, it has been reported that extruded Co-Mo or Ni-Mo alumina-supported catalyst particles in commercial plants are commonly used to remove the associated sulfur and nitrogen impurities by hydrodesulfurization and hydro-denitrogenation methods. It is also known that relatively rapidly reacting industrial hydro-desulfurization and hydro-denitrogenation catalysts are routinely pelleted into millimeter-sized particles of various shapes to minimize reactor pressure loss before being put into hydrotreating reactors. Thus, internal diffusion is considered to be inescapably a limiting factor for hydro-desulfurization and hydro-denitrogenation. It has been reported that the optimization of the physical features of the catalyst, such as particle shapes and pore structures, is more essential than the optimization of the catalyst chemical compositions, such as active components, promoters, and supports to maximize catalyst usage and lengthen the catalytic lifespan. In addition, it has also been reported that catalyst particle shapes and pore structure engineering are essential to overcome internal diffusion restrictions in the hydro-desulfurization and hydro-denitrogenation of gas oil [577]. The particle shape factor and PSD have been reported to have a substantial impact on the mass transfer behavior of HDS catalyst supports [578]. It has been found that the average hydro-desulfurization and hydro-denitrogenation reaction rate was mostly changed by the catalyst particle shapes to modify the hydro-desulfurization and hydro-denitrogenation performance factor [577].

## 5. Tuning the Shape of Particles by Mechanical Grinding

So far, the evolution of particle shape, which plays a crucial role in numerous engineering and industrial applications such as civil and chemical engineering and mining, food, and pharmaceutical industries [335], has received little attention. In fact, industrial applications require industrial minerals with the appropriate particle shape. For example, the GCC filler to be used in papermaking should consist of particles with an aspect ratio of 1–1.5 and a surface area of 5–12 m$^2$/g, and the grain shape is required to be rhombic for GCC to give a porous surface feature to the paper [411,579–582]. Another example is the spherical modification of the particle shape by grinding in a ball mill so that the flaky natural graphite battery can be used as the anode material in LIBs [449]. As another example, when comparing aluminum flake particles of various sizes and shapes, it can be given

that the electrical conductivity increases as the aspect ratio diminishes [583]. Therefore, particle shape must be controlled and manipulated because it influences many properties and behaviors of particulate materials in various processes and industrial utilization.

While the intrinsic characteristics of the material and the production process are known to play a major role in determining particle shape, it has been noted that the applied settings for any size-reduction system will typically rely on the type of equipment, the characteristics, and the size of the particles being shattered. In other words, it has been reported that the type of mill has a major effect on the particle shape, although the properties of the material are also a factor [584]. It is well known that the specific type of breakage affects the particle shape of mill products. For example, massive fracture generally leads to a non-spherical particle, which has sharp edges exposed at the intersection of progressive cracks, whereas attrition mode makes particles round by chipping of the edges and corners or abrasing of the surface. As particle sizes and applied stresses often follow normal distributions, it is very natural to obtain a distribution of product shapes. Therefore, milling conditions that use an appropriate breakage mode play a crucial role in controlling the morphology of the final product.

Moreover, it might be feasible to exert some control over shape by choosing the right device, since the effect seems to be stronger for low-energy equipment [585]. For example, ball mills, ring roll mills, and hammer mills have been reported to produce mill products with decreasing particle roundness [330,584]. In another case study, it has been reported that iron ore particles with high circularity and low aspect ratio values can be produced by using a ball mill (Figure 31a) while particles with low circularity and high aspect ratio values can be obtained by using HPGR (Figure 31b) [586]. Indeed, Holt [417] suggested that retention systems such as ball mills create more rounded particles, whereas single-pass devices such as roll crushers typically produce angular particles. In addition, the following specific shapes of particles can be obtained by using specific crushers and mills: cubical products from an impact mill (one pass) [32], cubical products from a roll crusher [587], sharp products from a gyratory crusher [32], spherical abrasive particles by a cyclone with walls [588], round products by autogenous grinding [114], the highest elongated quartz particles from rod mill products compared to the ball and autogenous mill [119], the higher elongated magnetite particles from the ball mill compared to the rod mill [589], and also the higher elongated platinum group mineral particles from the ball mill compared to the stirred mill [590]. Furthermore, it has been reported that the shape variability caused by the grinding of plastic scraps depends on the mill type and, therefore, impact mills have been reported to be used in a pilot plant operation in Germany as cutter mills produce thin plastic wires, which are more challenging to process [373].

In addition, it has been reported that materials having globular, cigar-shaped, and flaky particles can be prepared by grinding using a hammer, disc, ball mill and vibratory pulverizer, respectively. It has been reported that enrichment ratios of 28.3, 24.0, 23.6 and 21.7 were obtained, respectively, when these materials produced using a hammer, disc, ball mill, and vibratory pulverizer were subjected to separation with Knelson concentrator (gravity concentration). Therefore, when preparing these samples for enrichment with a Knelson concentrator, it has been reported that using a mill capable of producing round particles would be more advantageous than those producing cigar-shaped and flaky particles [369]. In a recent work examining the shapes of particles from all streams of the shaking table concentration circuits in a Turkish chromite concentration plant by using a recent technique such as DIA [107], it has been reported that table concentrate products are more rounded than tailing products (Figure 32) and it has been reported that this may be due to the different movements of the grains on the table (i.e., round particles tend to roll while elongated particles tend to slide on the deck). In another study, it was reported that different ground scheelite particles reveal cleavage surfaces at different rates, resulting in different product shapes, which is of primary importance in powder modification technology. It has also been reported to cause different surface reactivities to organics, which may help in optimizing the flotation process [591]. Since irregular particles are considered to

be more favorable than spherical particles in terms of higher flotation rate and flotation recovery in many studies [119,383,386,592], it has been reported that choosing the right grinding method with the appropriate grinding time can improve the flotation separation performance [593].

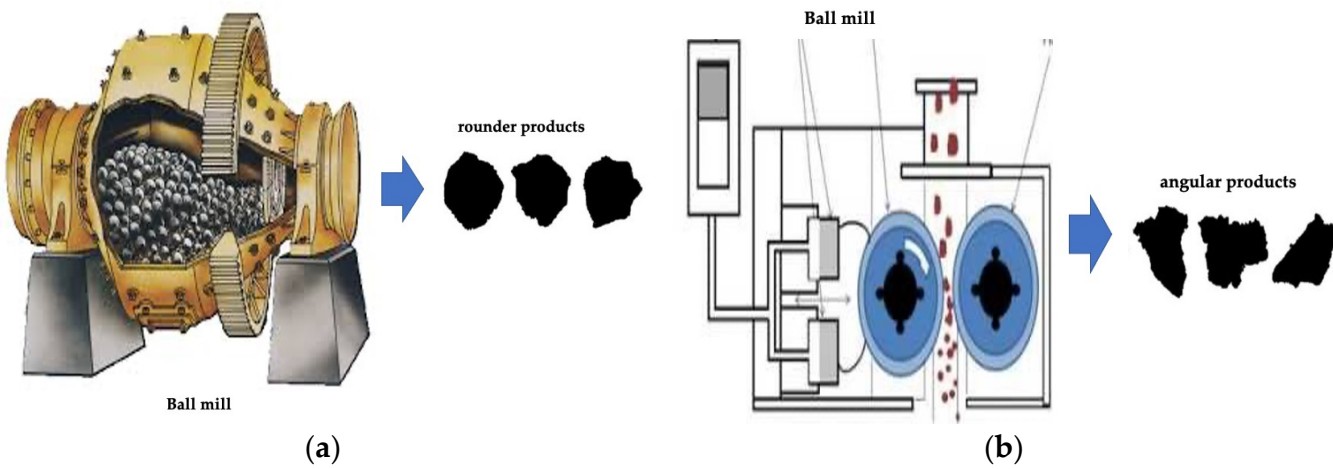

**Figure 31.** Comparison of particle shape by grinding using (**a**) a ball mill and (**b**) HPGR [586].

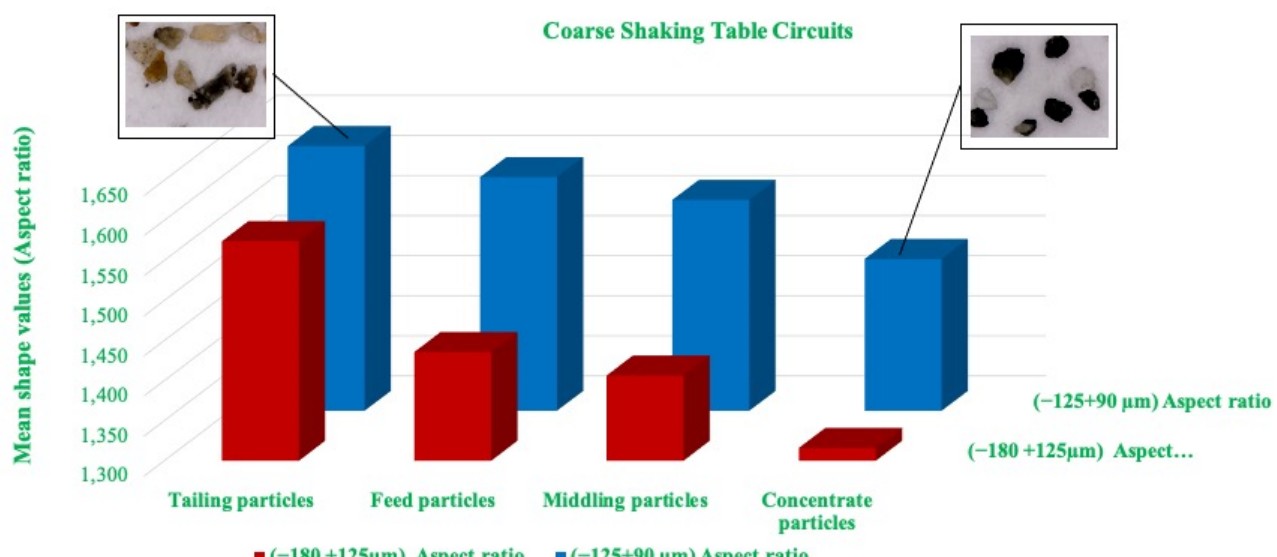

**Figure 32.** Average aspect ratio values determined by DIA for coarse shaking table circuit of chromite ore (modified from [107]).

When irregularly shaped particles, such as needles, are ground, their aspect ratio tends to be reduced and their shape tends to become more spherical. Generally, milling tends to make particles more spherical [270]. As observed by Dumm and Hogg [594], the rounding effect in ball mills increases with extended grinding times. Kaya et al. [585] confirmed this conclusion for coal and quartz minerals. As seen in Figure 33, more rounded particles can be obtained by extended grinding [593]. For an industrial example, it is reported that aluminum powders and flakes, which are still utilized for the majority of produced pigments nowadays, can be produced by wet ball mill grinding for about 5–40 h with lubricants [245]. A similar example is the conversion of graphite, which is used as an anode material in LIBs, from a flaky structure to a spherical structure in the first stage of the process, since it gives higher anode performance in terms of packing density [450].

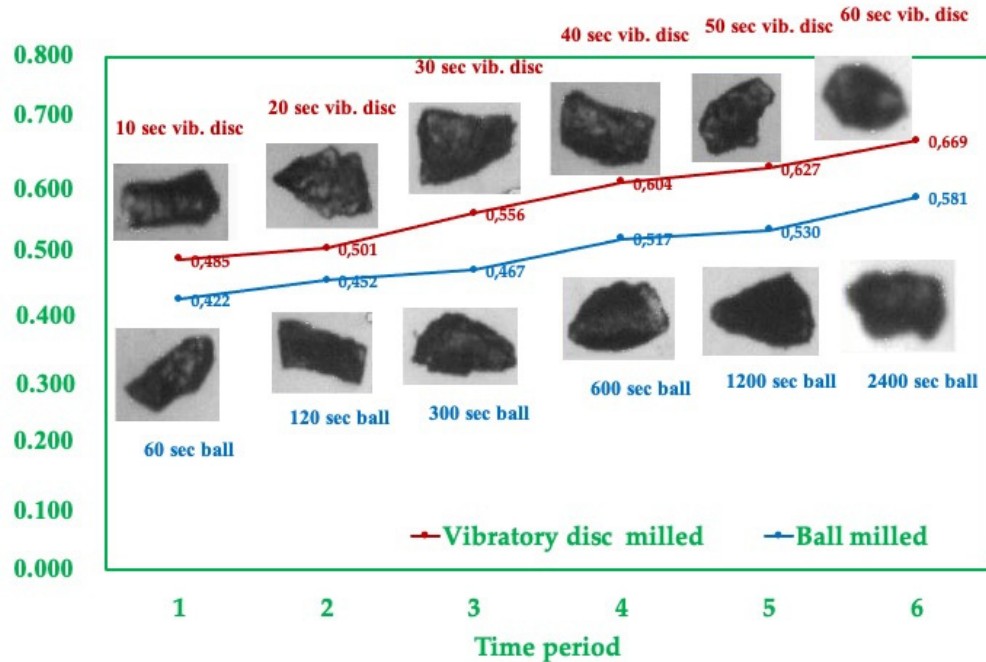

**Figure 33.** Changing particle shape depends on grinding time by using different mills [593].

In light of all the information mentioned above, tuning the particle shape of the material by selecting the appropriate size reduction device is significant for lower energy consumption, lower cost, higher quality industrial end-product, and higher separation process recovery.

## 6. Modeling

Since particle shape is important for understanding the behavior of particles in the process to increase efficiency, time-dependent shape models can be used to estimate the milling time required to grind the particles (with lower energy and cost) for the intended use. For instance, recent research has revealed that the shape of the particles ground by a specific mill, such as ball and vibratory disk mills, can be modeled using dynamic image analysis. Mathematical models have been derived that can be used to accurately predict average shape values such as circularity and aspect ratio based on milling time. It was concluded that the average circularity and aspect ratio values of the vibratory disk and ball mill change with grinding time by obeying the mathematical model of linear ($C_{av.} = at + b$) and power fitted ($1/BRAR_{av.} = a \times t^b$) equations with more than 94% ($R^2$) correlation coefficients [593].

## 7. Conclusions

A wide variety of industrial processes and applications have begun to benefit from advances made through the study of particulate systems by applying the principles of particle characterization in the production of chemicals, batteries, catalysis, nanomaterials, pharmaceuticals, ceramics, and foods because particle shape has a substantial impact on how they fracture, react, sinter, float, agglomerate, and fluidize. Therefore, it is one of the key factors in predicting the behavior of particles either individually or in groups in various engineering applications from macroscale to nanoscale. Moreover, it is well known that as the shape of the particles deviates from the ideal sphere, they behave differently from the homogeneous spherical particles that are traditionally used as models. In other words, the differences in particle size and shape of many materials from the macroscale to the nanoscale are responsible for the difference in their behavior. Thus, it has been revealed from the literature that the shape of the solid particles has a significant impact on the selection of suitable process parameters and the accomplishment of the ideal

properties for the final product in many engineering disciplines for various particulate systems from aggregate to nanoparticles. Furthermore, understanding the role of mineral grain shape in applications or processes using grains helps them to be used more efficiently and successfully. The following examples are striking results in the various areas explored in this review.

It has been reported that spherical particles are more suitable for concrete with high compressive strength, ultrasonic pulse velocity, unit weight, and slump values, while highly angular crushed aggregates are preferred for the concrete pavement and ballast aggregate particles in the concrete pavement and the railway track because of the good interlocking they offer.

It has been suggested that it is more advantageous to use rod-shaped proppants, which can produce cracks with higher conductivity, to provide hydraulic fracturing in oil and gas extraction.

It is known that glass bead particles used in highway road paints should have round shapes, as they provide sufficient retroreflectivity as well as strength.

According to studies on the viscosity and rheological properties of suspended particles, it has been found that the intrinsic viscosity of the solution rises rapidly when the shape of the grains changes from spherical to elongated. Similarly, it was discovered that while all other conditions remained the same, non-spherical particles settled more slowly than spherical particles, since rotating anisotropic particles such as plates and needles are very effective at filling up empty space.

It has been reported that non-spherical particles are favored for high packing density and thus high compaction for designing pneumatic conveying and multiphase reactors in bulk solids handling and processing for the agricultural, ceramic, food, mineral, mining, and pharmaceutical industries, whereas round particles have higher discharge rates. In addition, they demonstrate high flowability due to their free movements, as in additive manufacturing.

Spherical metal additive particles are said to offer the best melting and spraying properties and boost feedstock flowability.

In the pharmaceutical industry, it has been reported that spherical particles can be compressed more than irregular ones because they can be reorganized more readily, while the highest reactivity was observed in flaky particles.

While it was determined that angular particles are difficult to pack because they cannot rotate and slide such as spherical particles, elongated powder compacts have lower deformation than spherical powder compacts or sand particles used in various engineering disciplines such as material engineering, mechanical engineering, hydraulics, and geotechnical engineering.

In powder metallurgy, irregular particles with angular shapes and relatively high aspect ratios were reported to have higher abrasive properties. In addition, spherical particles have reportedly been found to be the best choice for thermally sprayed coatings because they flow effectively and provide optimal conditions for particle deposition. Despite their low flow rate, non-spherical particles are extensively utilized in conventional powder metallurgy due to their advantageous pressing and sintering properties. They have also been said to have additional points of contact with nearby particles during pressing as a result of their irregular shape, which improves the possibility that they will interlock and reinforce the green compact before sintering.

In mineral processing operations, separation also depends on particle shape. In classification operations by hydrocyclone, it has been noted that spherical particles tend to be separated more effectively than flaky particles because platy particles such as mica, even if they are relatively coarse, frequently tend to be discharged as overflow. In the electrostatic separation, since it is determined that the spherical particles are thrown the farthest and the flake particles are thrown the nearest, it may be possible to separate the particles with different shapes. Similarly, differently shaped particles can be easily separated in the shaking table, as in the separation of mica from feldspar. In the process of

separating hydrophobic mineral particles from hydrophilic mineral particles by flotation, it is a common opinion that irregular particles are preferred over spherical particles because they have a higher contact surface for the attachment of the particles to the bubble, and they also have sharp edges for the successful rupture of the liquid film. In contrast, round particles are also known to negatively affect flotation by exhibiting higher entrainment than angular, elongated particles, especially in the size range below 45 μm. Therefore, it has been concluded that it is possible to obtain the best-behaving particle shape in the process by using the appropriate mill for industrial use and controlling the grinding time.

It has been discovered that layered-shaped fillers with high aspect ratios are favored in reinforcement properties since they transfer stress more effectively compared to spherical fillers (such as mica and silica) for rubbers. On the other hand, it has been reported that particles with a low aspect ratio have a disadvantage in affecting modulus and strength compared to particles with a high aspect ratio for dolomite fillers used in fiberglass fabrication, but are advantageous in increasing impact strength. It is also accepted that the platy-shaped mineral fillers give the best-desired results in paints and coatings, cosmetics, abrasives and ceramics, and papermaking.

Raw natural flake graphite, which is an indispensable raw material as the anode material of lithium-ion batteries, is converted into a spherical shape to give a higher electrochemical performance.

In magneto-rheological fluids, which are beneficial for many applications in a variety of engineering sectors, rod-like microparticles are favored. Moreover, the transition from a superparamagnetic state to a ferromagnetic state has only been possible thanks to nanochain structures developed in recent years. On the other hand, it has been reported that spherical-shaped nanoparticles are generally preferred for nanofluids, which are used for heat transfer.

Many studies on the dissolution of drugs consisting of pharmaceutical particles have reported that non-spherical particle colloids show low dissolution rates due to the increase in the mean hydrodynamic boundary layer thickness. In addition, ellipsoidal nanoparticles have slower rotation dynamics in blood, which makes them ideal for drug delivery. Finally, it is also claimed that the performance of vascular targeted carriers, which offers a new chance to improve disease diagnosis and therapy, may be enhanced by departing from the traditional spherical shape. In studies where nanoparticles are used for advanced immunotherapy, it has been shown that as the shape of particles deviates from spherical shape, their efficiency increases, their circulation time increases, and their uptake performance changes.

## 8. Future Recommendations

Particle technologies have the potential to be very successful worldwide, but the widespread adoption and applications of this technology are still not at the desired level. It has been suggested that the shape of the particles should be taken into account when operating and choosing processes or applications and more quantitative research should be studied on these connections, since particle shape is a primary property affecting many secondary properties for many systems from macroscale to nanoscale.

Although this work presents some important conclusions on the impact of particle shape on their properties and behaviors in various processes for many fields of engineering, there are many areas where further works experimentally investigating the role of particle shape on their properties and processes could be undertaken. Thus, it is hoped that in the future more quantitative investigations related to particle shape effect should be performed in the field of industrial applications. In other words, since this review gives a significant basis for these future works, future research studies are necessary for the investigation of the underlying mechanisms of the effect of particle shapes on the macroscopic to nanoscopic behavior of various materials using experimental and numerical approaches. For instance, nanoparticles can offer cutting-edge technologies capable of customizing

special properties by controlling the particle size, shape, and other factors in many fields, especially medical applications.

Furthermore, it is believed that this review, which releases the linkage between particle shape and many secondary properties, will help further development in the field of beneficiation processes, such as flotation, electrostatic, gravity, magnetic separation, leaching behavior of minerals, and sorting. However, rigorous and comprehensive research must be carried out to pinpoint these links and to bring more benefits to technological development and the economy.

For further studies to investigate the effect of particle shape, the ability to determine the geometric properties of particles using tools such as computerized image processing vision systems, which are now supported by the latest methods such as artificial intelligence (AI), will improve progress in this regard. Moreover, it is important to use new approaches such as the "deep learning model" for the applicable estimation of particle shape parameters since commonly used techniques are limited by user-dependent uncertainty and complex, computationally challenging algorithms.

**Funding:** This research received no external funding.

**Data Availability Statement:** Not applicable.

**Acknowledgments:** Not applicable.

**Conflicts of Interest:** The authors declare no conflict of interest.

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
