# Peer review of "A Review of Particle Shape Effects on Material Properties for Various Engineering Applications: From Macro to Nanoscale"

_minerals, doi:10.3390/min13010091_

Round 1

Reviewer 1 Report

This is a well-organized and systematic review works investigating the influence of particle shape on the various engineering performance and material properties. This paper is methodical and logical. In this review, the definition of particle shape parameters, the characterization method of particle shape parameters, the application of particle shape characteristics in engineering materials and the shape control method were summarized. I think this paper is a good review of the effect of particle shape on the engineering performance and material properties. This work has great significance for the popular science education and for scholars to understand the achievements related to particle shape. 

However; two revisions should be made for a good paper for this journal:

1. The quality of all figures in this paper need be improved. At present, the readability of figures in the paper is poor.

2. In section of 8, I would like to see some suggestions about how to improve the production efficiency of engineering project such as particle flotation and settlement from the view of particle shape.

3.  I think these papers as follow are help for the discussion in this study, please quote them in reference.

Journal of Cleaner Production, 2018, 195(10):470-475.

As final remark, I recommend the acceptance of this manuscript after minor revision.

Author Response

Response 1: Thank you for your constructive comments. The quality of the figures is improved

Response 2: Thank you for your constructive comments. The effect of particle shape on the flotation was revised based on the suggestion of the reviewer. Section 4.5 consists of the effect of the shape of settling particles in gravity sedimentation.

Response 3: Thank you for your constructive comments. Reference 387 (Ma et al., 2018) has already been used in our original manuscript. Besides, Line 1654-56 and Line 1668-1670 have already discussed the possible reason for higher floatability.

(“It has been found that regular and irregular particles behave differently in the attachment of particles to a bubble for flotation since it has been reported that the edge shape of particles and their wettability both influence the adhesion force in flotation” and

“It has been demonstrated that in the flotation system since the prismatic particles adhere to the bubble surface more easily than the round particles [398], sharp edges can successfully cause the liquid film to rupture and increase flotation recovery [24,384,399].”)

Based on the suggestions of Reviewer, the following discussion was added between Line 1671 and Line 1677 as:

According to Ma et al. [387], during particle-bubble adhesion, the angular shape can easily tear the water film, which shortens the duration of attachment and speeds up floating. Additionally, compared to non-spherical particles, spherical particles have a smaller contact surface for the attachment of the particles to the bubble, which increases the likelihood that particles will detach from a rising bubble. Non-spherical particles are more likely to float because they have a greater chance of attachment and a lower chance of detachment.

Reviewer 2 Report

The manuscript is a good review describing the role of particle shape in a wide range of powder processing fields, ranging from nano- to millimeters. The content of this review is of much importance because the particle shape exerts a large effect even in the coarse-size region, and is particularly dominant in the fine-size region where the influence of gravity becomes small. In addition, many natural (mineral) resources are relatively bulky, but artificial (waste) resources differ greatly in shape, so the impact is enormous also for recycling engineering. The contents of this review are summarized from the measurement of particle morphology, which is expected to increase in importance near in the future, and the application.

Author Response

Response 1: Thank you for your constructive comments. I hope this review will contribute to the literature and is helpful at the industrial scale, and most of all inspire new research and developments, in the behavior of particulate materials in the processes or applications in various industries. Moreover, maybe it will lead to new applications and new treatments in a new field.

Reviewer 3 Report

This is a comprehensive and informative review of particle size and certainly motivates for the consideration of particle shape in all areas. The paper is very long and covers a wide range of applications. The figures are of quite a low quality, it is essential to improve the figure quality to make the paper publishable. Further to this, the paper has a very general focus on a large number of industries and research areas, the author should consider whether Minerals is the best place for a paper of this nature, undoubtedly a more general journal would find wider readership, generally the minerals industry already considers particle size as a very important factor.

Author Response

Thanks for your valuable points and comments. The quality of the figures is improved. Since the journal “Minerals”, which is an open-access journal, not only covers mineral processing but also other mineral industry area, I hope this review paper will contribute to the literature, is helpful at an industrial scale, and serves all disciplines. As the particle size decrease especially down to nano size its shape gains vital importance.

Reviewer 4 Report

A review of particle shape effects on material properties for various engineering applications: From macro to nanoscale present different synthesis routes of NPs preparation and its various applications which is helpful at industrial scale. The study will facilitate nano-engineers and help in application of NPs at industrial scale. However, there are some deficiencies which must be addressed.

In abstract the authors are focused on introduction. Here the data collection and main findings should be discussed. Also discuss main sections of the review.

In figure 1 the figure quality is poor.

Figure 3 quality is very poor text is not readable.

Line 104 should be cited with relevant study. The following studies would be helpful.

https://doi.org/10.1007/s10534-022-00417-1, https://doi.org/10.3390/coatings12101505,

Section 3.5 write full name of the heading.

Line 1374 which recent study add reference.

Line 1994 should be cited with relevant study. Here reference is missing so the following study could be helpful.

https://doi.org/10.1002/jemt.23553,

The article is very lengthy but include very useful information and covered different aspects related to the topic.

However, in many place the authors discuss about some study but did not cited so the article should be revise thoroughly.

Specific findings from the review must be mentioned in the conclusion.

Harmful effects and major hazardous materials must be discussed in the introduction.

Conclusion looks like a background of the study. Conclusion must be findings based and proposing gaps ad future recommendations.

Author Response

Response 1: Thanks for your valuable comment. It was revised based on the suggestions of Reviewer. This article, which has been compiled with what is known in the literature, aims to shed light on making more in-depth research. Response 2: Thanks for your valuable comment. The quality of the figure 1 is improved. Response 3: Thanks for your valuable comment. The quality of the figure 3 is improved.

Response 4: Thanks for your valuable comment and suggestions. As can be clearly seen from the paragraph it was related to the reference [23]. Since Introduction is very lengthy, the suggested papers (https://doi.org/10.1007/s10534-022-00417-1, https://doi.org/10.3390/coatings12101505 and https://doi.org/10.1002/jemt.23553) were cited in Section 4.14 since they are related to the nanoparticles for agriculture and biological applications.

Response 5: Thanks for your valuable comment. The full name of the heading (3.5. Brunauer, Emmett, and Teller (BET) technique) was written based on the suggestion of the reviewer.

Response 6: Thanks for your valuable comment. It was checked and found that it is related to the reference [18].

Response 7: Thanks for your valuable comment. They were checked and found that they are related to the references[457,460–468]. Since Introduction is very lengthy, the suggested paper (https://doi.org/10.1002/jemt.23553) was cited in Section 4.14 since it is related to the nanoparticles for biomedical applications.

However, in many place the authors discuss about some study but did not cited so the article should be revise thoroughly.

Response 8: Thanks for your valuable comment. It was checked and found that all the references are correct. In addition, I try to summarize the shape effects on various applications (at least 14 sections) as shortly as possible by core references (more than 595), since this review deals with the shape effects of all particulate materials in various industries from macro to nanoscale. It was revised based on the reviewer’s comments. Therefore, specific findings from the review were mentioned in the conclusion.

Response 9: Thanks for your valuable comment. Specific findings from the review were mentioned in the conclusion based on the reviewer’s comments.

Response 10: Thanks for your valuable comment. Since harmful effects and major hazardous materials are beyond the scope of the present review. Electronic waste is important for us to recover metallic particles by physical and chemical methods as mentioned in Lines 1427-1429. So metallic particles were already mentioned in Lines 33; 43; 714; Section. 4.8. Metal additive particles; Line 1259; Lines 1271-1332. Plastics were already mentioned in Lines 35 and 40; Lines 716 and 740, since they are particulate materials. Therefore, it is mentioned only in the relevant places in order not to disturb the semantic integrity of the Introduction. Besides, Introduction is very lengthy and this work emphasizes that mineral separation depends on the particle size and shape, the separation of plastics and electrostatic separation of electronic waste materials were used as an example in Sections 4.10.3.2. and 4.10.3.3., respectively. Besides, the following statements were already used for this aim as;

Line 1540; “Recently, it is considered an efficient and clean method for recycling metals such as copper, aluminum, lead, tin, and iron from electric and electronic waste materials, which can be a harmful effect on our environment [354–359].”

 Line 1618; “It has been reported that shape factors, combined with a mathematical description of the partition curve, helped in the process modeling, selection, and design of dense medium separation systems for recycling plastic [374]”.

Based on the suggestions of the reviewer harmful effects and major hazardous materials were discussed in the appropriate Sections 4.10.3.2. where sorting of plastics and electronic wastes to recover metallic particles.

Lines 1429-1432 were added as: “Furthermore, in the digital age we live in, it is inevitable to separate and recycle metals and plastics in order to prevent the waste generated by all electrical and electronic vehicles from harming the environment and to bring them into the economy.”

In addition, Lines 1611-1620 states some discussion about plastics and recycling as:

“Nowadays, it is becoming increasingly vital to separate post-consumer plastics with various polymeric compositions to reduce waste and increase recycling [374]. When the impact of particle shape on the dynamic dense medium separation of plastic was investigated, it has been found that using salt solutions as a medium, single particle sizes and shapes may be separated with great precision in a Tri-Flo separator”

Response 11: Thanks for your valuable comment. It was revised based on the reviewer’s comments. Conclusion was revised as findings based and proposing gaps ad future recommendations were already given in the next Section 8 Further research.

General response: Finally, the English language and style were improved by using a grammar software tool